



# Intercomparison of photogrammetric platforms for spatially continuous snow depth mapping

Lucie A. Eberhard[1, 2], Pascal Sirguey[3], Aubrey Miller[3], Mauro Marty[4], Konrad Schindler[5], Andreas Stoffel[1], Yves Bühler[1]

[1]WSL Institute for Snow and Avalanche Research SLF, Davos Dorf, 7260, Switzerland
[2]D-Baug, ETH Zürich, Zürich, 8093, Switzerland
[3]National School of Surveying, University of Otago, PO Box 56, Dunedin, New Zealand
[4]Swiss Federal Institute for Forest, Snow and Landscape Research WSL, Birmensdorf, 8903, Switzerland
[5]Photogrammetry and Remote Sensing Group, ETH Zurich, Switzerland

*Correspondence to*: Lucie A. Eberhard (lucie.eberhard@slf.ch)

**Abstract.** Snow depth has traditionally been estimated based on point measurements collected either manually or at automated weather stations. Point measurements, though, do not represent the high spatial variability of snow depths present in alpine terrain. Photogrammetric mapping techniques have made significant progress in recent years and are suitable to accurately map snow depth in a spatially continuous manner, over larger areas, and at various spatial resolutions. However, the strengths and weaknesses associated with specific platforms and photogrammetric techniques, as well as the accuracy of the photogrammetric performance on snow surfaces have not yet been sufficiently investigated. Therefore, industry-standard photogrammetric platforms, including high-resolution satellites (Pléiades), airplanes (Ultracam), Unmanned Aerial Systems UAS (eBee+) and ground-based (single lens reflex camera), were tested in a timely manner for snow depth mapping in the alpine Dischma valley (Switzerland) in spring 2018. Imagery was acquired with airborne and space-borne platforms over the entire valley, while UAS and ground-based photogrammetric imagery were acquired over a subset of the valley. For independent validation of the photogrammetric products, snow depth was measured by probing, as well as using remote observations of fixed snow poles. When comparing snow depth maps with manual and snow pole measurements the root mean square errors (RMSEs) and the normalized median deviations (NMADs) are 0.52 m and 0.47 m for the Pléiades snow depth map, 0.17 m and 0.17 m for the Ultracam snow depth map, 0.16 m and 0.11 m for the UAS snow depth map. Ground-based had to few measurements to be statistically relevant. When using the eBee+ snow depth map as ground truth, the RMSEs and NMADs are 0.44 m and 0.38 m for the Pléiades snow depth map, 0.12 m and 0.11 m for the Ultracam snow depth map, 0.21 and 0.19 m for the ground-based snow depth map. Because of the accuracy and precision of the Ultracam dataset we finally compared the Ultracam snow depth map to the Pléiades snow depth map over a large part of the Dischma valley and calculated a RMSE of 0.92 m and a NMAD of 0.65 m. By comparing for the first time more than two platforms, this study provides comparative measurements between platforms to evaluate the specific advantages and disadvantages of them for operational, spatially continuous snow depth mapping in alpine terrain over small and large areas.



Keywords: Snow depth, Photogrammetry, Satellite, Airplane, UAS, Ground-based

# 1    Introduction

The range of applications for accurate high-resolution snow depth mapping is diverse. Snow depth is defined as the vertical distance from the base to the surface of the snowpack (Fierz et al., 2009) and can vary significantly over short distances horizontally within a meter-scale (Lundberg et al., 2010; Griessinger et al., 2018; Dong, 2018). Several fields rely on accurate information about how snow depth changes across a landscape. First, accurate snow depth distribution estimates are necessary for snow water equivalent (SWE) modelling in snow hydrology (Steiner et al., 2018). Second, modelling snow drift accumulations and detecting avalanche release zones to estimate avalanche hazard require reliable information on snow depth (Schön et al., 2015). Furthermore, mapping the mass balance of avalanches is important for numerical avalanche dynamic simulation tools such as Rapid Mass Movement Simulation (RAMMS) (Christen et al., 2010; Bartelt et al., 2016). In addition, snow depth mapping enables rapid documentation of avalanche accidents, which is required immediately after the event due to rapidly changing weather and snow conditions (Bühler et al., 2009; Lato et al., 2012; Korzeniowska et al., 2017). The tourism industry would also benefit from high-resolution snow depth maps in ski resorts to better redistribute snow on slopes throughout the season (Spandre et al., 2017). Furthermore, mapping snow depth at high spatial resolution is desirable to support the monitoring of sensitive alpine ecosystems in a changing climate (Wipf et al., 2009; Bilodeau et al., 2013) because the seasonal snow cover is a rapidly changing climate characteristics (IPCC 2007).

Traditionally, snow depth measurements have been obtained as point measurements manually or at automated weather stations. These measurement methods have obvious limitations because point measurements give little indication about the spatial distribution of snow depth (Nolan et al., 2015). Snow depth distribution can eventually be approached by interpolating sparse values (Cullen et al., 2016). Furthermore, manual snow depth probing disturbs the snow surface and may leave significant areas of terrain unsampled because of avalanche danger or challenging topography.

Emerging technologies such as laser scanning have already produced continuous snow depth maps (e.g., Hopkinson et al., 2001; Hopkinson et al., 2004; Deems et al., 2013; Telling et al., 2017). Airborne laser scanning (ALS) covers typically large areas with an average spatial resolution of 1 m and achieve a vertical accuracy of 0.1 m (Deems and Painter, 2006; Deems et al., 2013). However, ALS remains expensive and requires an available aircraft (Lopez-Moreno et al., 2017). Also, not every laser scanner is suitable for snow depth mapping. A laser (airborne or terrestrial) with a wavelength of 1064 µm is the best choice for obtaining acceptable results on snow due to the physical properties of the snowpack, i.e. dry or wet snowpack (Deems et al., 2013). Furthermore, a small laser beam footprint is desirable and can be achieved by ensuring that the laser beam remains perpendicular to the snow surface. Terrestrial laser scanning (TLS) can achieve accuracies below 0.10 m beyond



1000 m (Prokop, 2008). Recently, very long-range TLS have been used for analysis of the spatial distribution of a snowpack up to 3000 m with absolute errors ranging from 0.2 to 0.6 m (Lopez-Moreno et al., 2017).

Laser scanning techniques remain relatively expensive compared with photogrammetry. Satellite-based, airplane-based, Unmanned Aerial Systems (UAS)-based and ground-based data have been used for photogrammetric snow depth mapping, although rarely compared in a single study. A first study using imagery from Pléiades satellites mapped snow depth at 2 m spatial resolution with a standard deviation of 0.58 m compared to manual measurements (Marti et al., 2016; Deschamps-Berger et al., 2020). Aerial images acquired with a Leica ADS80/100 optical scanner have allowed snow depth to be produced with a RMSE of 0.3 m (Bühler et al., 2015; Boesch et al., 2016). Using a consumer camera on a manned aircraft, a standard deviation of 0.1 m was determined for the snow depth compared to manual measurements (Nolan et al., 2015). Photogrammetric UAS surveys are a promising method used and characterised by several studies to map snow depth. With UAS data, vertical snow depth accuracies of 0.1-0.15 m have been achieved by several researchers (Vander Jagt et al., 2015; Bühler et al., 2016; De Michele et al., 2016; Harder et al., 2016; Cimoli et al., 2017; Redpath et al., 2018; Avanzi et al., 2018; Eker et al., 2019). Finally, ground-based photogrammetry has been used for snow observation, snow drift tracking and avalanche detection with accuracies of 0.1-0.3 m (Prokop et al., 2015; Basnet et al., 2016). Ground-based photogrammetry is currently the only method which can produce digital surface models (DSMs) of an avalanche flowing downwards during an avalanche release experiment (Vallet et al., 2001; Vallet et al., 2004; Dreier et al., 2016). Other techniques such as laser scanning have acquisition times that only allow the acquisition of DSMs before and after the avalanche release (Prokop et al., 2015). This makes ground-based photogrammetry a valuable monitoring solution, benefitting also from a relatively lower cost compared with other monitoring solutions such as TLS (Toth and Jozkow, 2016; Basnet et al., 2016).

Promising results from a range of photogrammetry techniques and platforms demonstrate the potential to operationalise photogrammetric snow depth mapping. To date, the available photogrammetric platforms have only been partially investigated w.r.t. their performance on snow (e.g., Bühler et al., 2017). A comprehensive assessment is needed to compare snow depth products from ground-based, UAS, airplane and satellites platforms against each other. Each platform has its advantages and disadvantages, but each must be able to cope with the challenges of imaging alpine environments, including steep terrain and rapidly changing weather conditions. A key advantage of photogrammetric satellite and airplane data is the large area that can be mapped promptly and the mapping of completely inaccessible terrain. Also, UASs allow decimetre-scale snow depth maps even when the accessibility of terrain is restricted i.e. because of avalanche danger and perform well in a winter alpine environment. UAS with photogrammetric platforms are fast, easy to handle and have a high recording availability, i.e. if the operator owns a UAS, they can fly whenever required. Ground-based photogrammetry achieves reasonable results for small areas of interest such as avalanche release zones if the areas are nearly perpendicular to the camera locations. Therefore, the different platforms available today for photogrammetric snow depth mapping can improve the research in alpine environments and fulfil the needs of spatially continuous snow depth mapping on different scale.





This study documents a photogrammetric intercomparison campaign on 6 April, 7 April, and 11 April 2018. For the first time, optical data from a high-resolution satellite, an airplane, an UAS and ground-based platform were collected over the same area and within a short time frame (6 days). To validate the photogrammetric results with independent snow depth values, manual

snow depth measurements were collected and fixed snow poles were used.

## 2    Test site Dischma valley and Schürlialp

The Dischma valley is an alpine valley in the region of Davos, Switzerland, which has been the focus of a range of snow-related studies (Baggi and Schweizer, 2008; Bühler et al., 2015). For a representative photogrammetric study, a test site with a diversity of terrain types was needed, including both artificially disturbed and undisturbed terrain. The Dischma valley covers

altitudes from 1550 m a.s.l. to 3150 m a.s.l. with prevailing northeast and southwest aspects. In the south part of the Dischma valley, the vegetation changes between flat alpine meadows on the bottom to bushes and alpine roses on the slopes and hilly alpine terrain on the upper slopes. The northern and lower elevation region of the Dischma is dominated by alpine forests. The year-round inhabited areas are located in the northern region of the Dischma, the alpine pasture areas in the southern part are only inhabited in summer.

Satellite and aerial data were captured over an area that included the Dischma and surrounding ridges covering an extent of approximately 140 km$^2$. For the UAS and the ground-based platforms, a smaller test site was selected around Schürlialp, covering ca. 4 km$^2$ and reaching up to ca. 2350 m a.s.l. on each side of the valley (Figure 1). In the Schürlialp test site the main aspects are northeast and southwest. The slope angle ranges from 0° to 45°, with a typical slope angle between 30° to 35°.

Interesting features of the Schürlialp area are gullies channeling downslope winds and producing contrasted snow depth distribution. Manual snow depth measurements as well as 15 fixed snow poles observable from a distance with binoculars provided reference measurements in the Schürlialp test site, including steep and inaccessible terrain.

Data from snow measurement stations distributed sparsely around our study site provides context for general snowfall patterns

in the Dischma valley during our study period. In particular, it documented less than 10 cm of snow melt between 6 and 11 April (Figure 2). Only the low elevation stations Davos Flüelastrasse (5DF, 1560 m a.s.l) and Matta Frauenkirch (5MA, 1655 m a.s.l) lost more than 20 cm of snow. At higher altitudes, a small amount of new snow was measured (2 cm above an altitude of 2400 m. a.s.l.). These measured values support our assumption that the change in snow depth minimal despite the time difference between data acquisition and a comparison of datasets could be made.





**Figure 1: Overview of the test sites: the red polygon is the area which was recorded by the Pléiades and the Ultracam imagery. The blue polygon is the area covered by the eBee+ imagery and the black polygons represent the area covered by the ground-based images. The purple triangles represent the location of automatic and manual snow measuring stations around Davos. The blue star in the inset map shows the location of the Schürlialp test site. (Swiss Map Raster© 2019 swisstopo (5 704 000 000), reproduced by permission of swisstopo (JA100118).)**



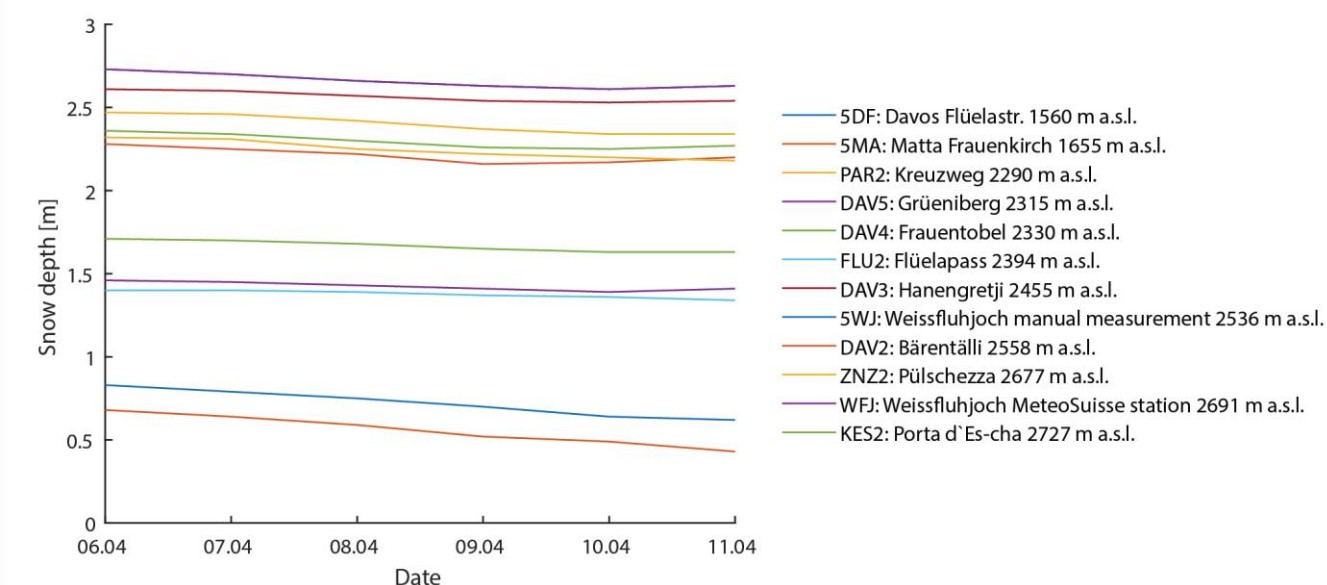

**Figure 2: Overview of the measured snow depth of the automatic and manual snow measuring stations around Davos for the period of photogrammetric data acquisition (see Figure1 for locations). (Data © SLF and MeteoSchweiz)**

## 3    Platforms and data

By acquiring satellite, airplane, UAS and ground-based data over a short timeframe (6 days), a comprehensive dataset bringing together small- and large-scale photogrammetric platforms is available for intercomparison. The Pléiades constellation involves two very high-resolution optical satellites with proven performance to derive digital surface models (Stumpf et al., 2014). The Ultracam Eagle M3 is a digital aerial high-resolution large format camera from the Ultracam series for state-of-the-art high-resolution aerial photogrammetry (Legat et al., 2016; Vexel Imaging GmbH, 2019). The eBee+ RTK is a survey

drone (UAS) equipped with high-resolution camera and a dual-frequency GNSS sensor capable of Real Time Kinematic positioning, all contributing to deliver very accurate digital surface models (Benassi et al., 2017). The camera used to capture ground-based data is a digital single reflex (SLR) Canon 750D. With this set of photogrammetric platforms, the Ground Sampling Distance (GSD) range from 0.04 m/pixel to 0.5 m/pixel. To achieve the triangulation of satellite data and the airplane data, independent ground control points (GCPs) and check points (CPs) were collected around Davos during summer. They

consisted of features such as roof corners, bridges and other clearly distinguishable man-made features. However, many of these points are not visible in the imagery either as they are covered by snow in winter or because of the low resolution (Pléiades). Therefore 10 additional control points were distributed throughout Dischma, six of them on the test site Schürlialp. Seven of the GCPs were laid out on 6 April and 3 more on 7 April. They consisted of 0.8 x 0.8 m white tarps with a black cross and a white square in the middle. All GCPs were positioned with a differential GNSS (Trimble Geo XH 6000) with a

position accuracy of 0.1m. In the following subsections, we present each platform and the corresponding data.



### 3.1  Satellite: Pléiades

A cloud-free Pléiades-1B image triplet was acquired on 7 April 2018 between 10:17 and 10:19 am. The panchromatic and multispectral bands (red, green, blue, near-IR) of Pléiades very-high resolution sensor achieve a spatial resolution of 0.5 and 2 m respectively. The 12-bit radiometric resolution of Pleiades imagery provides a dynamic range capable to resolve contrast in dark shaded areas, as well as across highly reflective snow surfaces. From 694 km above ground, the image triplet was acquired along a descending orbit tracking east of Switzerland (across-track incidence angles of 17.4°, 12.1° and 9.7°, respectively). Along-track incidence angles of -16.3°, 7.6° and 17.9° resulted in three stereo pairs with Base-over-Height ratio (B/H) of 0.42 (images 1 and 2: pair P12), 0.19 (images 2 and 3: pair P23) and 0.62 (image 1 and 3: pair P13). Although both P12 and P13 have B/H recommended for photogrammetric work (>0.25, Astrium, 2012), P23 is below the usual standard to process an accurate DSM due to acute parallax angles. Meanwhile, larger B/H ratio, such as stereo P12, can yield unresolved areas due to terrain obstruction in steep topography. In addition, complicated parallax can modify the appearance of ground features, in turn challenging stereo-matching. For this study, we processed the three stereo pairs and considered occlusion and accuracy of each DSM to create a single merged surface product, as explained in section 4.2.

### 3.2  Airplane: Ultracam Eagle M3

Airborne imagery was acquired with an Ultracam Eagle M3 by the company Flotron on 11 April 2018 between 11:00 and 12:00. Unfortunately, the data could not be acquired on the same day as the satellite triplet due to technical issues. The meteorological conditions during the data acquisition were partly cloudy and only the northern part of the Dischma valley was cloud free. From 512 images, only 242 images could be used for photogrammetric processing. Fortunately, no noticeable snow fall event occurred between 6 April and 11 April 2018, and the temperature was too low to allow for significant snow melt (maximum 10 cm between 6 and 11 April at the low elevation stations, see graph in Figure 2). The Ultracam Eagle M3 features a CCD image sensor with 450 megapixel (MP) and a pixel size of 4 µm x 4 µm (see Table 1 for more information). The Ultracam Eagle M3 was mounted with the 120 mm focal length lens and flown at mean altitude of 1780 m above ground level (a.g.l.), resulting in a GSD of ca. 6 cm/pixel. The Ultracam images were recorded with a radiometric resolution of 14 bit. The images delivered were 4 bands (RGBI) geotagged images. Furthermore, the data were delivered with camera positions and orientations with a GNSS accuracy of 0.2 m, an Inertial Measurement Unit (IMU) accuracy of 0.01° (omega, phi, kappa) and corrected for lever arm and boresight calibration.

### 3.3  UAS: eBee+ with S.O.D.A. camera

UAS data of the Schürlialp area was collected on 7 April 2018 at 9:27 am for 1.5 h with an eBee+ RTK of SenseFly equipped with the S.O.D.A camera, changing the battery twice. This imaging payload features a 1-inch CMOS sensor with 20 MP (see Table 1 for more information) built specifically for photogrammetric applications. The images were recorded in the JPEG format with a radiometric resolution of 8-bit for each channel. Flying at 182 m a.g.l. on average and with lateral and forward





overlaps of 70% and 60%, respectively, the eBee+ survey captured 1550 images with an average GSD of 0.04 m. A characterizing feature of the eBee+ RTK is the onboard differential GNSS which measured a mean horizontal accuracy of the camera positions around 0.02 m and mean vertical accuracy around 0.03 m. For RTK operation of the eBee+ was referenced directly in the Swiss coordinate system LV95LN02, relative to mount point VRS_GISGEOLV95LN02 of the national GNSS

network.

### 3.4    Ground-based: Canon EOS 750D

The ground-based images were collected with a Canon EOS 750D on 7 April 2018 starting at 10.37 am for 1 hour. The Canon EOS 750D is a digital SLR camera featuring an APS-C CMOS sensor with 24,2 MP resolution. We used a zoom lens (18-55 mm) and set the focal length at 43 mm. The GSD of this ground-based recording changes strongly across the slope, which

affects the photogrammetric results. Also, for a stable and accurate photogrammetric model the ray intersection angle is optimal around 90°-100° and defined by a sufficient B/H ratio (Luhmann et al., 2014). When taking photos from the ground, there are many occlusions. This are all configuration elements that make ground-based recording challenging. Since we took the pictures from the bottom of the Dischma valley, some slopes to the southwest and to the northeast are more than one kilometer away. Thus, the GSD varies between 0.01 m/pixel and 0.1 m/pixel with a mean GSD of 0.05 m/pixel. Towards the north and south

side, we have a flat valley floor for which our ground-based setup is not suitable. Therefore, we use the focal length of 43 mm as a compromise to achieve a mean GSD in the range of the other platforms for the selected camera locations. To ensure stable recording conditions, a tripod was used to take pictures from 5 different vantage points. The tripod was placed at each location and then the camera was rotated on the tripod head. The entire setup was moved in one piece so that the focal length stayed fixed. To have information about the camera location, the differential GNSS (Trimble Geo XH 6000) was placed on the top of

the camera and this position was measured with an accuracy of 0.1 m. Finally, a total of 268 images were recorded on the 7 April 2018 with a radiometric resolution 8-bit in JPEG format. With this ground-based setup we covered the slopes of the northern part of the Schürlialp test site (see Figure 1).





**Table 1: Summary of the photographic data collection with the satellite, airplane, UAS and ground-based platforms.**

| | | Satellite: Pléiades | Airplane: Ultracam Eagle M3 | UAS: eBee+ with SODA camera | Ground-based: Canon EOS 750D |
|---|---|---|---|---|---|
| **Platform information** | **Sensor type** | Pushbroom scanner TMA optics | CCD image sensor | 1-inch CMOS-sensor | APS-C CMOS-Sensor |
| | **Sensor resolution** | Panchomatic array assembly: 5 x 6000 (30,000 cross-track) pixels Multispectral array assembly: 5 x 1500 (7500 in cross-track) pixels | 450 MP | 20 MP | 24.2 MP |
| | **Focal length** | 12.905 m | 122.7 mm | 10.6 mm | 43 mm |
| | **Pixel size** | 13 µm x 13 µm in panchromatic band | 4 x 4 µm | 2.4 x 2.4 µm | 3.7 x 3.7 µm |
| | **Radiometric resolution** | 12-bit | 14-bit | 8-bit | 8-bit |
| | **Image type** | Multispectral TIFF and panchromatic TIFF | High resolution multi-channel RGBI TIFF | sRGB JPEG | sRGB JPEG |
| **Acquisition details** | **Acquisition date** | 07.04.2018 | 11.04.2018 | 07.04.2018 | 07.04.2018 |
| | **Start of Acquisition** | 10.17 am | 11:05 am | 9:27 am | 10.37 am |
| | **Number of pictures** | 3 | 521 (242 cloud free) | 1550 | 268 |
| | **Area covered** | 140 km$^2$ | 75.7 km$^2$ | 3.59 km$^2$ | 1.12km$^2$ |
| | **Mean flight height** | 694 km a.g.l. | 1780 m a.g.l. | 181 m a.g.l. | Mean distance from the Target: 1 km |
| | **Mean GSD** | 0.7 (resampled to 0.5) m/pixel for the panchromatic band (nadir) 2.8 (resampled to 2) m/pixel for the multispectral bands (nadir) | 0.06 m/pixel | 0.04 m/pixel | 0.05 m/pixel |



### 3.5 Reference datasets

#### 3.5.1 Manual snow depth measurements and fixed snow depth poles

Manual snow probing measurements at 27 locations in the Schürlialp test site were performed on 6 April (17 measurements) and 7 April 2018 (10 measurements). Between 6 April and 7 April 2018, the weather was sunny, not too warm and without

precipitation making the manual measurements comparable. The automatic stations around Davos (Figure 1, Figure 2) had lost a maximum of 0.04m between these two days at the lowest elevation station Davos Flüelastrasse (5DF) at an altitude of 1560 m a.s.l.. For each snow probing location, the snow depth was measured plumb vertical with an avalanche probe at each corner and in the middle of a 1 x 1 m square. The position of the square center was recorded with a differential GNSS (Trimble Geo XH 6000). As manual snow probe measurements are only possible in terrain safe from avalanches, 15 fixed snow poles were

installed throughout the Schürlialp area in summer (see Figure 3). The snow depths values were read off the poles with the help of binoculars or zoomed photos. The snow poles had every half meter marked by pointer and at every 10cm red tape subdivided the pole further leading to a measuring accuracy of around 5cm (see Figure 3). At the time of the campaign, the snow depth could be read from 10 snow poles. The other five poles were not visible due to a lack of contrast against the snow or previous avalanches that had bent them.

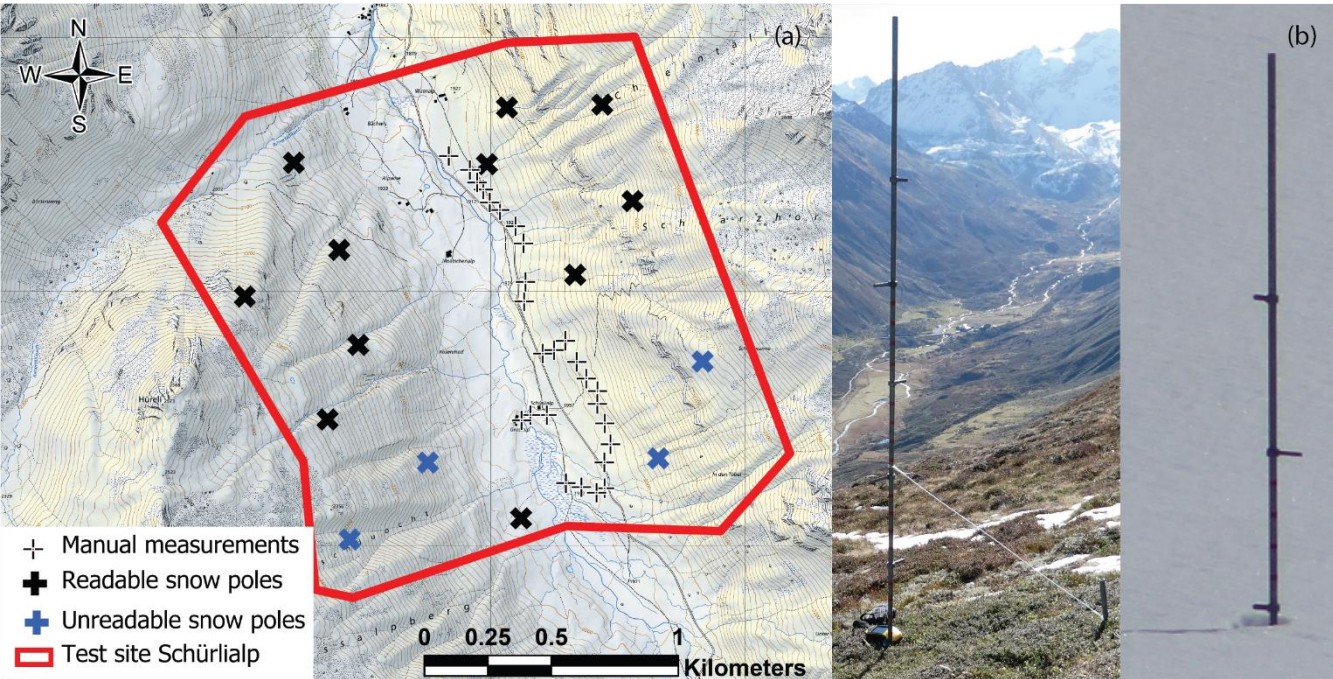

**Figure 3: Panel (a) shows the distribution of the snow poles and the manual snow measurements on the Schürlialp test site. The snow poles are separated into the readable (black crosses) and unreadable ones (blue crosses). The thin black crosses show the locations of the manual measurements. The two images on panel (b) show a snow pole in summer and winter. The snow poles have a hinge at**
**the foot and are tensioned back with a nylon cord. This way they simply fold down in the event of an avalanche and are not dragged along. (Swiss Map Raster© 2019 swisstopo (5 704 000 000), reproduced by permission of swisstopo (JA100118)).**





### 3.6    Summer reference data sets

Mapping snow depth map requires an accurate snow-free reference surface. For this study two summer DSMs were considered. For the snow depth maps on the Schürlialp test site, an eBee+ flight was performed on 27 June 2018 yielding a final DSM with spatial resolution of 0.09 m. For further information about the processing workflow see section 4.4. For producing snow

5    depth maps extending beyond the Schürlialp test site, a DSM with a spatial resolution of 0.5m derived from an airborne laser scan (ALS) of the Dischma valley was used. It was carried out by Milan geoservice GmbH with an ALS of type Riegl LMS-Q 780. Milan geoservice GmbH delivered an oriented, unclassified point cloud in the reference system LV03 LN02. Ground points classification and DSM derivation was done using the software lastools (ground classification: lasground_new with flags –wilderness and –ultra_fine)

**Table 2: Description of the summer data sets used for the calculation of the snow depth maps.**

| | | **UAS: eBee+** | **ALS: LMS-Q 780** |
|---|---|---|---|
| **Acquisition details** | **Acquisition date** | 27.06.2018 | 05.08.2015 – 06.08.2015 |
| | **Number of pictures** | 1449 | - |
| | **Covered area** | 3.66 km$^2$ | 100 km$^2$ |
| | **Mean flight height** | 184 m a.g.l. | 2330 m a.g.l. |
| | **Point density** | 155 points/m$^2$ | 11.8 points/m$^2$ (after postprocessing) |
| | **GSD** | 0.04 m/pixel | - |
| | **Resolution DSM** | 0.09 m/pixel | 0.5 m/pixel |

### 4    Data processing

Before a photogrammetric snow depth map or an orthoimage can be generated, a DSM must first be produced from the images

15    of each platform. We used the software packages Agisoft Metashape version 1.5.3/1.5.5 (for Airplane, UAS, ground-based data) and a combination of ERDAS Imagine 2018, Ames Stereo Pipeline (ASP 2.6.2) and GDAL (for satellite data) for photogrammetric triangulation, restitution of DSM, and production of orthoimages from the different platforms. Once DSMs and orthoimages were created as raster datasets, snow depth maps were calculated using ArcGIS Pro version 2.4.2 by subtracting the summer DSM from the winter DSM. The resulting snow depth maps were validated and compared using 2

20    different comparison strategies in order to evaluate the performance of the individual platforms and workflows. More specific aspects of data processing and performance evaluation are provided in the following sections.





## 4.1 Coordinate systems

Using data in the same horizontal coordinate system with the same vertical datum is fundamental for the calculation of snow depth maps. This requires documentation and verification of the coordinate systems and vertical datums used across the processing workflow for each dataset. For example, the geometry of Pléiades data is defined in terms of WGS84 ellipsoid, both for planimetry and elevation (height above ellipsoid, HAE). Other data such as the summer ALS DSM were delivered in the Swiss coordinate system LV03/LN02. All GNSS data (GCP, eBee+) was recorded in RTK mode based on a swipos-GIS/GEO correction stream using the LN02 height system.

Because the conversion from ellipsoidal heights (WGS84) to LN02 is only achieved by means of interpolation, we define the new swiss height system LHN95 and the local reference system LV95 as the main reference frame for this study. The height system LHN95 (Landeshöhennetz 1995) is derived from geopotential number and provides rigorous orthometric heights with consideration of the Alpine uplift (Schlatter and Marti, 2005). Since the datasets were provided either on LN02 or on WGS84, all conversions from LN02 to LHN95 and WGS84 to LHN95 were handled using the REFRAME library provided by swisstopo. REFRAME was used to create conversion grids to accommodate: (i) WGS84 to Bessel ellipsoidal height separation (deterministic calculation); (ii) Bessel to LHN95 separation (CHGEO2004 geoid model), and (iii) LHN95 to LN02 separation (HTRANS).

## 4.2 Satellite data processing workflow

Processing of Pléiades satellite images involved triangulation in ERDAS Imagine 2018, surface restitution in Ames Stereo Pipeline (ASP 2.6.2, Beyer et al., 2018), DSM post-processing and production of orthoimage with custom scripts in GDAL 2.4.1. Satellite image triplet bundle block triangulation (BBA) is best performed on WGS84 to ensure unambiguous Rational Polynomial Coefficient (RPC) modelling. 14 GCPs from field survey (see section 3) with decimeter accurate coordinates on LV95 and Bessel HAE were converted with REFRAME to UTM32N (ETRS89) and WGS84 HAE. Triangulation was completed on the 50-cm resolution panchromatic images with manual input and manual refinements of the 14 GCPs and 32 Tie Points to achieve robust BBA solution. Final quality assessment of the triangulation was derived from Leave-One-Out Cross-Validation (LOOCV) (Sirguey and Cullen, 2014) whereby each GCP is set as a check point in turns to generate an independent residual, yielding 0.43 m CE90 (Circular Error of 90%) and 0.43 m LE90 (Linear Error of 90%).

Dense stereo-matching at full resolution (50 cm) was completed with ASP using a hybrid global-matching approach (Hirschmuller, 2008; d'Angelo, 2016; Beyer et al., 2018). DSMs were produced from a point cloud at 2 m resolution on UTM32N/WGS84, reprojected to LV95 with GDAL (cubic convolution) and height adjusted to LHN95 using conversion grids mentioned in section 4.1. Maps of ray-intersection errors from stereo-matching with ASP measure the minimal distance





between rays for pairwise stereo, and are indicative of the quality of the match. In tri-stereo configuration, maps of intersection errors are used to weight the contribution of pairwise DSMs into a blended DSM with GDAL (Sirguey and Lewis, 2019). The small B/H ratio of the pair P23 resulted in significantly higher noise that compromised the tri-stereo blending. Alternatively, blending only both DSM members P12 and P13 provided a better surface, with noise comparable or better than the bi-stereo

with the largest B/H ratio (P13). P23 was only used to fill gaps remaining from the two-members blending. The final DSM was used to orthorectify each of the three images, and the three pan-sharpened orthoimages were then blended together to create a single final orthoimage.

Despite the robust survey quality indicated by LOOCV, a remaining 27.5 arcsec tilt (66.7 ppm, or ±1m over 15km) along the

northwest-southeast axis of the imagery was detected in the blended DSM after differencing with the summer ALS DSM. To correct the tilt, points were manually placed along snow-free roads in the imagery, and spot elevations were extracted from the blended DSM and ALS surfaces. A hyperplane was fit through the residuals to create a corrective grid covering the imagery footprint which was used to adjust the blended DSM. Finally, the ray-intersection error map for the blended DSM was used to set all cells of the DSM to no data where the ray intersection error is greater than one panchromatic pixel or 0.5 m as larger

errors were found to be often indicative of erroneous stereo-matching

### 4.3   Airplane data processing workflow

The Ultracam images are distinguished by their high dynamic range of a 14-bit radiometric resolution. There are different software solutions provided to process such images but we decided to use Agisoft Metashape. Agisoft Metashape is used for images acquired with frame sensors of RGB or multispectral type (Westoby et al., 2012; Agisoft, 2019). Agisoft Metashape

supports also the high dynamic range associated with the 14 bits radiometric resolution of the Ultracam images. Since the southern part of the Dischma valley was cloud-covered, the Ultracam images were manually sorted into cloud-free and cloud-covered images. As the Ultracam camera positions were delivered in the height system LN02, the coordinates of the positions had to be converted into LHN95 with REFRAME for input into Agisoft Metashape. The use of the CHGeo2004 geoid model in Agisoft Metashape then allows for consistent processing in the vertical height system LHN95.

After sorting, the images were imported into Agisoft Metashape and aligned. Before alignment, the camera parameters were fixed in Agisoft Metashape to a focal length of 122.7 mm and 0.004 mm x 0.004 mm pixel sizes (see Table 1). The Ultracam is a professional photogrammetric camera that has been accurately calibrated by the vendor so that a refinement of the internal camera parameters by Agisoft are not desirable. To improve the geolocation accuracy after alignment, 29 GCPs distributed

over the Dischma valley were imported into Agisoft. Fifteen CPs were used to control the geolocation accuracy (see section 3 for more information about GCP and CP). The CPs resulted in a RMSE of 0.14 m for the XY coordinates and a RMSE of 0.19 m for the Z coordinates (see Agisoft LLC, 2019 for exact definition of the RMSE). After alignment and refinement of the geolocation accuracy, the dense point cloud was built with the depth filtering method "aggressive". The filtering method



"aggressive" gives, in our experience, the best results for snow-covered surfaces and filters out most outliers, leading to cleaner surface models. The DSM was made from the dense point cloud at a 0.11 m/pixel resolution without interpolating voids. Finally, an orthoimage at a resolution of 0.5 m/pixel was created based on the DSM.

### 4.4    UAS processing workflow

We processed the eBee+ data using integrated sensor orientation (ISO) without GCPs and using only CPs to assess the accuracy of the DSM. SenseFly, the manufacturer of the eBee+, claims that the eBee+ can achieve accuracies in order 0.03 m horizontal and 0.05 m vertical (level of accuracy of 1-3 x GSD) using this method (Benassi et al., 2017; Roze et al., 2017). Benassi et al., 2017 proved a RMSE of 0.02 - 0.03 m for the horizontal coordinates of checkpoints and a RMSE of 0.02 – 0.1 m for the vertical coordinates of CPs for a flight with RTK solution but without GCPs.

The eBee+ has an IMU on board for flight control for which the accuracy and calibration are not given by the manufacturer. Therefore, we have processed the imagery without IMU but with the GNSS data only. Since the mount point applied the corrections for the Swiss coordinate system LV95 LN02 during the flight, the camera positions of the eBee+ had to be transformed into the vertical coordinate system LHN95 before processing could take place. This was done by first exporting

the camera position of the eBee+ images stored in Exif to a text file. This text file served as input for REFRAME to convert the positions into the correct vertical coordinate system LHN95. The transformed positions were then imported back into Agisoft Metashape to overwrite the old camera positions. Again, the use of the CHGeo2004 geoid model in Agisoft Metashape allowed for consistent processing in the vertical height system LHN95. The images were then aligned. A dense point cloud was produced with the filtering mode "aggressive". Finally, the DSM was created with a resolution of 0.09 m/pixel with no

interpolation. An orthoimage was produced with a resolution of 0.04 m/pixel using the DSM. We checked accuracy using six of the signaled CPs on the Schürlialp test site. They resulted in a total RMSE for the XY coordinates of 0.05 m and a total RMSE for the Z coordinates of 0.1 m.

The summer eBee+ flight was processed with the same workflow as the winter eBee+ flight. This resulted in a DSM with a

resolution of 0.09 m/pixel and an orthoimage of 0.04 m/pixel. For the summer eBee+ flight six CPs were signaled. The RMSE of the CPs resulted in a RMSE for the XY position of 0.02 m and a RMSE of 0.05 m for the Z coordinate.

### 4.5    Ground-based processing workflow

Ground-based snow depth mapping is a compromise between measurement requirements and time. Therefore, due to the

avalanche situation and the logistical effort that would have been necessary, no control points could be distributed over the area during the recording. Furthermore, the GCPs/CPs used for the Pléiades, Ultracam and eBee+ are not visible on the ground-based images. Only the camera positions were measured with a dGNSS (see section 3.4) during recording. However, with this



method it was not possible to determine the exact center of the camera. For this reason, the measurement accuracy of the camera position when input into Agisoft Metashape was set to 0.2 m.

To refine the georeferencing of the ground-based images, features such as stones, bushes and house corners emerging from the snow were detected manually on the ground-based images to serve as GCP. The features were then identified on the eBee+ summer flight and their coordinates extracted. Nine GCPs were identified with this approach. For this comparison we did not use CPs, because the model was aligned to the eBee+ summer DSM. To process ground-based images with our acquisition setting, it was necessary to define camera stations for each acquisition location in Agisoft Metashape. The images were therefore sorted into the five camera stations. Using the previously defined GCPs, the images were aligned and the geolocation accuracy was refined. Again, the dense point cloud was created with the filter mode "aggressive". Finally, a DSM with a resolution of 0.11 m/pixel was calculated and an ortho image with a resolution of 0.06 m/pixel was created.

### 4.6    Snow depth map validation and comparison strategies

Three comparison strategies were developed to compare the photogrammetric data and investigate the performance of the different platforms (see Figure 4). Comparison 1 aims to validate the snow depth maps for the Schürlialp test site using the manual and snow pole measurements (described in detail in section 4.6.1). Comparison 2 compares the different snow depth maps with the spatially dense eBee+ snow depth map as ground truth (described in detail in section 4.6.2). The eBee+ summer reference is used for calculating the snow depth maps of comparison 1 and comparison 2. Finally, in comparison 3, to show the potential of measuring snow depth distribution over larger areas, snow depth maps of the Pléiades and the Ultracam are calculated with the ALS summer scan (described in section 4.6.3). Section 4.6.4 describes the accuracy measures used within this paper.





Comparison 1: manual reference

$HS_{platform}$:
- $HS_{Pléiades}$
- $HS_{Ultracam}$
- $HS_{eBee+}$
- $HS_{ground-based}$

compare to

Point ground-truth:
- Manual measurements
- Snow pole measurements

$HS_{platform} = DSM_{platform (winter)} - DSM_{eBee+ (summer)}$

Comparison 2: spatially dense eBee+ reference

$HS_{platform}$:
- $HS_{Pléiades}$
- $HS_{Ultracam}$
- $HS_{ground-based}$

compare to

Spatial ground-truth:
- $HS_{eBee+}$

$HS_{platform} = DSM_{platform (winter)} - DSM_{eBee+ (summer)}$

Comparison 3: snow depth maps for the entire Dischma valley

$HS_{platform\_ALS}$:
- $HS_{Pléiades\_ALS}$

compare to

Spatial ground-truth:
- $HS_{Ultracam\_ALS}$

$HS_{platform\_ALS} = DSM_{platform (winter)} - DSM_{ALS (summer)}$

**Figure 4: Flowchart illustrating the three comparisons strategies.**

### 4.6.1 Comparison 1: manual reference

For comparison 1 only the Schürlialp test site was considered. This allowed us to investigate accurately the performance of the individual platforms by comparing them with the manual measurements and the snow poles measurements (manual ground truth). The snow depth maps are calculated with the eBee+ summer DSM of 27 June 2018. To keep interpolation errors as low as possible, the Winter DSMs were exported at their highest resulting spatial resolution: Pléiades DSM 2 m/pixel, Ultracam



DSM 0.11 m/pixel, eBee+ DSM 0.09 m/pixel and ground-based DSM 0.11 m/pixel. For the calculation of snow depth maps by subtracting grids, it is important that the summer and winter DSMs have not only the same cell size, but also identically aligned grid cells. Therefore, to ensure that the cells were completely congruent, the eBee+ summer DSM was exported in Agisoft Metashape with the same cell size as the winter DSM of each platform, and each winter DSM was then aligned to the

later via resampling with cubic convolution in Arcgis Pro. The summer DSM was then subtracted from the winter DSM resulting in the corresponding platform snow depth map. Finally, to compare the snow depth maps of the different platforms with the manual ground truth, a buffer with a radius of 0.7 m (i.e. the half-diagonal of the 1 m x 1 m sample square) was created from the center position of the manual ground truth. For each snow depth map, the mean value and the standard deviation were calculated within this buffer area. Because the selected buffer has a smaller area than the resolution of the

Pléiades data, the cell value was extracted at the position of the snow depth measurements and the snow poles for this data. In section 4.6.4 further details of the accuracy and precision measures calculated are defined.

### 4.6.2    Comparison 2: spatially dense eBee+ reference

The high accuracy of UAS data for snow depth mapping has been successfully tested in various studies (Vander Jagt et al., 2015; Bühler et al., 2016; De Michele et al., 2016; Harder et al., 2016; Cimoli et al., 2017; Redpath et al., 2018; Avanzi et al.,

2018; Eker et al., 2019). With comparison 2, we compare the spatially continuous snow depth map of the eBee+ with the snow depth maps of the other three platforms. Therefore, the winter and summer DSM of the eBee+ were exported from Agisoft Metashape at a resolution of 0.11m/pixel (for ground-based and Ultracam comparison) and 2 m/pixel (for Pléiades comparison). These DSMs are then aligned to the snow depth maps to be compared via cubic convolution resampling. Finally, the winter DSMs are again subtracted from the summer DSMs resulting in three snow depth maps of the eBee+ used as ground

truth for comparison 2. With these snow depth maps the metrics and plots described in section 4.6.4 are calculated.

### 4.6.3    Comparison 3: Snow depth maps of the entire Dischma valley

The summer ALS scan covers the entire Dischma valley, a much larger area than the eBee+ summer flight. Therefore, we calculated the snow depth maps of the Pléiades and the Ultracam data using the summer ALS scan. We re-exported the Ultracam DSM in a 2 m resolution from Agisoft. With the cubic convolution resampling function, we aligned the Pléiades and

the Ultracam DSM to the ALS DSM and subtracted the summer ALS DSM from each Winter DSM. To compare at a larger scale, we now use the Ultracam snow depth map as ground truth to calculate the accuracy and precision measures described in section 4.6.4.

### 4.6.4    Accuracy and precision measures

For the evaluation of the snow depth maps according to the different comparisons, a selection of accuracy and precision

measures is calculated (see Table 3). Accuracy defines how close an estimated value is to a standard or accepted value of a given quantity (Maune (Ed.) and Naygandhi (Ed.), 2018). Precision (dispersion) on the other side is a measure for how close





measurements agree with each other despite a possible systematic bias (Maune (Ed.) and Naygandhi (Ed.), 2018). The root mean square error (RMSE) is a common measure of accuracy and the standard deviation (STD) a common measure of precision and measures the dispersion of the data in relation to the mean (Maune (Ed.) and Naygandhi (Ed.), 2018). To detect systematic vertical offsets of the snow depth maps, the mean bias error (MBE) and the Median of the bias errors (MABE) were calculated.

The MABE is less sensitive to outliers than the MBE and the difference between the two measures gives an indication whether large outliers are present. A more robust measure to outliers than the STD is the normalized median absolute deviation (NMAD) (Höhle and Höhle, 2009). To illustrate the accuracy measures by graphical means boxplots were calculated for the first two comparison strategies. The boxplot summarizes the statistical measures of the median, quartiles, span and interquartile distance in one graph and allows a graphical interpretation of the data. We decided to also consider a cleaned version of the

errors of each platform comparison whereby errors greater than 2*STD were classified as outliers and removed. There are different methods described in literature to remove outliers, one is to remove 3*STD (Höhle and Höhle, 2009; Novac, 2018). We prefer to be more rigorous and exclude (almost) all outliers, so we used 2*STD. In case of a normal distribution 95.45% of all measured values can be found in the interval of plus-minus 2*STD. With the cleaned data the accuracy and precision measures as well as the boxplot were calculated again. The boxplot of the cleaned data should serve mainly as control and

should have smaller whiskers and boxes than the boxplot of the raw data. Additionally, we calculated scatterplots for comparison 2 and comparison 3 to illustrate the dispersion between the snow depth maps compared to the reference snow depth map (eBee+ snow depth map for comparison 2 and Ultracam snow depth map for comparison 3). We calculate the Pearson correlation coefficient ($R^2$) to evaluate the relationship between the reference snow depth and the model snow depth.

**Table 3: Accuracy and precision measures adapted from Höhle and Höhle, 2009.**

| | |
|---|---|
| $\Delta HS_j$: error of the snow depth at location $j$ <br> $HS_j^{ref}$: reference snow depth at location $j$ <br> $HS_j^{model}$: model snow depth at location $j$ | $\Delta HS_j = HS_j^{model} - HS_j^{ref}$ |
| Root mean square error (RMSE) | $RMSE^{model} = \sqrt{\dfrac{1}{n}\sum_{j=1}^{n}\Delta HS_j{}^2}$ |
| Mean bias error (MBE) | $\hat{\mu}^{model} = \dfrac{1}{n}\sum_{j=1}^{n}\Delta HS_j$ |
| Standard deviation (STD) | $\hat{\sigma}^{model} = \sqrt{\dfrac{1}{(n-1)}\sum_{j=1}^{n}(\Delta HS_j - \hat{\mu})^2}$ |
| Median of the bias errors (MABE, 50% quantile) | $m_{BE}{}^{model} = median(\Delta HS)$ |

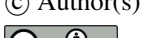



| Normalized Median Absolute Deviation (NMAD) | $NMAD^{model} = 1.4826 * median(\lvert\Delta HS_j - m_{BE}\rvert)$ |
|---|---|
| Threshold for detecting outliers | $\lvert\Delta HS_j\rvert > 2 * \hat{\sigma}^{model}$ |

## 5    Results

The largest area of 140 km$^2$ was covered by the Pléiades satellite, the Ultracam flight covered a surface of 75.7 km$^2$ in the northern part of the Dischma valley, the eBee+ a surface of 3.59 km$^2$ around Schürlialp and finally the ground-based data the smallest surface of 1.12 km$^2$. Figure 5 shows the orthoimages and thus the extent of the area covered by each platform. The
5    red polygon indicates the extent of the Schürlialp test site on each orthoimage. The orthoimage of the Pleiades is completely cloudless in contrast to the orthoimage of the Ultracam which only covers the northern part of the Dischmatal. The orthoimage of the eBee+ illustrates the good quality of the eBee+ flight. Finally, the orthoimage of the ground-based imagery reveals the problem of the suboptimal recording geometry.

10    The following subsections will present the results in detail according to the comparison strategies described in section 4.6. Section 5.1 shows the results of the comparison 1. Section 5.2 continues with the results of comparison 2. Finally, in section 5.3 the snow depth maps of the entire Dischma valley are illustrated.





**Figure 5: Panel (a) orthoimage of the Pléiades data, panel (b) orthoimage of the Ultracam data. The area recorded by the Ultracam is theoretically larger than the area recorded by the satellites, but due to the clouds only part of the images could be used for producing DSM and orthoimage. Panel (c) the orthoimage of the eBee+ data and panel (d) orthoimage of the ground-based data. The red polygon in panel (a), (b), (c) and (d) indicates the Schürlialp site test. The violet stars in panel (d) indicate the five camera positions of the ground-based recordings. (Swiss Map Raster© 2019 swisstopo (5 704 000 000), reproduced by permission of swisstopo (JA100118), Pléiades data© CNES 2018, Distribution Airbus DS).**



## 5.1    Results of comparison 1: manual reference

Using the workflow described in the section 4.6.1, a snow depth map was calculated for each platform. The snow depth maps are shown in Figure 6 and an extract of them in Figure 7. When comparing the maps graphically, all maps show similar snow depth distribution patterns. Also, the histogram in Figure 8 illustrates the similar snow depth distribution between the different

platforms. The fact that the manual and snow pole measurements do not show negative values in the histogram is due to the measurement method. Negative values for manual and snow pole measurements are gross human errors and can be excluded. However, negative values exist on photogrammetric snow depth maps due to vegetation (Figure 7) but also photogrammetric processing. The negative values due to vegetation result from the fact that our summer reference for the calculation of snow depth maps is a DSM. Since some vegetation is pressed down by the snow load in winter, lower or even negative snow depths

result. Bühler et al., 2016 have also identified this effect. Feistl et al., 2014 investigated in field studies on how much vegetation can be pushed down by the snow load. Depending on the vegetation type the difference between the summer height and the height below snow was between 10 and 20cm. They examined only certain vegetation types, namely long grass, short grass and dwarf shrubs. So other alpine vegetation types as green alder (*alnus alnobetula*) are missing. *Alnus alnobetula* is a vegetation type present at different places on the Schürlialp test site. It is pressed down by the snow and negative snow depth

spots are visible on the snow depth maps (Figure 7, violet negative patterns visible on every snow depth map).

The Pleiades data have the largest dispersion (STD = 0.77 m and NMAD = 0.43 m) in comparison to the manual and snow pole measurements. The boxplot (Figure 9) illustrates the dispersion as well. The negative MBE (-0.46 m) and negative MABE (-0.40 m) suggest that the Pléiades snow depth map is systematically lower compared to the manual and fixed snow pole

measurements. This is again depicted by the boxplot where almost 75% the error values are below zero. The difference between the STD and NMAD indicates a large outlier and justifies the removal of all values greater than 2*STD to further characterize errors. Subsequently, the RMSE (0.52 m) and the STD (0.39 m) of the Pleiades data improve significantly. The NMAD has deteriorated by 0.04 m. This comes from the fact, that the Pléiades data had one large outlier ($\Delta H$ = -4.47 m) which influenced the NMAD the other way around. The Pleiades snow depth map however remains negatively biased at the location of manual

and snow pole measurements. It is important to note here the limitation of the triangulation of the Pléiades imagery. Most of the GCPs were collected in summer and only a few could be identified and placed in the images of the Pleiades. Also remains the placement uncertainty of the GCPs due to the resolution of the imagery. The remaining shift was corrected on the basis of points along the roads in three valleys (Sertig valley, Dischma valley, Flüela valley), but without specifically targeting the Schürlialp test site. Thus, the shift does not characterize the technology, but shows the limitation due to triangulation and

absolute orientation approach. Also, it should be kept in mind that the Pléiades images are obtained from space and the resolution of the DSM is 2m.



The Ultracam returns a RMSE of 0.20 m, the eBee+ 0.21 m. The difference is negligible, i.e., the accuracies can be considered equal (note, for the Ultracam only 27 out of 37 measurements could be considered). Also, the STD of the Ultracam and the eBee+ are the same (STD = 0.20 m). The NMAD (0.17 m) of the Ultracam is 0.03 m higher than the one of the eBee+ (0.14 m) but again the difference is small. The boxplot illustrates finally a higher dispersion of the errors of the Ultracam despite the

small differences in the accuracy and precision measures compared to the eBee+. The MBE (0.03 m) of the Ultracam is slightly positive but MABE (-0.03 m) is negative by the same value. Therefore, we consider this difference of 0.06 m negligible and assume that the Ultracam snow depth map is not biased. This result is also supported by the boxplot for both not cleaned and cleaned data. By applying the threshold for outliers RMSE, STD and NMAD are found to be equal at 0.17 m for Ultracam, and slightly better (0.11 to 0.16 m) for eBee+. The MBE (-0.07 m) and MABE (-0.07 m) indicate that the eBee+ snow depth

map has a slight negative bias. The boxplot supports this finding. This is again consistent with Bühler et al., 2016 who found a slight underestimation of snow depth for UAS. Normally it should be the same for the Ultracam but this can due to the difference in processing workflow as the Ultracam was processed with GCPs and the eBee+ without. However, boxplots and accuracy measures of cleaned versions of the snow depth maps, show that the eBee+ has the best accuracy compared to manual ground truth. For the ground-based snow depth map, only four snow pole measurements could be considered and therefore the

statistical statements are not meaningful. For completeness they are nevertheless listed in Table 4.






**Figure 6: Snow depth maps of the Schürlialp test site from the Pléiades (panel (a)), the Ultracam (panel (b)), the eBee+ (panel (c)) and the ground-based data (panel (d)). On the ground-based snow depth map the camera positions are indicated with violet stars. The scale on the snow depth maps ranges from 0 m (red) to 3 m (blue). The red polygon delimits the Schürlialp test site. The black polygon indicates the extract shown in Figure 7. (Swiss Map Raster© 2019 swisstopo (5 704 000 000), reproduced by permission of swisstopo (JA100118))**





**Figure 7: Extract of the Schürlialp snow depth maps shown in Figure 6 with the scale ranging from -3 m to 3 m to illustrate the negative snow depths caused by the vegetation. In this extract the dark violet spots are mainly bushes of the species *alnus alnobetula* pressed down by the snow. (Swiss Map Raster© 2019 swisstopo (5 704 000 000), reproduced by permission of swisstopo (JA100118))**



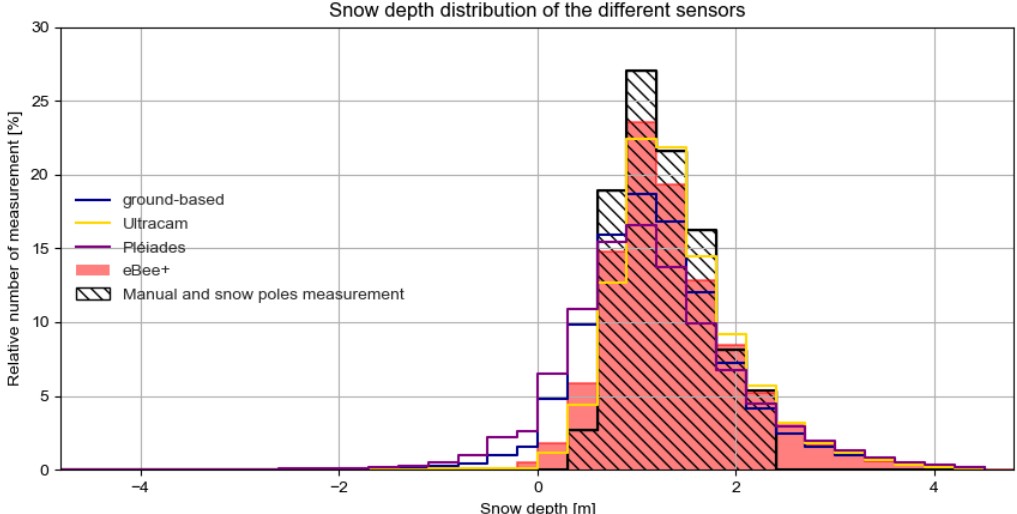

**Figure 8: Histogram of the snow depth distribution of the different platforms over their entire processed extent (Figure 6). The snow depth is shown on the x-axis with a bin size of 0.3 m. The y-axis shows the relative number of measurements. This is the total number of counts per bin divided by the total number of measurements and multiplied by 100. The average snow depth calculated with the Pléiades snow depth map is 1.15 m with a STD of 0.96 m. The average snow depth derived from the Ultracam snow depth map is 1.41 m with a STD of 0.67 m. The eBee+ data shows an average snow depth of 1.35 m measured with a STD of 0.68 m. The average snow depth of the ground-based data is 1.22 m with a STD of 0.80 m. The manual and snow pole measurements resulted in an average snow depth of 1.27 m and a STD of 0.44 m (all not shown).**

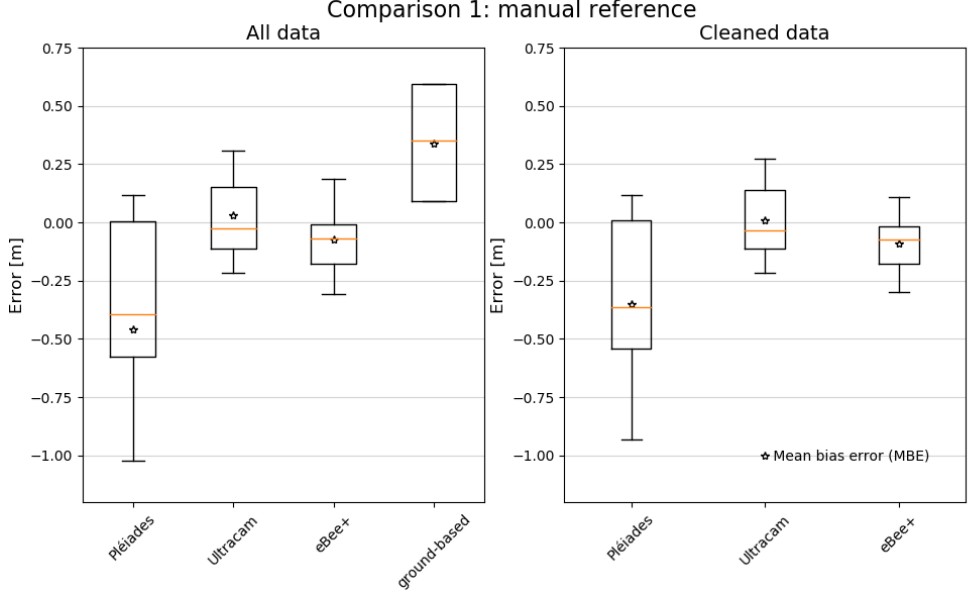

**Figure 9: Boxplot of Comparison 1. The orange line depicts the median, the star the MBE. The 25th and 75th quartile are represented by the boxes and the 5th and 95th quartile by the whiskers. The left boxplot shows the raw data and the right boxplot the data where plus-minus 2*STD is removed. Because of the low sample size of the ground-based applying such a threshold doesn't have a statistical relevance and therefore the ground-based data is not shown in the right boxplot anymore.**





**Table 4: A summary of the accuracy measures of snow depths maps compared to the manual and snow pole measurements for the Pléiades, Ultracam, eBee+ and ground-based imagery. For the Pléiades, Ultracam and the eBee+ the threshold of plus-minus 2\*STD was applied and the newly calculated accuracy measures are depicted in the column clean.**

|  | Pléiades | | Ultracam | | eBee+ | | ground- |
|  | Raw | Clean | Raw | Clean | Raw | Clean | based |
|---|---|---|---|---|---|---|---|
| **RMSE [m]** | 0.90 | 0.52 | 0.20 | 0.17 | 0.21 | 0.16 | 0.54 |
| **MBE [m]** | -0.46 | -0.35 | 0.03 | 0.01 | -0.07 | -0.09 | 0.34 |
| **STD [m]** | 0.77 | 0.39 | 0.20 | 0.17 | 0.20 | 0.13 | 0.42 |
| **MABE [m]** | -0.40 | -0.36 | -0.03 | -0.04 | -0.07 | -0.07 | 0.35 |
| **NMAD [m]** | 0.44 | 0.47 | 0.17 | 0.17 | 0.14 | 0.12 | 0.51 |
| **Number of measurements** | 37 | 36 | 27 | 26 | 37 | 34 | 4 |

## 5.2    Results of comparison 2: spatially dense eBee+ reference

5   The fact that UAS imagery allows for an accurate and spatially distributed calculation of the snow depth even in alpine terrain has been demonstrated in recent publications by (Vander Jagt et al., 2015; Bühler et al., 2016; De Michele et al., 2016; Harder et al., 2016; Cimoli et al., 2017; Redpath et al., 2018; Avanzi et al., 2018; Eker et al., 2019). Comparison 1 has also confirmed that the eBee+ snow depth map with an RMSE of 0.16 m and a NMAD of 0.11 m for the cleaned data is within the expected accuracy for processing with ISO and without GCP. Due to the rather low number of manual and snow pole measurements, which are also mainly located at the valley floor, the data analysis of comparison 1 is limited. Therefore, comparison 2 allow us to more comprehensively analyse the performance of Pléiades, Ultracam and ground-based snow depth maps. The boxplot (Figure 10) shows that all snow depth maps are in a similar range also after cleaning the data by the threshold of 2\*STD.

Again, the Pléiades data show the highest error dispersion compared to the Ultracam and ground-based data. The boxplot of the cleaned data confirm that the Pléiades snow depth map are negatively biased over the Schürlialp test site (MBE = -0.21 m, MABE = -0.19 m.) Even after cleaning, MBE and MABE remain negative with -0.18 m (Table 5). The scatter plot of the Pléiades data in Figure 11 also illustrates the lower snow depths of the HS map Pléiades compared to the HS map eBee+ which explains also the lowest correlation of $R^2 = 0.62$.

20   According to Comparison 1, we assume that the HS map Ultracam must correlate strongly with the HS map eBee+. The MBE and the MABE as well as the scatterplot confirm this assumption with a $R^2$ of 0.94. The RMSE of the cleaned Ultracam snow depth map improved by 0.04 m from 0.17 m to 0.12 m. All other values did not improve at all or by 0.02 m at most when applying the threshold. The right boxplot in Figure 10 also displays the low dispersion of the Ultracam data compared to the eBee+ data, which is further underlined by the low value of around 0.12 m of RMSE, STD and NMAD (cleaned data).





For the ground-based data, the snow depth map correlates also strongly with the the eBee+ snow depth map with a $R^2 = 0.81$ (Figure 11). The MBE of 0.0 m and the MABE of -0.02 m show that the ground-based HS map has no overall shift. This was to be expected since the ground-based snow depth map was georeferenced with the summer eBee+ DSM. The Boxplot (Figure 10) shows a larger dispersion of the measurements than for the Ultracam. The threshold has removed the major outliers and

RMSE (0.21 m), STD (0.21 m), and NMAD (0.19 m) are now very similar.

Finally, the lower scatter plots in Figure 11 assess the correlation of the eBee+ snow depth map with only the snow depths in the range from 0 m to 5 m to the corresponding snow depth of the different platforms. This gives us an indication of how the snow depth maps correlate in the positive and most relevant range for most modelling purposes. 0 m to 5 m was selected based on the snow depth distribution shown in the histogram of Figure 8. The histogram shows the main snow depth distribution in

this range. The $R^2$s of the lower scatterplots show that even in only positive range of eBee+ snow depth map the snow depth maps of the different platforms correlate still strongly and the $R^2$s show hardly any difference.

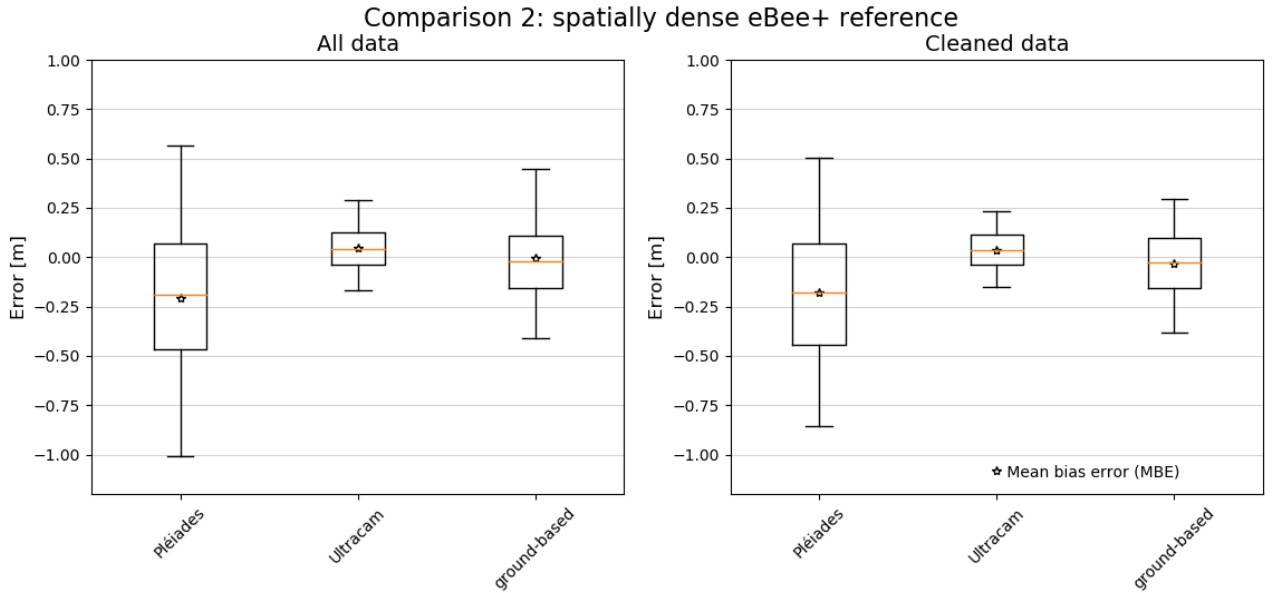

**Figure 10: Boxplot of Comparison 2. The orange line depicts the median, the star the MBE. The 25th and 75th quartile are represented**
**by the boxes and the 5th and 95th quartile by the whiskers. The left boxplot shows the raw data and the right boxplot the data where plus-minus 2*STD is removed from the raw data.**



**Table 5: Accuracy and precision measures calculated for comparison 2: The column raw contains the values for the accuracy measures for all the values and the column cleaned the results of the accuracy measures where plus-minus 2\*STD are removed. For the Pléiades data 4.2 % of the data is removed. For the Ultracam 4.9% of the data is removed. Overall, 4 % of the ground-based data were not considered when thresholded at plus-minus 2\*STD.**

| | Pléiades | | Ultracam | | Ground-based | |
|---|---|---|---|---|---|---|
| | **Raw** | **Cleaned** | **Raw** | **Cleaned** | **Raw** | **Cleaned** |
| **RMSE [m]** | 0.63 | 0.44 | 0.17 | 0.12 | 0.35 | 0.21 |
| **MBE [m]** | -0.21 | -0.18 | 0.05 | 0.04 | 0.0 | -0.03 |
| **STD [m]** | 0.59 | 0.4 | 0.16 | 0.12 | 0.35 | 0.21 |
| **MABE [m]** | -0.19 | -0.18 | 0.04 | 0.04 | -0.02 | -0.03 |
| **NMAD [m]** | 0.4 | 0.38 | 0.12 | 0.11 | 0.19 | 0.19 |
| **Number of measurements** | 854,519 | 818,834 | 174,072,072 | 165,622,459 | 43,554,271 | 41,797,672 |

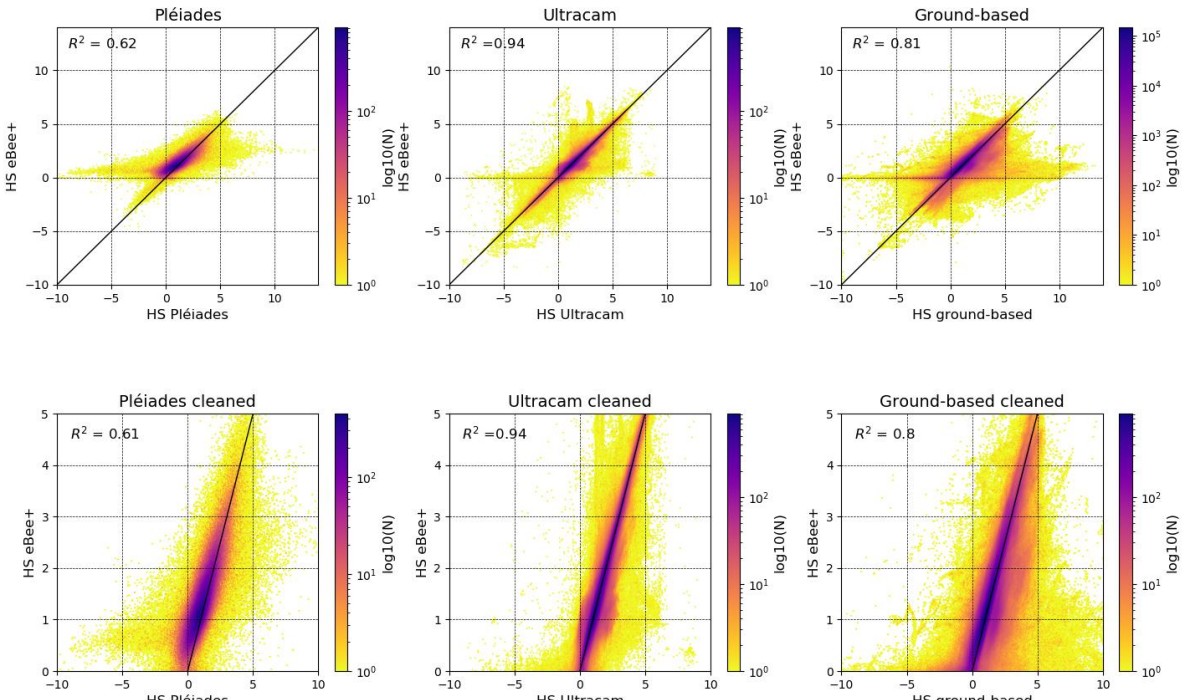

**Figure 11: Scatterplot of Comparison 2: The three upper scatterplots show the platform snow depth on the x-axis and the corresponding eBee+ snow depth on the y-axis. For the lower scatterplot the snow depths of the eBee+ snow depth map in the range of 0 m to 5 m were considered and compared to the corresponding value of the platform to be compared. The black line is the regression line x = y. The scatterplots have a logarithmic scale, which is shown on the right side. R² is calculated for each scatterplot.**





### 5.3     Result of comparison 3: Snow depth maps of the entire Dischma valley

A large-area, high-resolution snow depth map opens up benefits for many applications. Therefore, the snow depth map of the entire Dischma valley is calculated with the summer ALS scan for the Pléieades and Ultracam imagery and is shown in Figure 12. Because of the results of comparison 1 and comparison 2 we use now the Ultracam snow depth map as a ground truth.

5    What can be seen well on both snow depth maps in Figure 11 are the photogrammetric problems of the forest. For forested areas ALS or TLS provide better results as they can penetrate the vegetation layers and record signals from the ground. We have not performed a segmentation into snow and forest for these snow depth maps, so we have a high RMSE (2.2 m) and STD (2.2 m) depicted in Table 6 as well as a large difference between MBE (-0.02 m) and MABE (-0.18 m) when using all data. Again, this underlines the benefit of the strict threshold of 2*STD. After thresholding, the RMSE (0.92 m) and STD (0.9

10   m) diminished by half and indicate an accuracy and precision in the same order.

When looking at the correlation between Pléiades and Ultracam in Figure 14, an $R^2$ of 0.74 indicates a strong correlation. As we are also interested in the correlation in the range 0 m to 5 m of the reference data set, we calculate the scatterplot and the correlation for this range as well. There the correlation is clearly weaker with $R^2 = 0.21$. This again comes on one hand from the bias of the Pléiades snow depth maps which is also shown by the MBE (raw: -0.02 m, cleaned: -0.17) and the MABE (raw:

15   -0.18 m, cleaned: -0.19 m). This bias is also visible graphically in the boxplot (Figure 13) as well as in the scatterplot (Figure 14). On the other hand, we do not know how the Ultracam snow depth map is biased over this entire area.

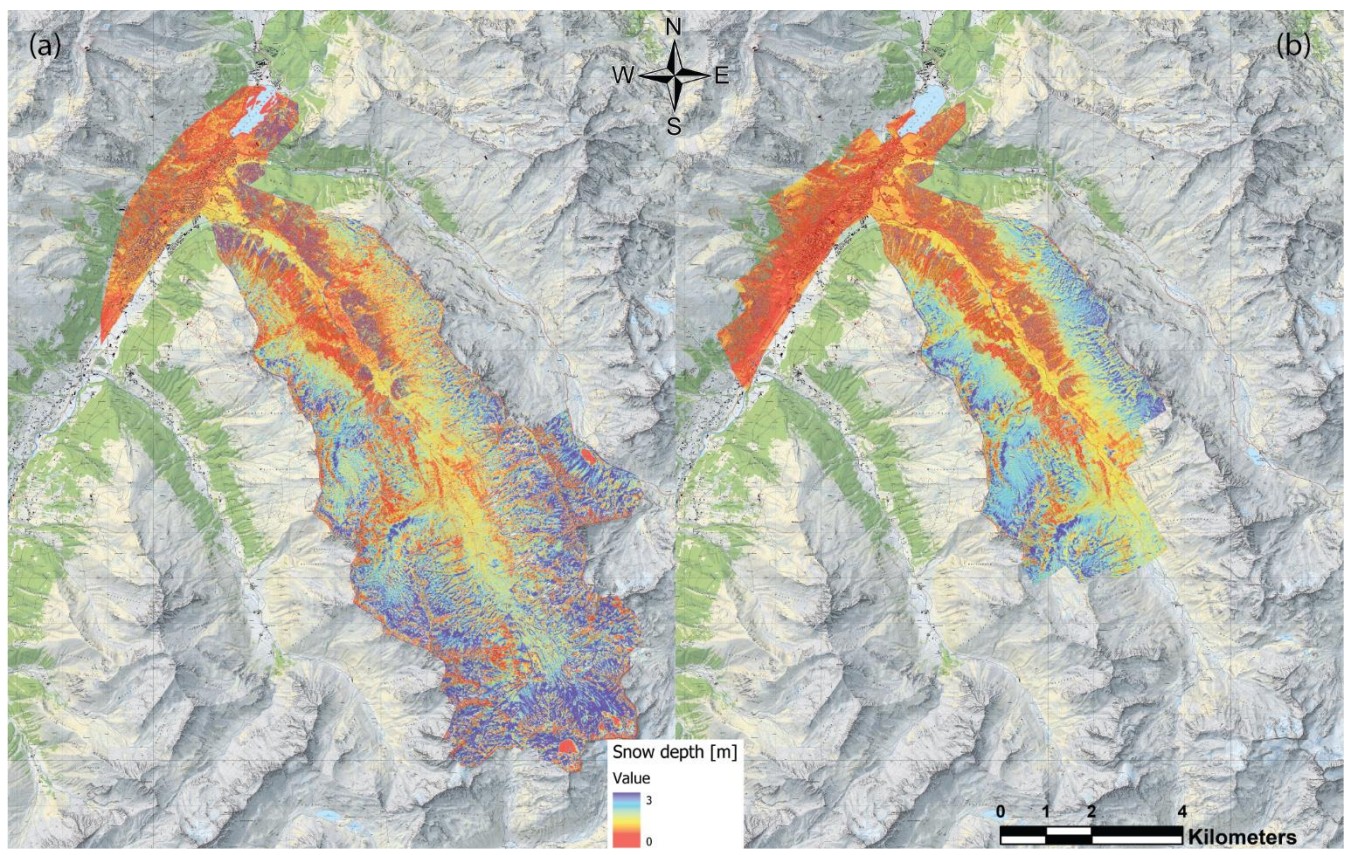

**Figure 12: Snow depth maps of the Pléiades (a) and the Ultracam (b) platforms for the whole Dischma valley. The snow depth is ranging from 0 m (red) to 3 m (blue) (Swiss Map Raster© 2019 swisstopo (5 704 000 000), reproduced by permission of swisstopo (JA100118)).**

5 **Table 6: Accuracy and precision measures calculated for comparison 3: The column "raw" contains the results for all the values and the column "cleaned" the results where plus-minus 2\*STD are removed. 3.5 % of the data is removed by applying the threshold of 2\*STD.**

|  | Raw | Cleaned |
|---|---|---|
| **RMSE [m]** | 2.2 | 0.92 |
| **MBE [m]** | -0.02 | -0.17 |
| **STD [m]** | 2.2 | 0.9 |
| **MABE [m]** | -0.18 | -0.19 |
| **NMAD [m]** | 0.95 | 0.65 |
| **Number of measurements** | 8,637,387 | 8,333,955 |



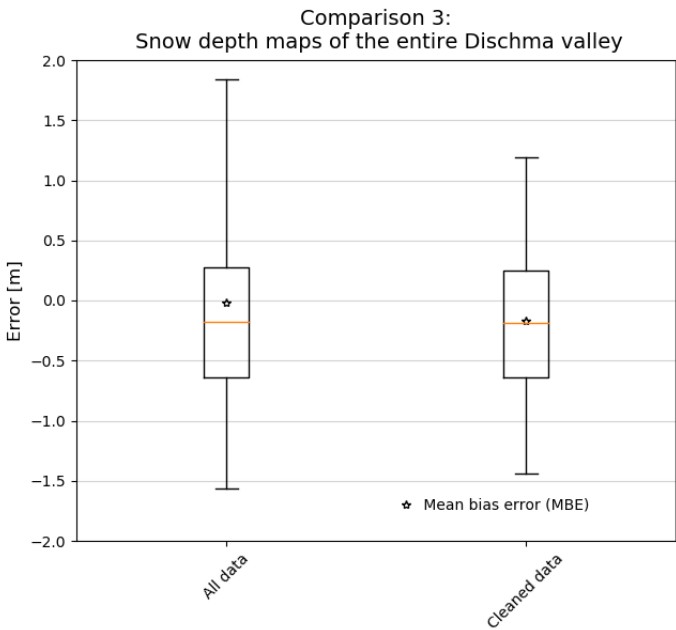

**Figure 13: Boxplot of Comparison 3. The orange line depicts the median, the star the MBE. The 25th and 75th quartile are represented by the boxes and the 5th and 95th quartile by the whiskers. The left boxplot contains all data and the right boxplot the data where plus-minus 2\*STD is removed.**

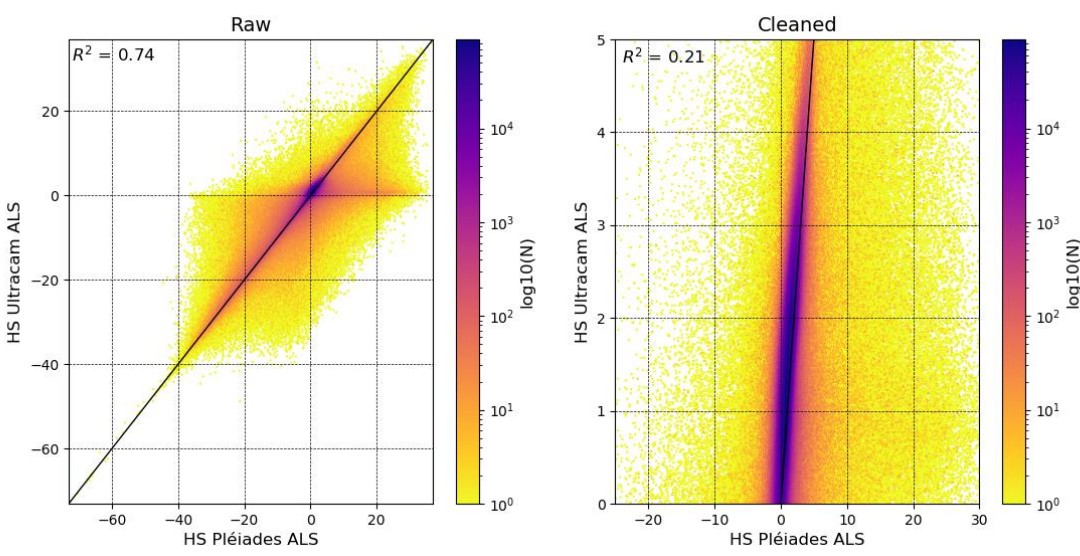

**Figure 14: Scatterplot of Pléiades snow depth map compared to the Ultracam snow depth map. The left scatterplot shows all the data plotted. The right scatterplot shows only the points where the Ultracam snow depth is between 0 and 5 and the corresponding Pléiades snow depths. The black line is the regression line x = y. The scatterplots have a logarithmic scale, which is shown on the right side. The square of the correlation coefficient is calculated for each scatterplot.**





## 6    Discussion

In this study we compare four different photogrammetric platforms focusing on their performance for spatial continuous snow depth mapping in high alpine terrain. Every platform has its distinct advantages and disadvantages (summarized in Table 7 at the end of the discussion). In this section we discuss the results presented in section 5 and describe our experiences using these

platforms. Furthermore, we give recommendations on potential applications.

### 6.1    Satellite photogrammetry: Pléiades

Very high-resolution optical satellites (GSDs of 0.3 to 0.7 m) have the main advantage that they can cover several hundreds of square kilometers from space in a few seconds if there are no clouds present. Today only two studies are published investigating the performance of Pléiades DSMs for snow depth mapping: Marti et al., 2016 achieve a STD of 1.47 m and a

NMAD of 0.78 m in comparison to an UAS dataset; Deschamps-Berger et al., 2020 achieve a RMSE of 0.8 m and a NMAD of 0.69 m in comparison to airborne LiDAR snow depth maps covering an area over 137 km$^2$. In this investigation we achieve a RMSE of 0.44 m (0.63 uncleaned) and a NMAD of 0.38 m (0.4 m uncleaned) in comparison to a UAS snow depth map. This is higher than the previously reported accuracies, but our validation is limited to an area of 3.6 km$^2$ at the Schürlialp test site. Our RMSE is in the range of the GSD of the input satellite imagery (0.5 m) and we therefore assume that we are very close to

the maximum achievable accuracy. Even though we have compared the satellite snow depth maps with the snow depth maps of the Ultracam, we cannot accept this as a complete validation. Although we know that the Ultracam has a high accuracy and precision at the Schürlialp test site, we lack validation for the whole valley and therefore the RMSE (0.92 m) and NMAD (0.65 m) of this comparison are of limited value. Nevertheless, they show that the Pléiades snow depth map is similarly accurate over the whole area as on the Schürlialp test site.

Our findings limit the range of applications for satellite-based snow depth maps. Shallow snow packs, present over large parts of the tundra regions of the northern hemisphere, have mean snow depth values in the range of 0.5 m (Sturm et al., 2008) and can therefore not be captured with sufficient reliability by the Pléiades platform. In mountain regions on the other hand, where larger mean snow depth values are present and the amplitude of the values is generally larger, satellite-based snow depth maps

could provide important information. Typical applications that could profit are the mapping of snow avalanches (Bühler et al. 2019) and the estimation of water resources stored in snow (Jonas et al. 2009) for drinking water supply or hydropower (Farinotti et al. 2012). In many remote regions of the world, where access is difficult and dangerous, satellite photogrammetry is the only feasible method to gather spatial continuous snow depth information. However, to orientate the imagery and to accurately align the snow-free and the snow-covered DSMs, ground control points GCPs, well distributed over the entire area,

are necessary. If this is not possible, the DSMs could be aligned based on a master scene but this is often very difficult in alpine, snow covered terrain because not enough points can be identified in both scenes accurately.





## 6.2 Airplane photogrammetry: Ultracam

Airborne photogrammetry campaigns depend on the availability of suitable airplanes and camera systems as well as on flight permissions. If this is given, entire catchments of several hundreds of km² can be covered within 1 – 2 hours with ground sampling distances of 0.05 to 0.25 m. The investigations of Bühler et al., 2015 and Boesch et al., 2016 with an ADS80/100

optical scanner and Nolan et al., 2015 with a Nikon D800E frame camera reported accuracies of 0.10 m to 0.3 m. In our study we use an Ultracam Eagle M3 frame camera and achieve a RMSE of 0.12 m (0.17 m uncleaned) compared to the UAS snow depth values.

Due to the higher GSD airborne photogrammetry allows for snow depth maps in a higher resolution than the satellite-based

photogrammetry and reaches accuracies close to UAS photogrammetry. This is particularly the case if low flight heights above ground are planned resulting in high GSDs of 0.05 to 0.1 m. In alpine terrain the GSD varies a lot due to topography. Therefore, as the airplane flight lines are usually at fixed elevations, highly elevated areas are closer to the sensor than regions in the valleys, resulting in varying GSDs within the investigation area. Airplanes can also fly below high clouds; however, diffuse illuminations corrupt the contrast of the snow-covered surfaces and may result in insufficient DSM qualities with holes and

outliers. Due to the medium to high accuracy and precision and the large possible coverage, airplane photogrammetry is a very valuable tool for all applications where the investigation area is reachable with an airplane and an accuracy in the range of 0.1 to 0.3 m is acceptable. Also, airborne photogrammetry depends on GCPs or the alignment to a master scene for orientation. Because the GSD is higher than in the satellite data, the accurate identification of points might be easier.

## 6.3 UAS photogrammetry: eBee+

UAS photogrammetry proved to be an accurate, flexible and reliable tool for snow depth mapping achieving accuracies in the range of 0.05 to 0.2 m (Bühler et al., 2016; Harder et al., 2016; De Michele et al., 2016; Redpath et al., 2018). In this study we applied ISO based on onboard RTK without applying ground control points. Our experience with more than 150 flight missions shows, that the snow depth values generated under favorable illumination conditions are of very high quality and are often even more trustworthy than the manual probe measurements. Therefore, we can apply the eBee+ snow depth map as ground

truth for the validation of satellite and airplane photogrammetry.

The main limitation of UAS photogrammetry is that current systems can only cover limited areas in the range of a few km² due to technical limitations and legal regulations. The pilots have to be very close to the investigation site to fly the drone in line-of-sight. Furthermore, starting and in particular landing the drones in high-alpine terrain is challenging in particular if

wind gusts of more than 15 m/s are present. At present higher wind speeds make it impossible to operate any known mapping drone. On the other hand, UAS enable the flexible generation of accurate and precise DSMs. Therefore, they can be applied





for all applications requiring spatial continuous snow depth information as long as a limited area is acceptable. An example for such an application is the measurement of snow redistribution by wind at a specific mountain ridge (Walter et al., 2019).

## 6.4 Ground-based photogrammetry: Canon EOS 750D

Ground-based photogrammetry needs no high-end photogrammetric equipment and only a powerful consumer camera can be
enough to fulfill the task. Ground-based photogrammetry is a suitable tool if only smaller areas mainly in steep terrain facing towards the cameras should be covered. If fixed installed cameras are used, a very high temporal resolution of several measurements per day is possible. The spatial resolution declines with the distance to the camera position and the inclination away from the sensor. Our results demonstrate that flat regions are often not visible, so a spatial continuous coverage can only be achieved in steep terrain facing to the camera. The achieved precision and accuracy in this study was ranging from 0.2 m
to 0.5 m. Potential applications for such platforms are e.g. the mapping of snow avalanche mass balances (e.g. Thibert et al., 2015) or the mapping of snow depth variations at confined locations such as mitigation measures along roads (e.g. Basnet et al., 2016) where large snow depth values are expected.

## 6.5 Influence of vegetation

Our study confirms that vegetation plays a not neglectable role for snow depth mapping (Bühler et al., 2016; Harder et al.,
2016; Redpath et al., 2018). Photogrammetric DSMs over forested areas are limited. However, bushes can also produce holes and negative snow depth values in a snow depth map. For example, all four snow depth maps showed the same negative snow depth patterns due to the species *alnus alnobetula* (Figure 7). These bushes stand tall (up to 2 – 3 m) during summer and are pressed to the ground by the snow cover in winter resulting in negative snow depth values of up to 3 m (Figure 7). Therefore, we suggest airborne laser scanning techniques for investigating snow depth near forests. If bushes are present, filtering a
summer laser scan for producing a digital terrain model (DTM) instead of a DSM can improve the snow depth measurements over vegetated areas.

 

**Table 7: Summary of the RMSEs/NMADs and main advantages/disadvantages of the investigated photogrammetric platforms. Except for the eBee+ the RMSEs/NMADs are the cleaned values from comparison 2. For the eBee+ the RMSE and NMAD are taken from the comparison 1, therefore they are higher than the once of the Ultracam.**

| Platform | RMSE [m] / NMAD [m] | Main advantages | Main disadvantages |
|---|---|---|---|
| **Satellite (Pléiades)** | 0.44 / 0.38 (Comparison 2) | • Very large coverage possible<br>• Fast acquisition times<br>• Covering remote regions<br>• High temporal resolution possible | • Limited spatial resolution which results in limited accuracy and precision<br>• Data acquisition costly<br>• GCPs / alignment necessary |
| **Airplane (Ultracam)** | 0.12 / 0.11 (Comparison 2) | • Large coverage<br>• High accuracy<br>• Suitable for most applications | • Data acquisition costly<br>• GCPs / alignment necessary<br>• Dependent on external availability |
| **UAS (eBee+)** | 0.16 / 0.11 (Comparison 1) | • Very high accuracy<br>• Economic and flexible<br>• No GCPs necessary | • Limited coverage<br>• Wind, especially gusts critical<br>• Starting and landing demanding |
| **Ground-based (Canon EOS 750D)** | 0.21 / 0.19 (Comparison 2) | • Minimal material required<br>• Economic and flexible<br>• High temporal resolution possible | • Limited coverage with holes in flat/obstructed terrain<br>• Limited accuracy<br>• GCPs / alignment necessary |

## 7    Conclusions

5   In this study we tested for the first time a wide range of available photogrammetric platforms in a timely manner covering the same area for their performance in snow depth mapping. Satellite (Pléiades) imagery demonstrated its capability to map large areas with a high temporal resolution but low spatial resolution. A RMSE of 0.44 m and a NMAD of 0.38 m at a snow depth map resolution of 2 m/pixel (cleaned data comparison 2) underline these statements. For higher spatial resolution and large coverage but lower temporal resolution airplane (Ultracam) imagery is suitable. The comparison to the UAS (eBee+) snow

10   depth map demonstrated the high accuracy of the Ultracam snow depth map (resolution 0.11 m/pixel) with a RMSE of 0.12 m and a NMAD of 0.11 m. UAS images are an economical and flexible method for mapping snow depth with high accuracy (RMSE of 0.16 m and NMAD of 0.11 m in comparison 1) and high spatial resolution (0.09 m/pixel snow depth map resolution). But coverage and temporal resolution of an UAS is limited. Finally, one advantage of ground-based (Canon EOS 750D) images

is that they require minimal material and allow high temporal resolution. However, coverage, accuracy and precision are limited and not possible for every application. Nevertheless, the snow depth map of ground-based images achieved an RMSE of 0.21 m and an NMAD of 0.19 m with a snow depth map resolution of 0.11 m/pixel (cleaned data in comparison 2).

5 With these investigations we demonstrate that digital photogrammetry is a powerful tool to spatial continuously map snow depth distribution in alpine terrain. As such information was mostly missing in the past, various important applications such as avalanche warning, ecological investigations in alpine environments or hydropower generation can profit from data acquired by this new technology in the future. With further advancements in sensor technology for all tested platforms, we expect even better accuracies and coverages in the future. We assume that digital photogrammetry could become the preferred method for 10 snow depth mapping. It is more flexible and less costly than laser scanning for most applications, as long as the vegetation cover is negligible.

**Data availability**

The photogrammetric and reference data sets will be published on ENVIDAT (https://www.envidat.ch) upon the final publication of this paper.

15 **Author contribution**

LE, YB and AS designed and performed the experiments. PS, AM and YB processed the satellite imagery. LE processed the Ultracam, UAS and ground-based imagery. LE and YB prepared the manuscript with contributions from all co-authors.

**Competing interests**

The authors declare that they have no conflict of interest.

20 **Acknowledgements**

We would like to thank the Swiss National Science Foundation (SNF) for funding this project under Grant N° 200021_172800 and Grant N° IZSEZ0_185651. P. Sirguey and A. Miller were funded by MBIE Endeavour Smart Idea research project "Quantifying environmental resources through high-resolution, automated, satellite mapping of landscape change" (UOOX1914). Methodological approaches for processing Pléiades data were developed in part with funding from 25 University of Otago Research grant "Glaciers in the picture" (ORG-0118-0319) and GNS research grant "Topographic mapping of Franz Josef glacier "(GNS-DCF00043). We would also like to thank Elisabeth Hafner, Kevin Simmler, Konstantin Nebel, Daniel von Rickenbach, Julian Fisch, Michael Hohl and Christian Simeon for their help with the various fieldwork tasks.



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
