# Peer review of "Intercomparison of photogrammetric platforms for spatially continuous snow depth mapping"

_The Cryosphere, 2020_

## Short Comment (SC1) · 8 Jun 2020

Eberhard et al. present an inter-comparison of methods to measure distributed snow depth maps in alpine terrain. They combined snow-on and snow-off digital terrain models (DTM) calculated with various methods to produce snow depth maps. In particular, they evaluated the accuracy of a snow depth map derived from spaceborne photogrammetry (satellite Pléiades) using snow depth maps measured with ground based photogrammetry (1.12 km$^2$), UAS photogrammetry (3.59 km$^2$) and airplane photogrammetry (75.7 km$^2$). They found that the accuracy of the Pléiades snow depth map depends on the reference snow depth map and on the use of a 2 x STD filter of the snow depth

residuals (STD: Standard Deviation).

In section 6.1, the authors compared the accuracy of a Pléiades snow depth map with previous studies of Marti et al. (2016), Deschamps-Berger et al. (2020). Using the accuracy of their Pléiades maps and the one obtained from an Unmanned Aircraft System (UAS) survey over 3.59 km$^2$ (RMSE*=0.44 m, NMAD**=0.38 m), they concluded that they achieved higher accuracies than above cited works. We think that this comparison should be discussed in the light of two important methodological differences.

First, the reference snow depth map was calculated with different methods. This impacts the accuracy calculation as each method has its own uncertainty and are not available on the same areas. Deschamps-Berger et al. (2020) evaluated their snow depth maps against a reference snow depth derived from airplane laser scanning over 138 km$^2$ (RMSE=0.80, NMAD=0.69). In Eberhard et al. (2020), the reference dataset was obtained using UAS photogrammetry over 3.59 km$^2$ (RMSE*=0.44 m, NMAD**=0.38 m). At this spatial scale, UAS photogrammetry is expected to have a higher accuracy than airplane laser scanning. The accuracy assessment in Deschamps-Berger et al. (2020) actually compares well to the one calculated by Eberhard et al. (2020) with the reference snow depth map measured by airplane photogrammetry over 75.7 km$^2$ (RMSE = 0.92, NMAD = 0.65 m). Eberhard et al. (2020) puts more weight on the validation using UAS photogrammetry because of the lack of validation for the airplane survey for the whole study site. However this airborne method was well evaluated in previous studies (Nolan et al., 2015, Bühler et al., 2015).

Second, in Eberhard et al. (2020), only the snow-on DTM was calculated with Pléiades images. The snow-off DTM is always common to the evaluated and the reference snow depth map. It is calculated with the same method as the reference snow-on DTM (ground-based, UAS, airplane photogrammetry) which are expected to have higher accuracy than Pleiades-derived DTM. In Marti et al. (2016) and Deschamps-Berger et al. (2020), both the snow-on and snow-off DTM were calculated with Pléiades images. This is a major difference as the accuracy of a snow depth map results from the

combination of the accuracy of the snow-on and the snow-off DTM.

César Deschamps-Berger, CESBIO, Toulouse, France

Simon Gascoin, CESBIO, Toulouse, France

Etienne Berthier, LEGOS, Toulouse, France

*RMSE: Root-Mean-Square Error **NMAD: Normalized Median Absolute Deviation

Bühler, Y., Marty, M., Egli, L., Veitinger, J., Jonas, T., Thee, P. and Ginzler, C.: Snow depth mapping in high-alpine catchments using digital photogrammetry, The Cryosphere 9(1), 229–243, doi:10.5194/tc-9-229-2015, 2015.

Deschamps-Berger, C., Gascoin, S., Berthier, E., Deems, J., Gutmann, E., Dehecq, A., Shean, D., and Dumont, M.: Snow depth mapping from stereo satellite imagery in mountainous terrain: evaluation using airborne lidar data, The Cryosphere Discuss., https://doi.org/10.5194/tc-2020-15, in review, 2020.

Marti, R., Gascoin, S., Berthier, E., De Pinel, M., Houet, T. and Laffly, D.: Mapping snow depth in open alpine terrain from stereo satellite imagery, The Cryosphere 10(4), 1361–1380, doi:10.5194/tc-10-1361-2016, 2016.

Nolan, M., Larsen, C. and Sturm, M.:Mapping snow depth from manned aircraft on landscape scales at centimeter resolution using structure-from-motion photogrammetry, The Cryosphere, 1445–1463, doi:10.5194/tc-9-1445-2015, 2015.

---

## Referee Comment (RC1) · David Shean (Referee) · 30 Jul 2020

Review of "Intercomparison of photogrammetric platforms for spatially continuous snow depth mapping" by Eberhard et al.
https://tc.copernicus.org/preprints/tc-2020-93/#discussion

David Shean
July 29, 2020

This paper considers several photogrammetric approaches that can be used to produce gridded surface elevation products (satellite stereo, aerial, UAS and terrestrial), which can in turn be differenced to provide snow depth estimates. These methods and products are evaluated over one test site in the Swiss Alps, using photogrammetric data collected within a short time period, contemporaneous ground-truth observations and several external datasets. The authors provide some analysis of their snow depth map quality, and qualitative comparisons of the strengths and weaknesses of the different methods.

In general, this is a nice methods paper. The amount of work presented is substantial. The campaign planning was excellent, the photogrammetry processing and methods are sound, the resulting datasets are impressive, and the writing is generally good. There are many DSM processing challenges involved with precise snow depth mapping, especially when surfaces in the "snow-on" DSM are almost entirely snow-covered. The authors use more traditional, labor-intensive approaches to integrate control data early in the process, rather than alternative approaches involving point-cloud or DSM co-registration later in the process. Some decisions about data processing and presentation should be revisited, as they will impact the quantitative results, though they may not have a substantial impact on the qualitative conclusions. The writing in the results and discussion requires some improvement and I recommend that the more senior authors on the paper dedicate time to help refine these sections.

As a fellow photogrammetry enthusiast/evangelist, I enjoyed reading this paper, and had a lot to say, offering many general and specific comments. I realize that not all of these changes will be possible, but I hope that the authors will consider these suggestions to improve this paper and their future work. I look forward to reading the response and seeing this nice study published!

General Comments
- There is ambiguity with the term "ground-based", as many readers may interpret this to be "ground-truth" (I made this mistake a few times). I would suggest using "terrestrial photogrammetry" or "close-range photogrammetry" throughout the text to avoid issues.
- I would offer that snow depth on glaciers and ice sheets is critical for properly measuring glacier mass balance, beyond the applications you list
- The summer reference data sets were acquired on June 27, 2018 and August 5-6, 2018. Were all surfaces completely snow-free during both periods? For some mountain sites with similar elevation here in Pacific Northwest, we see considerable snow in late June. Do you expect differences in the vegetation during these periods? How do you account for errors introduced by these issues?

- I wasn't clear on the strategy used for blending of the independent Pleiades DSMs to create a final triplet DSM product. If the bundle block adjustment was successful, and both P12 and P13 are usable, shouldn't P23 also be usable, even with more limited convergence angle? The text cites Sirguey and Lewis (2019), which is listed as "Sirguey, P., and Lewis, C.: Topographic mapping of Franz Josef glacier region with Pléiades satellite imagery, University of Otago, 2019." - I can't find a published version of this reference, which I think is a technical paper? Need to describe the method in this text if it is not yet published elsewhere. The last sentence of this section mentions a filter to remove triangulation error >0.5 m on the blended DSM. Why not apply this filter to the individual DSMs before blending?
- The +/-1 m (total of 2 m magnitude) over 15 km "tilt" of blended Pleiades DSM is a concern, and seems to indicate that the bundle adjustment was unsuccessful, or the GCPs used are inaccurate or poorly distributed. It's not clear why a more rigorous co-registration approach was not used here rather than a hyperplane fit (presumably least-squares? Need to provide more details on this correction) through residual values at sparse points along roads. I would think a 3D rotation would be more suitable. How does the reader know that this grid correction did not introduce additional error over snow-covered surfaces at greater distances from the points on the roads in valley bottoms?
- Figures 5 and 12 show a clipped version of the Pleaides orthomosaic and snow depth products, with data only shown for the polygon defining the Dischma valley study area. It seems like it might be valuable to show the full unclipped extent of these products, maybe in a supplementary figure. The unclipped products should cover a larger area, adjacent valleys/peaks, and hopefully several of the AWS stations identified in Figure 1.
- The 0.1 m accuracy (is this both horizontal and vertical? Why don't we see expected ~2x higher error in vertical?) for GCP and CP is relatively high for modern GNSS systems, and this will propagate to all of the photogrammetric solutions. Does the 0.1 m value represent RMSE? How was the differential GNSS processing performed? I don't remember seeing any mention of this, so perhaps a real-time correction was used. In the future, I recommend using a survey-grade GNSS for your positions, or longer occupation times with static occupations at each if RTK accuracy is poor.
- For validation, several different reference datasets were used. In the snow depth error analysis, it's unclear whether you can isolate the error from the HS_platform from the error in the "spatial ground truth". Do you assume that HS_ALS(Summer) has 0 error? The lidar vendor documentation/metadata should include some assessment of product error (likely ~1-10 cm). This needs to be accounted for. I think what we really care about here is the accuracy for each HS_platform and whether we can, for example, combine two Pleiades DSMs to independently measure snow depth. If you can isolate platform-specific error (not integrating an external reference DSM), that would be really valuable, though I realize this is challenging.
- Using a 2*STD filter is not necessarily more rigorous. More aggressive, yes. With this filter, you're inevitably classifying many "valid" points as outliers and removing. This filter will always reduce your apparent error, and some might perceive this "cleaning" followed

by lower reported error as a bit dishonest. Since you're also defining robust metrics of error - median for bias, NMAD for spread, and using quantile/percentile values (25, 50, 75%) for the boxplots, you should be able to rely on those, rather than these "cleaned" values. I would avoid calling them "cleaned" - you can use "filtered". There is value in presenting and working with "filtered" maps in figures, but try to use the robust descriptive stats in the text/tables.

- Using Agisoft's "aggressive" filtering will remove many of the largest outliers. ASP also uses an outlier-removal filter during gridding of triangulated points to produce a DSM (point2dem utility removes points with triangulation error > 3*75th-percentile). If you had used different filter settings during DSM generation, your 2*STD filter would produce different results.
- Do your errors actually have a normal distribution? Looking at box plots in figure 9, I would say no. So some of these assumptions about 2*STD and percentiles may not hold. Showing histograms (with some transparency) of delta_HS from the different sources on a single plot might be more informative than the box plots.
- I don't think the variable "HS" was ever defined in the text or captions, though it is used in several figures and tables.
- The results section was a bit hard to follow at times - dense presentation of numbers and statistics. The writing could use some additional proofreading as well - hopefully one of the more senior authors can help with this. There was a lot of interpretation and discussion presented in the results section. Some of the more speculative interpretations should probably be moved to the discussion section.
- Rather than computing a snow depth for 4 DSM inputs against the same summer reference DEM (with it's own spatially variable error), and then doing an accuracy analysis of the snow depth rasters, you might perform your analysis directly on the DSMs. If you trust the DSM_eBee(winter), that can be your reference. Differencing each input DSM against this reference will allow you to characterize the error of the input DSM, without any additional error introduced by the summer reference or the resampling process. Alternatively, you could compute a per-pixel median grid from the input DSM values, and then difference each input against that median, though this works better with n >> 4
- I'm not used to seeing the "mean bias error" (MBE) and "median of the bias errors" (MABE). You are calculating these as the mean and median of the signed (not absolute) error values, right? Both will capture any bias in the data. Also, should be definition of MABE be "median absolute bias error"? The equation in Table 3 does not indicate that absolute values were used. You're missing a "model" superscript on m_BE in the NMAD definition in Table 3.
- I'm not sure that the "comparison 1" is representative of true accuracy of the snow depth products, as the sample size of the reference manual/pole measurements is pretty limited with n=4-37, and the spatial distribution of the "ground-truth" is not ideal, with manual samples clustered along the valley bottom, where gridded snow depth values show stronger spatial autocorrelation. The ~11 usable pole samples on valley walls have good spatial distribution. I expect that if you isolated the two "ground-truth" sources

(manual vs pole), the resulting accuracy statistics would differ considerably. My guess is the apparent error for manual points will be small, but you'll see much larger errors for the ~11 pole samples.

- If you have maps of vegetation (perhaps from differencing a lidar DSM and DTM, though I think your lidar was unclassified?), you could produce a vegetation mask. This could be used to analyze your HS product accuracy only over pixels that are snow, which will allow you to test whether most of your negative snow depth values are from vegetation (primary hypothesis presented), or due to some other issue (bias, noise in DSM, etc). This is important b/c your HS maps don't cover the same extent, so some are preferentially sampling vegetated areas, and may appear to have more error as a result.
- Figure 7 shows some of the detailed snow depth products, which is really valuable. It shows the noise in the Pleaides HS product compared to the ultracam and eBee HS products. But the focus is on artifacts in all HS maps due to vegetation. It is clear that the source for these artifacts is the summer ALS DSM product that was subtracted from each of the 4 photogrammetry DSMs provided. It might be valuable to show a panel with a color shaded relief map of your summer ALS DSM and a snow-free orthoimage showing this vegetation. Could also show similar panels for a snow-on DSM and orthoimage. This is tied to the suggestion of providing an evaluation of the DSM products, rather than the derivative HS products.
- Can you add some discussion of the maximum snow depths that you're seeing in the gullies? The scales are cut off at 3 m, and I expect that you could have real snow depth of >5 m locally. Your products capture this detail really well, which is a major selling point!
- It's important to remember (and state) that snow depth maps are just a specific application of a DEM difference map. There are many studies out there offering accuracy analysis of the latter, and it's important to consider these. Photogrammetric DEMs can also suffer from elevation- and/or slope-dependent errors (e.g., Shean et al., 2016). It may be worth considering how this affects your results.
- There are several figures/tables that could be moved to a supplement, which would help reduce length and improve flow
- In your Table 7, you present some qualitative advantages and disadvantages. You might consider breaking out into columns for coverage, acquisition time, cost, accuracy, resolution, repeat, and other. One thing that is missing from this is the "processing time," in addition to acquisition time. It might be useful to mention somewhere the time required to process each dataset (manual interaction and compute), software costs, equipment costs, etc. At the end of the day, these are often the deciding factors...

Specific Comments
Rather than "RMSEs" and "NMADs", I recommend "RMSE and NMAD values are…"
25: "too few"

25: Based on abstract, unclear why ground-based obs were used with eBee but not with snow pits; I think the issue is that the ground-based obs don't intersect with "manual and snow pole measurements"
29: specify "more than two photogrammetry platforms"
30: replace "the specific advantages and disadvantages of them" with "their specific advantages and disadvantages"

18: Specify the sensors or methods used for "manual" or "AWS" - e.g., probing, GPR, sonic ranging, etc.; Also, point measurements themselves are not the problem - the issue is their sparse spatial density, especially over large regions. One can probe every 10 cm on a relatively small grid and get lots of valuable information about the local spatial distribution of snow depth.
27: Should include Painter et al reference for ASO: https://doi.org/10.1016/j.rse.2016.06.018
31: What kind of accuracy? Is this position accuracy, and 0.1 in horizontal and vertical? Or snow depth? Several important factors for TLS, so careful with a blanket statement like this. See also study by Currier et al (2019): https://dx.doi.org/10.1029/2018WR024533

8: Would be valuable to mention other VHR satellite image options for snow depth mapping. See our preliminary publication on WorldView stereo snow depth: "Spatially extensive ground-penetrating radar snow depth observations during NASA's 2017 SnowEx campaign: Comparison with In situ, airborne, and satellite observations
D McGrath, R Webb, D Shean, R Bonnell, HP Marshall… - Water Resources Research, 2019"
9: Should include medium-format (Phase One) camera SfM mapping: "Assessing the Ability of Structure From Motion to Map High-Resolution Snow Surface Elevations in Complex Terrain: A Case Study From Senator Beck Basin, CO
J Meyer, SMK Skiles - Water Resources Research, 2019"
Chris Larson's work in Alaska?
23-24: Consider rewording to emphasize that few studies have evaluated these platforms for the specific application of snow depth mapping. Many past studies have done this for other surface types and scientific/mapping applications.
26: Mention high-albedo surfaces (sensor saturation), limited surface texture for fresh snow (poor stereo correlation results)
28: UAS can offer cm-scale products; all depends on altitude
31: Other issues beyond ownership can prevent flying whenever required - regulations, certified pilot availability, weather
35: "scales"

19: "surface slope"
20: "contrasted snow depth distribution" is a bit unclear - I think you're saying that downslope winds from gullies lead to snow deposition at the base of the gullies?
24: "Data… provide" (singular, not plural)

24: not just "snowfall" though, as you have wind redistribution and melt, I think "snow depth evolution" might be better term

25: "it" - I think you mean the snow stations?

28-29: I agree that things didn't change much, but if you consider the observed change as a percentage, then 20 cm of a ~65-70 cm snowpack is pretty substantial! You might consider framing these observed changes as a percentage of your expected measurement error. For example, 20 cm is well within the Pleiades snow depth accuracy, but much larger than the sUAS snow depth accuracy.

10: eBee+ RTK is a "fixed-wing" - should mention this, as many readers may think "drone = quadcopter"

13: "triangulation" is not the word I would use here, as it could be confused with stereo triangulation to produce a point cloud. I think you mean consistent geolocation.

17: OK, so these are visible in the Ultracam and eBee imagery, but probably not visible in Pleiades?

19: "positioned" to "The positions of all GCPs were determined using…"

2: "Pleaides-1B stereo image triplet"

7-9: I would also recommend providing the combined off-nadir angles for each image, rather than the signed across and along-track components. I think your B/H numbers are just accounting for along-track off-nadir angles? If possible, good to provide convergence angle for each pair in addition to B/H.

10: Given your scene relief, this is probably acceptable

15: Provide time zone for times (and all other instances in paper)

16: Could delete sentence about not acquiring on same day, as we don't know what the technical issues were

20 and Table 1: would be nice to provide sensor dimensions in addition to total pixel count

21: "Large-format CCD"

22: focal length in Table 1 is 122.7.

24: "I" is near-infrared? Recommend consistency with terminology used for Pleiades section

28-29: OK, so three flights total? Better to state that, rather than changing the battery twice.

Page 8:

7: time format 10.37 vs. 10:37, and time zone

7: What is imaging interval? 1 second, 1 minute? Ah, it wasn't clear that this was a terrestrial photogrammetry survey! I thought you had installed time-lapse stereo cameras. Maybe state this earlier in the section, before you get into details on variable GSD. Could move lines 16-22 or at least 16-19 here.

11: That sounds really large for a convergence angle, and earlier you said B/H of 0.25-0.42 is acceptable? Why different here?

19: Really cool to see this setup! :) I used a similar rig for terrestrial and oblique aerial surveys in Greenland and Pacific NW: https://dshean.github.io/technology/sfm/ (Fig 4), back before consumer UAS and integrated cameras were up to the task

12: Hmm. I think you can do better than 5 cm on the pole shown in fig 3b. Probably down to 1-2 cm precision. Is the inaccurate reading issue also related to the depression in the snow surface immediately around the pole? If so, state this. For future work, you might consider a small quadcopter that you can fly to "inspect" the poles from closer range to make more precise readings.

4-6: provide date range for ALS survey. I see August 5-6, 2015?
9: Provide citation for LAStools. Martin is pretty clear about this, especially if you're using an unlicensed version :)
16: "NASA Ames Stereo Pipeline (ASP, Shean et al., 2016, Beyer et al., 2018) version 2.6.2 (corresponding Zenodo DOI)" - please follow latest citation instructions https://github.com/NeoGeographyToolkit/StereoPipeline#citation. Please include Shean et al. 2016, as you are using core ASP functionality for processing Earth observation imagery that was implemented during that effort.
16: not clear what you mean by "GDAL (for satellite data)" - for orthorectification? Resampling?
17: For reference, you can also use the ASP geodiff command or generic gdal_calc.py for simple raster operations outside of ArcGIS.
17: No mention of co-registration before subtraction here? I assume this will be discussed elsewhere…

Page 12:
2: Suggest "analyzing" rather than "using"
9-16: I didn't completely follow all of this, but it sounds like something you clearly thought about carefully :). Any coordinate system transformation of raster data will require some resampling/interpolation, it's a question of error introduced by the transformation. If you can provide some sense of this error, that would be valuable. It's likely several cm.
I initially thought the LV03/LN02 were from 2003/2002, but just checked and it was 1903/1902! Good call to convert to LHN95. I would think the lidar vendor should update their products…
20: Already mentioned version number and citation earlier, don't need to repeat here.
21: Should state you're using the RPC camera model first, then talk about updating during bundle adjustment
22/30: Should avoid starting sentence with "14" and acronyms "DSMs"
27: Is it expected that the CE90 and LE90 are identical? I would expect a factor of ~2 degradation in LE90.

Page 13:

2: See general comment about the citation here and question about blending. I wouldn't say "with GDAL". This is not a standard GDAL command line utility or API function. I think you created a custom script that would weight values from your 3 input DSMs based on the ASP triangulation error map? I recommend you consider the ASP dem_mosaic weighted average blending approach.

9-15: This "tilt" is a bit troubling, and potentially indicates issues with the GCPs, either their spatial distribution and/or recorded position errors. See general comment.

13-15: The last sentence should be moved to the end of the above paragraph (line 7).

19: Note that Agisoft can process grayscale images, and RPC models these days

29: OK, so refining interior camera parameters is not desirable, so you explicitly disabled from the initial alignment solution? Did you add calibrated lens distortion coefficients, or did you allow Agisoft to solve for these? If so, which coefficients? Provide more detail here.

29: Careful about interchanging GCP and CP here. I don't think you have 29 GCPs, and if you're using points to control geolocation, they are GCPs, not check points.

2: I'm really glad that you used a factor of 2x here, and did not export an oversampled DSM at the native image GSD! This is a common issue.

5: I think this was done using the SenseFly eMotion software? This first paragraph is a bit confusing, as you're already talking about the accuracy of the DSM but you haven't told us how the DSM was produced. I recommend you move this paragraph later, and integrate with the last sentence of paragraph 2 (lines 21-22), where you actually report the observed RMSE.

If your base station position is accurate (I think you used NTRIP caster for real-time corrections, not a local base), then you should be able to achieve the accuracy you report without GCPs.

15: "EXIF metadata"

32: dGNSS - check for consistency in terminology used elsehwere

1-2: I think you mean it was not possible to determine the precise offset of the GNSS antenna phase center relative to the center of the camera detector. Could probably estimate vertical offset and constrain a bit better than 0.2 m horizontal and vertical, but not a huge issue.

3-5: What you did here should work. Technically, you want to use a consistent grid for all output products, with same grid cell size, projection and origin. If the spatial extent of the two rasters varies significantly, then you can use something like GDAL's -tap option to force output grid extent to use whole integer multiples of the output grid cell size. Can also do this manually in Agisoft when specifying output extent - round left, top, right and bottom bounds to nearest multiple of the cell size. A shared raster origin (upper left coordinates) or this -tap approach avoids the need for a second resampling step, which is going to further degrade resolving power of your products and can introduce additional error.

7-9: I like this approach. Approximately how many pixels were sampled within this circle?

10: So this sounds like nearest neighbor sampling for Pleiades snow depth grid. I recommend bilinear or cubic for point sampling like this, esp if you see relatively large pixel-to-pixel variability in snow depth grid values near the sites.

Page 18:
12: See general comment about the 2*STD filter
12-13: Do your errors actually have a normal distribution? Looking at box plots in figure 9, I would say no. So some of these assumptions may not hold. Overlapping histograms of delta_HS from the 3 sources might be more informative than the box plot.

Page 19:
3-5: recommend using "area" for all instances instead of "surface"
6: already said ultracam only covers northern part of valley due to clouds elsewhere (which is really too bad, data quality over cloud-free areas looks great). Does "Dischma valley" = "Dischmatal"?
6-7: what does "good quality of eBee+ flight" mean, and how does an orthoimage illustrate this?

Page 21:
3: "graphically" sounds strange to me, I think you mean "qualitatively"
7-8: "errors introduced during photogrammetric processing" - the processing itself is not responsible for the negative values
9: OK, yes, using a DSM is a problem, but the acquisition date of your summer reference DSM will also be very important in terms of the vegetation growth cycle, leaf-out, etc. Also, you are assuming that your photogrametrically derived DSM is capturing the true top of canopy, which is not necessarily the case (esp with "aggressive" filtering that will remove isolated shrubs/trees)
9 and 14: recommend "depressed" or "compressed" rather than "pressed down"
17-22: What is the total sample size for manual and snow pole measurements again? I would argue that some of your larger apparent error is due to limited sample size, and likely the nearest neighbor sampling approach for the coarser Pleaides snow depth grid.
21-23: Careful with this kind of statement, as it sounds like subjective error reporting…
23: "decreased" instead of "deteriorated"
23: The NMAD and median should not be strongly influenced to one outlier, unless your sample size is an issue.
25: Again, triangulation here is a bit ambiguous, as it could be confused with the triangulation to produce the point cloud. I think you're talking about the integration of GCPs during the bundle adjustment routine.
27-29: This information (about the roads used to correct Pleaides DSM) should be moved to the methods section.
29: I think you're trying to say that one sample Pleaides triplet is not necessarily representative of the capabilities of the sensor.
30: "...approach for snow-covered images" - this problem will be mitigated with snow-free conditions
30-31: Not sure this last sentence is necessary

Page 26:
10: recommend changing "data analysis" to "accuracy analysis of comparison 1 is potentially not representative of the true accuracy of the snow depth products" or something along those lines. See general comment on this issue.

Page 27:
1: Note that this is correlation of per-pixel values, and should not be confused with spatial correlation
6-7: Might be useful to report the percentage of data removed with your 0-5 m filter here. BAsed on Figure 8, this should be minimal for ultracam, but you're removing a nontrivial sample of the Pleiades HS values.
8: Don't start sentence with "0 m…"
11: "R^2 values"

Page 29:
3: "imagery" should be "DSMs"
4: You can only use the ultracam HS as ground truth to evaluate the Pleiades HS, right? The way it is stated, it sounds like you're using to evaluate both.

Page 32:
8: There are also studies by Shaw et al. (2019?). Please review Simon Gascoin and Etienne Berthier's recent publications, as they are listed on other recent papers using Pleiades for accuracy of DEM difference maps, including snow depth
12: I don't think it's fair to present your "cleaned" values here
12: Again, rather than comparing DSM accuracy, you're comparing snow depth accuracy, which includes error from the reference dataset.
13: "This is higher" here is a bit confusing, as your numbers are lower. Consider rephrasing. Also "this" is ambiguous.
14: Why is one pixel considered the maximum??? Sub-pixel correlation should be capable of 0.1-0.2 px accuracy. Is the issue with your manual GCP identification? Still should be able to achieve sub-pixel with this, esp with modern GCP markers.
18-19: I don't follow the last sentence, but I may just be getting tired :) Consider rewording. What is "they"
21: Need to qualify this with "from Pleiades, without further correction". We have demonstrated better accuracy with WorldView-3 DEM difference (same as snow depth) products (see McGrath et al, 2019; Shean et al., 2016). And there are several systematic errors in the individual DSM products (e.g., CCD offsets, unmodeled attitude error ["jitter"], parallax issues in L1B camera models) that can be corrected to further improve accuracy. We are actively working on this, so hopefully Pleaides DSM accuracy can be improved!
25: I would avoid using the world "profit" to avoid confusion with commercial applications, "benefit" would be better

28-30: I disagree. ICP co-registration approaches can work very well, even when only sparse exposed surfaces are available. Co-registration using methods like Nuth and Kaab (2011) can also be used to take advantage of the slope and aspect-dependent dh - can then use limited snow-free surfaces for final vertical correction. I think you may be referring to cases when the entire scene is 100% snow-covered (rare for mountains).

Page 33:
14: Issues of contrast can also be problematic in satellite imagery. In my experience, it is more about fresh snowfall and whether image GSD is fine enough to capture relevant length scales of surface roughness.
18: An aircraft outfitted with high-end GNSS and IMU should be able to provide very accurate camera position and orientation data, eliminating the need for GCPs (as with your eBee RTK results)
19: Again "higher GSD" - careful here, as lower numeric value means better resolution
24-25: This is consistent with my experience using eBee RTK platform. But careful about generalizing to all "UAS photogrammetry" - a DJI Phantom with no GCPs is not nearly as capable

Page 34:
8: This is where using a 3-4 m pole can be beneficial.
14: "non-negligible"
20: Careful with this - using a DTM may help in some areas, but not others; "bushes" is pretty generic, and a bush with leaves appears very different than a leaf-free bush to a laser or camera

Figure 2:
Check TC date formatting standards - I initially read this as MM.DD
If you're going to remake this figure, it might be better to alter the aspect ratio to reduce the width of the x axis, so we can better see the magnitude of the change over the study period (right now all lines look really flat). Either that or add thin horizontal gridlines and a complementary right axis label.

Figure 4:
I would move this to supplemental figure - it is awfully large for the information you're trying to convey, or could be presented as a table
Need to define HS in caption
I'm still confused by comparison 3. Why is comparison 3 HS_platform_ALS?

Figure 5:
d) I don't see the violet stars?  If they are present, recommend making larger and using a color that won't blend in with the orthoimage

Figure 6:

The color ramp used here makes it very difficult to distinguish snow depth values between 0-1 m. It would be better to use a perceptually uniform, linear color ramp, ideally with labels for increments of 1.0 or 0.5 m intervals. I realize this may not be straightforward in ArcGIS.

Figure 7:
Maybe use "apparent snow depth" or "snow depth estimate" here and elsewhere in the paper, as it's not physically possible to have a negative snow depth
Why did color ramp change here to what looks like matplotlib plasma? Also, since you are showing values from -3 to 3 m, could be better to use a diverging color ramp. Lots of good resources on the theory behind these visualization approaches:
https://matplotlib.org/3.1.1/tutorials/colors/colormaps.html

Figure 8:
This is a nice figure, but why such a large bin size??? The quantization here makes comparison between different sources really challenging. I would recommend bin size of <5-10 cm, so we can properly assess each distribution.
Also, it would be valuable to create a mask for the common intersection of valid HS pixels (ie pixels where all 4 DSMs have a valid elevation), clip each input HS to the same mask, then produce a similar histogram, maybe as a second panel in this figure. This provides an "apples to apples" comparison, as your current histograms are sampling different portions of the domain, and there is no reason to expect your reported mean and std statistics to be the same.
In the caption, you can just say "normalized histogram" without details about dividing by total number and multiplying by 100.
What does "all not shown" mean? You have the Manual and snow poles measurement on histogram on the plot.

Figure 9:
Is "the median" the same as your MABE metric? I think MBE is just the mean error, right?
5th and 95th are not quartiles, they are percentiles. Fix in all other captions for box plots.

Figure 11:
I think this is a 2-D histogram showing density? So, not a scatterplot. I don't think your current caption says what color represents.
Probably want to say "y = x"
Should mention in caption why the bottom row is limited to range of 0-5 m on y axis. I think this is labeled "cleaned" but you're adding another filter here.

Table 1:
Time is local or UTC?
Add a row for sensor dimensions in pixels and/or mm. 450 MP is not really a "resolution" and we don't know dimensions.

What do you mean by Pleaides mean GSD of 0.7 (resampled to 0.5). Did you intentionally oversample, or are the L1B products delivered at higher res after some super-resolution processing beyond normal TDI?

Table 3:
Threshold for classifying outliers

References:
Several of these are "gray literature" and some contain errors (e.g., Deems and Painter, 2006 has no journal information, year listed twice)
Authors should review all references carefully, update according to TC policy, remove gray literature, and update lingering errors from citation manager software

---

## Referee Comment (RC2) · Edward Bair (Referee) · 7 Aug 2020

In "Intercomparison of photogrammetric platforms for spatially continuous snow depth mapping" several different photogrammetric approaches are tested in a sparsely vegetated study area in Switzerland. This manuscript is a thorough and useful comparison of state of the art photogrammetric tools for snow depth mapping. It does somewhat read like a commercial for the eBee+ UAS, however the authors have stated no conflicting interests and have demonstrated the eBee's advantages over the other platforms. I would recommend this manuscript for publication subject to a few minor changes.

1) The vegetation issues are discussed, but a solution is not presented. I suggest identification of these regions and spatial interpolation may be the best approach, but there are other solutions. Note that negative snow depths can occur in glacierized areas or areas with persistent snow cover as well with snow on/off differencing approaches.

2) There are numerous grammatical and stylistic errors. I suggest an English language service be used prior to publication.

3) The manual validation effort is impressive in scope but seems unnecessary. It seems to me that the resources used could have been better used on snow-covered and snow-free lidar flights, perhaps along with the Ultracam. And why weren't any forested areas sampled manually?

4) I don't agree that the Ultracam showed robust enough performance, especially in forested areas, to be called ground truth.

Other minor critiques are attached as an annotated PDF.

If the authors have any questions, I encourage them to contact me at nbair@eri.ucsb.edu.

Ned Bair 2020-08-06

Please also note the supplement to this comment:
https://tc.copernicus.org/preprints/tc-2020-93/tc-2020-93-RC2-supplement.pdf

**Supplement:**

[revised manuscript text omitted]

---

## Author Comment (AC1) · 3 Sep 2020

Dear Mr. Deschamps-Berger

Thank you very much for the comments. The point you make about the methodological differences that led to the reported values in each paper is very important and should have been clarified and discussed in Section 6.1, so that the relative performance across the three studies can be better interpreted.

We agree with your first point that we should discuss our results in relation to the size of the test area, which at 3.59 km2 is very small. But we put a lot of emphasis on

the validation with the eBee+, because the ultracam has never been tested on snow. Bühler et al, 2015 used an ADS80-SH52 sensor which is an optoelectronic line scanner and not an aerial camera. Furthermore, the imagery was processed with ATE SOCET SET. Also, the imagery had an average GSD of 0.25 m and not 0.06 m as we had. Nolan et al, 2015 used an aerial camera system but nothing similar to Ultracam.

To your second point we would like to say that we do calculate snow depth maps but we only compare the snow-on DSMs. Whenever we compare two snow depth maps, the snow-off reference is the same. Therefore, the error from the snow-off DSM is the same for both subsequent comparison DSMs and will not influence the results.

We will therefore refine our discussion, especially section 6.1 to clarify and expand the methodological differences you pointed out.

Bühler, Y., Marty, M., Egli, L., Veitinger, J., Jonas, T., Thee, P. and Ginzler, C.: Snow depth mapping in high-alpine catchments using digital photogrammetry, The Cryosphere 9(1), 229–243, doi:10.5194/tc-9-229-2015, 2015.

Nolan, M., Larsen, C. and Sturm, M.:Mapping snow depth from manned aircraft on landscape scales at centimeter resolution using structure-from-motion photogramme-try, The Cryosphere, 1445–1463, doi:10.5194/tc-9-1445-2015, 2015.

---

## Author Comment (AC2) · 4 Sep 2020

In "Intercomparison of photogrammetric platforms for spatially continuous snow depth mapping" several different photogrammetric approaches are tested in a sparsely vegetated study area in Switzerland. This manuscript is a thorough and useful comparison of state of the art photogrammetric tools for snow depth mapping. It does somewhat read like a commercial for the eBee+ UAS, however the authors have stated no conflicting interests and have demonstrated the eBee's advantages over the other platforms. I would recommend this manuscript for publication subject to a few minor changes.

**Dear Dr. Bair**

**We thank you very much for your review. We absolutely have no conflict of interest concerning the eBee+ UAS. The institute from which most authors are affiliated uses from a range of manufactures. After testing multiple platforms for operational use in alpine environments and for mapping snow depth we determined the eBee+ had the best performance. However, we will change the text in such a way that the name eBee+ is mentioned less.**

1) The vegetation issues are discussed, but a solution is not presented. I suggest identification of these regions and spatial interpolation may be the best approach, but there are other solutions. Note that negative snow depths can occur in glacierized areas or areas with persistent snow cover as well with snow on/off differencing approaches.

   **Vegetation errors are not easy to eliminate, especially when generated by grassland with various length and uncertain compaction between snow-free and snow-covered conditions during mapping. We agree with the related comment added directly to our manuscript that this limitation of the photogrammetric method maybe insurmountable. In principle, we also agree that a DTM may produce a small improvement for some vegetation, but not for all vegetation. LIDAR in particular is more capable to mitigate this effect thanks to penetration through vegetation as we stressed in Section 6.5. Nonetheless, we disagree that interpolation from open areas onto regions classified as vegetated are the best approach in general. While it may be true on relatively uniform terrain, we believe that in mountainous regions the variability in topography maybe too great to be accurately captured by interpolation from remote open areas. In our opinion, it is of rather minor importance for this specific publication as the investigation is focused primarily above tree line. But we will take this point into consideration and add a comment in section 6.5 of the paper.**

2) There are numerous grammatical and stylistic errors. I suggest an English language service be used prior to publication.

   **The paper will be proof-read by a native English speaker.**

3) The manual validation effort is impressive in scope but seems unnecessary. It seems to me that the resources used could have been better used on snow-covered and snowfree lidar flights, perhaps along with the Ultracam. And why weren't any forested areas sampled manually?

   **We thank the reviewer for his appreciation of our efforts towards validation, although we respectfully disagree with the comment that it was unnecessary. We agree that in principle, repeated lidar flight would have been desirable for validation. However and despite what the reviewer may assume, the associated cost remains very high and would have exceeded the project funding, especially given that an additional manned flight in summer would have been**

**required. We did attempt a winter TLS scan, but due to various recording problems including suboptimal viewing angles, we could not georeference the TLS scan correctly. Therefore, the manual measurements are an important and necessary independent control for the snow depth maps. The large area comparison is there to further assess the Pleiades data, recognizing that the Ultracam is significantly better on the small scale test area, and hence can be used as reference. If "ground truth" is really what bothers the reviewer, then you could propose to rephrase with "reference" and adding the above element as justification.**

**We did not sample any forest areas manually due to lack of time and manpower. We wanted to concentrate on the Schürlialp area with manual sampling.**

4) I don't agree that the Ultracam showed robust enough performance, especially inforested areas, to be called ground truth.

**We agree with you when specifically considering forested terrain. However, when considering the Ultracam DSM in non-forested terrain the RMSE was 0.17 m and the NMAD was of 0.12 m (raw data), which, we believe is sufficiently accurate to be compared with the Pleiades snow depth map over a larger area. The large area comparison is there to further assess the Pleiades data, recognizing that the Ultracam is significantly better on the small scale test area, and hence can be used as reference. We agree that ground-truth is not the right word here and will replace it with reference.**

Responses to questions in the paper:

Page 6:
Figure 2:
What type of automated sensors were used, how often were the mesurements recorded, and most importantly, how were the depths filtered? Snow depths from acoustic sensors never look that flat. I'm guessing a single time is shown from each day?

**For each day and each station one value is displayed. 5DF, 5MA and 5WJ are SLF observers reading the snow depth from a snow pole manually on a measurement field every day. The time for manual measurements is between 07:00 and 07:30. For the automatic stations, the newest value whose measurement falls within the time window 06:30 to 08:15 is displayed. The snow depth at the automatic stations is measured with ultrasound every 30 minutes. We will add this to the caption.**

Page 8:
3: That's unusually accurate for a drone. Are there other UAS with similar geolocational accuracy?

**Yes, current RTK UAS can attain such accuracies. Other systems with comparable accuracies are the WingtraOne or the Trimble UX HP.**

Page 11:
7: Why wasn't the ALS company hired for winter acquisitions. That would have avoided the bent/missing poles and likely would have produced more accurate snow depths.

**We agree with you that an ALS flight would have been useful for the comparison but unfortunately it was not feasible for the project as explained in our response to an earlier comment.**

Page 14:
26: Any speculation as to why the RMSE in the vertical direction of the summer flight was half that of the winter flight?

**At the time of the summer flight we had a better DGNSS (Stonex S800) available with an accuracy in position of 0.014 to 0.022 m and in altitude of 0.02 m. This of course affected the error calculation. We will specify this change of DGNSS in the revised paper.**

Page 20:
These images are pixelated and I cannot see the violet stars in (d)

**We will improve the image quality. The purple stars were unfortunately forgotten during the reproduction of the image.**

Page 21:
10: Negative snow depths are errors. In addition to vegetation, glacierized areas or those with permanent snow can show the same effect. The vegetation problems are a significant limitation of the photogrammetric methods vs lidar. It deserves to be mentioned that the photogrammetric methods have sometimes insurmountable problems in vegetation. No well-versed reader expects one method to work well everywhere.

**In many cases we agree with you, but dense vegetation (e.g. *Alnus alnobetula*) present in the study area can also pose a problem to LIDAR. Reutebuch et al., 2003 for example examined the accuracy of a lidar terrain model under a conifer forest canopy and found poorer accuracy for dense canopy. We are currently investigating this topic further by deploying LiDAR on a UAS platform, but this analysis will not be included in this study. But it is a fact that with photogrammetric techniques in dense vegetation you are at the limit at some point and you can never make a DTM like with a LIDAR. We will mention this in the discussion.**

**Reutebuch, S. E., Mc Gaughey, R. J., Andersen, H. E., & Carson, W. W. (2003). Accuracy of a high-resolution lidar terrain model under a conifer forest canopy. *Canadian Journal of Remote Sensing, 29*(5), 527-535. doi:10.5589/m03-022**

Page 24:
Panels a-d need to be captioned explicitly: Ultracam (a), Ebee +(b),….

**We will add this to the caption.**

Page 29:
5: I don't agree that the Ultracam was tested thoroughly enough to be used as ground truth, specifically its performance in forests

**Please see the comment below.**

9: All of this underscores the fact that photogrammetric methods are not suitable for estimating snow depth in forested areas, especially not as a ground truth. Without an accurate and independent validation datasets to compare to, e.g. snow on/off TLS or ALS, I suggest simply comparing the snow depths from the different sensors rather than claiming the Ultracam can be used as ground truth.

**We agree with you on this to a certain extent and will reformulate this statement. However, the comparison of Pléiades and Ultracam is nevertheless a reasonable comparison when considering the accuracy of the Pléiades data over a large geographic area and in particular within non-forested regions, which makes up > 80% of the comparison.**

Page 30:

Table: The average snow depth, based on the Schürlialp measurements is ~ 1.3 m. Thus, the uncertainty exceeds or is close to the average snow depth

**Yes, good point. We will add this to the discussion of the results.**

Page 34:

18: Vegetation is a problem for all remote sensing techniques in snow. Sometimes the best method is interpolation from open areas where you have more confidence in your measurements. Perhaps that is the best solution for filling these negative values?

**We do not agree in this situation. We believe interpolation would give valid results in some areas and not others since snow depth is highly variable, even over short distances, and dependent on underlying topography.**

Page 36:

10: And as you say, the snowpack needs to be deep. For example, the eBEE might be great for tundra snowpacks, except for the high windows and depths that are < 0.5 m.

**Yes, good point. We will add a comment in the Conclusion.**

---

## Author Comment (AC3) · 21 Sep 2020

Response to the "Review of "Intercomparison of photogrammetric platforms for spatially continuous snow depth mapping" by Eberhard et al."

https://tc.copernicus.org/preprints/tc-2020-93/#discussion

David Shean

July 29, 2020

This paper considers several photogrammetric approaches that can be used to produce gridded surface elevation products (satellite stereo, aerial, UAS and terrestrial), which can in turn be differenced to provide snow depth estimates. These methods and products are evaluated over one test site in the Swiss Alps, using photogrammetric data collected within a short time period, contemporaneous ground-truth observations and several external datasets. The authors provide some analysis of their snow depth map quality, and qualitative comparisons of the strengths and weaknesses of the different methods.

In general, this is a nice methods paper. The amount of work presented is substantial. The campaign planning was excellent, the photogrammetry processing and methods are sound, the resulting datasets are impressive, and the writing is generally good. There are many DSM processing challenges involved with precise snow depth mapping, especially when surfaces in the "snow-on" DSM are almost entirely snow-covered. The authors use more traditional, labor-intensive approaches to integrate control data early in the process, rather than alternative approaches involving point-cloud or DSM co-registration later in the process. Some decisions about data processing and presentation should be revisited, as they will impact the quantitative results, though they may not have a substantial impact on the qualitative conclusions. The writing in the results and discussion requires some improvement and I recommend that the more senior authors on the paper dedicate time to help refine these sections.

As a fellow photogrammetry enthusiast/evangelist, I enjoyed reading this paper, and had a lot to say, offering many general and specific comments. I realize that not all of these changes will be possible, but I hope that the authors will consider these suggestions to improve this paper and their future work. I look forward to reading the response and seeing this nice study published!

**Dear Prof. Shean**

**Thank you very much for your valuable and extensive review. We have addressed each of your comments below.**

**General Comments**

• There is ambiguity with the term "ground-based", as many readers may interpret this to be "ground-truth" (I made this mistake a few times). I would suggest using "terrestrial photogrammetry" or "close-range photogrammetry" throughout the text to avoid issues.

**We will use "terrestrial photogrammetry".**

• I would offer that snow depth on glaciers and ice sheets is critical for properly measuring glacier mass balance, beyond the applications you list.

We will add the application for properly measuring the glacier mass balance. We will cite Gascoin et al., 2011 and McGrath et al., 2015.

Gascoin, S., Kinnard, C., Ponce, R., Lhermitte, S., MacDonell, S., and Rabatel, A.: Glacier contribution to streamflow in two headwaters of the Huasco River, Dry Andes of Chile, The Cryosphere, 5, 1099-1113, 10.5194/tc-5-1099-2011, 2011.

McGrath, D., Sass, L., O'Neel, S., Arendt, A., Wolken, G., Gusmeroli, A., Kienholz, C., and McNeil, C.: Endof-winter snow depth variability on glaciers in Alaska, Journal of Geophysical Research: Earth Surface, 120, 1530-1550, 10.1002/2015jf003539, 2015.

• The summer reference data sets were acquired on June 27, 2018 and August 5-6, 2018. Were all surfaces completely snow-free during both periods? For some mountain sites with similar elevation here in Pacific Northwest, we see considerable snow in late June. Do you expect differences in the vegetation during these periods? How do you account for errors introduced by these issues?

All areas were free of snow during both summer flights. We will clarify this accordingly in the revised manuscript. There are indeed differences in the vegetation between the two summer DSMs. Nevertheless, when comparing two snow depth maps, we always resample the winter DSMs to align to the same summer DSM (i.e., either the UAS or the LiDAR). Therefore, the differences between both summer DSMs does not play any part in the comparative outcome of Comparison 2 and Comparison 3.

• I wasn't clear on the strategy used for blending of the independent Pleiades DSMs to create a final triplet DSM product. If the bundle block adjustment was successful, and both P12 and P13 are usable, shouldn't P23 also be usable, even with more limited convergence angle? The text cites Sirguey and Lewis (2019), which is listed as "Sirguey, P., and Lewis, C.: Topographic mapping of Franz Josef glacier region with Pléiades satellite imagery, University of Otago, 2019." - I can't find a published version of this reference, which I think is a technical paper? Need to describe the method in this text if it is not yet published elsewhere. The last sentence of this section mentions a filter to remove triangulation error >0.5 m on the blended DSM. Why not apply this filter to the individual DSMs before blending?

The reference cited is indeed a technical report. Our approach to blending surfaces is a simple weighted arithmetic mean between elevations from each bi-stereo surface, with the weight being the ray-intersection error, which is an output of *point2dem*.

In order to clarify this, we will modify the manuscript as follows: "In tri stereo configuration, maps of intersection errors are used to weight the contribution of pairwise DSMs into a blended DSM with GDAL (Sirguey and Lewis, 2019). In tri-stereo configuration, we generate a DSM and map of ray intersection error for each stereo-pair. We blend DSMs with GDAL using a weighted arithmetic mean, whereby the elevation from each constituent DSM is weighted by its corresponding ray intersection error (Sirguey and Lewis, 2019). A map of standard error of the weighted mean is generated by uncertainty propagation."

The bundle adjustment was successful and all pairs are usable. When producing the surface however, it became apparent from hillshades that P23 with the lowest B/H ratio resulted in a substantially more noise in areas of poor contrast. We illustrate this below with subset hillshades of the surfaces

produced form our triangulated triplet, namely the three bi-stereo (a, b, c), our blending of the three bi-stereo (e), the blending of only P12 and P23 (f). The ortho image (d) also illustrates the challenging contrast of our snowpack.

It is clear that decreasing B/H corresponds to increasing amount of noise from the restitution on smooth, undisturbed areas of our snowpack (see a, b, c). This is expected since the lower contrast promotes variability in stereo-matching, which then compounds with lower parallactic angles when B/H decreases, resulting in more dispersion in the triangulated height on poorly textured areas of our snowpack. It appears in our case that restitution with P23 in such poorly textured areas (c) generated a variability that propagated too much in the blended surfaces despite the reduced weight from larger intersection error (e). Note however that with more contrast/texture (e.g., several avalanches are visible on the northern slopes in the ortho), then stereo matching dispersion in P23 compares to other pairs which demonstrates that the issue is not from the triangulation, but rather support our hypothesis that contrast and B/H combine to dictate the dispersion of the restitution. While an indepth study may be desirable to characterize this further, including testing more stereo-matching options and satellite geometry in such environment, for the purpose of our comparison with other platforms, we only needed to test what we considered to be the best Pleiades surface, and it was obvious that the blending only P12 and P23 was more suitable (e) vs (f).

We do not remove points from individual DSM before the blending because the rationale of the blending is to statistically leverage (with the weighted mean) several estimates of elevation while considering their initial quality, with their contribution weighted by the uncertainty measured by the ray intersection error. We don't believe that applying such a threshold individually to each DSM beforehand finds better justification. On the contrary, we think it is more appropriate to complete the blending initially without prejudice, compute the propagated standard error of the weighted mean, and then apply the binary filtering only once to the latter only. This has the additional advantage of applying the threshold on a single map of standard error (analogous to the ray intersection error but applying to the weighted mean) representative of the statistical leveraging effect.

In order to further clarify our manuscript, we will modify: *"Finally, the <del>ray intersection error</del> map of standard error for the blended DSM was used to set all cells of the DSM to no data where the ray intersection error is greater than one panchromatic pixel or 0.5 m as larger errors were found to be often indicative of erroneous stereo-matching".*

---

## Author Response (AR1)

**Responses to the review of Dr. Bair**

In "Intercomparison of photogrammetric platforms for spatially continuous snow depth mapping" several different photogrammetric approaches are tested in a sparsely vegetated study area in Switzerland. This manuscript is a thorough and useful comparison of state of the art photogrammetric tools for snow depth mapping. It does somewhat read like a commercial for the eBee+ UAS, however the authors have stated no conflicting interests and have demonstrated the eBee's advantages over the other platforms. I would recommend this manuscript for publication subject to a few minor changes.

**Dear Dr. Bair**

**We thank you very much for your review. We absolutely have no conflict of interest concerning the eBee+ UAS. The institute from which most authors are affiliated uses from a range of manufactures. After testing multiple platforms for operational use in alpine environments and for mapping snow depth we determined the eBee+ had the best performance. However, we will change the text in such a way that the name eBee+ is mentioned less.**

1) The vegetation issues are discussed, but a solution is not presented. I suggest identification of these regions and spatial interpolation may be the best approach, but there are other solutions. Note that negative snow depths can occur in glacierized areas or areas with persistent snow cover as well with snow on/off differencing approaches.

   **Vegetation errors are not easy to eliminate, especially when generated by grassland with various length and uncertain compaction between snow-free and snow-covered conditions during mapping. We agree with the related comment added directly to our manuscript that this limitation of the photogrammetric method maybe insurmountable. In principle, we also agree that a DTM may produce a small improvement for some vegetation, but not for all vegetation. LIDAR in particular is more capable to mitigate this effect thanks to penetration through vegetation as we stressed in Section 6.5. Nonetheless, we disagree that interpolation from open areas onto regions classified as vegetated are the best approach in general. While it may be true on relatively uniform terrain, we believe that in mountainous regions the variability in topography maybe too great to be accurately captured by interpolation from remote open areas. In our opinion, it is of rather minor importance for this specific publication as the investigation is focused primarily above tree line. But we will take this point into consideration and add a comment in section 6.5 of the paper.**

2) There are numerous grammatical and stylistic errors. I suggest an English language service be used prior to publication.

   **The paper will be proof-read by a native English speaker.**

3) The manual validation effort is impressive in scope but seems unnecessary. It seems to me that the resources used could have been better used on snow-covered and snowfree lidar flights, perhaps along with the Ultracam. And why weren't any forested areas sampled manually?

   **We thank the reviewer for his appreciation of our efforts towards validation, although we respectfully disagree with the comment that it was unnecessary. We agree that in principle,**

**repeated lidar flight would have been desirable for validation. However and despite what the reviewer may assume, the associated cost remains very high and would have exceeded the project funding, especially given that an additional manned flight in summer would have been required. We did attempt a winter TLS scan, but due to various recording problems including suboptimal viewing angles, we could not georeference the TLS scan correctly. Therefore, the manual measurements are an important and necessary independent control for the snow depth maps. The large area comparison is there to further assess the Pleiades data, recognizing that the Ultracam is significantly better on the small scale test area, and hence can be used as reference. If "ground truth" is really what bothers the reviewer, then you could propose to rephrase with "reference" and adding the above element as justification.**

**We did not sample any forest areas manually due to lack of time and manpower. We wanted to concentrate on the Schürlialp area with manual sampling.**

4) I don't agree that the Ultracam showed robust enough performance, especially in forested areas, to be called ground truth.

**We agree with you when specifically considering forested terrain. However, when considering the Ultracam DSM in non-forested terrain the RMSE was 0.17 m and the NMAD was of 0.12 m (raw data), which, we believe is sufficiently accurate to be compared with the Pleiades snow depth map over a larger area. The large area comparison is there to further assess the Pleiades data, recognizing that the Ultracam is significantly better on the small scale test area, and hence can be used as reference. We agree that ground-truth is not the right word here and will replace it with reference.**

Responses to questions in the paper:

Page 6:
Figure 2:
What type of automated sensors were used, how often were the mesurements recorded, and most importantly, how were the depths filtered? Snow depths from acoustic sensors never look that flat. I'm guessing a single time is shown from each day?

**For each day and each station one value is displayed. 5DF, 5MA and 5WJ are SLF observers reading the snow depth from a snow pole manually on a measurement field every day. The time for manual measurements is between 07:00 and 07:30. For the automatic stations, the newest value whose measurement falls within the time window 06:30 to 08:15 is displayed. The snow depth at the automatic stations is measured with ultrasound every 30 minutes. We will add this to the caption.**

Page 8:
3: That's unusually accurate for a drone. Are there other UAS with similar geolocational accuracy?

**Yes, current RTK UAS can attain such accuracies. Other systems with comparable accuracies are the WingtraOne or the Trimble UX HP.**

Page 11:
7: Why wasn't the ALS company hired for winter acquisitions. That would have avoided the bent/missing poles and likely would have produced more accurate snow depths.

**We agree with you that an ALS flight would have been useful for the comparison but unfortunately it was not feasible for the project as explained in our response to an earlier comment.**

Page 14:
26: Any speculation as to why the RMSE in the vertical direction of the summer flight was half that of the winter flight?

**At the time of the summer flight we had a better DGNSS (Stonex S800) available with an accuracy in position of 0.014 to 0.022 m and in altitude of 0.02 m. This of course affected the error calculation. We will specify this change of DGNSS in the revised paper.**

Page 20:
These images are pixelated and I cannot see the violet stars in (d)

**We will improve the image quality. The purple stars were unfortunately forgotten during the reproduction of the image.**

Page 21:
10: Negative snow depths are errors. In addition to vegetation, glacierized areas or those with permanent snow can show the same effect. The vegetation problems are a significant limitation of the photogrammetric methods vs lidar. It deserves to be mentioned that the photogrammetric methods have sometimes insurmountable problems in vegetation. No well-versed reader expects one method to work well everywhere.

**In many cases we agree with you, but dense vegetation (e.g. _Alnus alnobetula_) present in the study area can also pose a problem to LIDAR. Reutebuch et al., 2003 for example examined the accuracy of a lidar terrain model under a conifer forest canopy and found poorer accuracy for dense canopy. We are currently investigating this topic further by deploying LiDAR on a UAS platform, but this analysis will not be included in this study. But it is a fact that with photogrammetric techniques in dense vegetation you are at the limit at some point and you can never make a DTM like with a LIDAR. We will mention this in the discussion.**

**Reutebuch, S. E., Mc Gaughey, R. J., Andersen, H. E., & Carson, W. W. (2003). Accuracy of a high-resolution lidar terrain model under a conifer forest canopy. _Canadian Journal of Remote Sensing, 29_(5), 527-535. doi:10.5589/m03-022**

Page 24:
Panels a-d need to be captioned explicitly: Ultracam (a), Ebee +(b),….

**We will add this to the caption.**

Page 29:
5: I don't agree that the Ultracam was tested thoroughly enough to be used as ground truth, specifically its performance in forests

**Please see the comment below.**

9: All of this underscores the fact that photogrammetric methods are not suitable for estimating snow depth in forested areas, especially not as a ground truth. Without an accurate and independent validation datasets to compare to, e.g. snow on/off TLS or ALS, I suggest simply comparing the snow depths from the different sensors rather than claiming the Ultracam can be used as ground truth.

**We agree with you on this to a certain extent and will reformulate this statement. However, the comparison of Pléiades and Ultracam is nevertheless a reasonable comparison when considering the accuracy of the Pléiades data over a large geographic area and in particular within non-forested regions, which makes up > 80% of the comparison.**

Page 30:

Table: The average snow depth, based on the Schürlialp measurements is ~ 1.3 m. Thus, the uncertainty exceeds or is close to the average snow depth

**Yes, good point. We will add this to the discussion of the results.**

Page 34:

18: Vegetation is a problem for all remote sensing techniques in snow. Sometimes the best method is interpolation from open areas where you have more confidence in your measurements. Perhaps that is the best solution for filling these negative values?

**We do not agree in this situation. We believe interpolation would give valid results in some areas and not others since snow depth is highly variable, even over short distances, and dependent on underlying topography.**

Page 36:

10: And as you say, the snowpack needs to be deep. For example, the eBEE might be great for tundra snowpacks, except for the high windows and depths that are < 0.5 m.

**Yes, good point. We will add a comment in the Conclusion.**

**Response to the review of Prof. Shean**

https://tc.copernicus.org/preprints/tc-2020-93/#discussion

David Shean

July 29, 2020

This paper considers several photogrammetric approaches that can be used to produce gridded surface elevation products (satellite stereo, aerial, UAS and terrestrial), which can in turn be differenced to provide snow depth estimates. These methods and products are evaluated over one test site in the Swiss Alps, using photogrammetric data collected within a short time period, contemporaneous ground-truth observations and several external datasets. The authors provide some analysis of their snow depth map quality, and qualitative comparisons of the strengths and weaknesses of the different methods.

In general, this is a nice methods paper. The amount of work presented is substantial. The campaign planning was excellent, the photogrammetry processing and methods are sound, the resulting datasets are impressive, and the writing is generally good. There are many DSM processing challenges involved with precise snow depth mapping, especially when surfaces in the "snow-on" DSM are almost entirely snow-covered. The authors use more traditional, labor-intensive approaches to integrate control data early in the process, rather than alternative approaches involving point-cloud or DSM co-registration later in the process. Some decisions about data processing and presentation should be revisited, as they will impact the quantitative results, though they may not have a substantial impact on the qualitative conclusions. The writing in the results and discussion requires some improvement and I recommend that the more senior authors on the paper dedicate time to help refine these sections.

As a fellow photogrammetry enthusiast/evangelist, I enjoyed reading this paper, and had a lot to say, offering many general and specific comments. I realize that not all of these changes will be possible, but I hope that the authors will consider these suggestions to improve this paper and their future work. I look forward to reading the response and seeing this nice study published!

**Dear Prof. Shean**

**Thank you very much for your valuable and extensive review. We have addressed each of your comments below.**

General Comments

● There is ambiguity with the term "ground-based", as many readers may interpret this to be "ground-truth" (I made this mistake a few times). I would suggest using "terrestrial photogrammetry" or "close-range photogrammetry" throughout the text to avoid issues.

**We will use "terrestrial photogrammetry".**

● I would offer that snow depth on glaciers and ice sheets is critical for properly measuring glacier mass balance, beyond the applications you list.

**We will add the application for properly measuring the glacier mass balance. We will cite Gascoin et al., 2011 and McGrath et al., 2015.**

**Gascoin, S., Kinnard, C., Ponce, R., Lhermitte, S., MacDonell, S., and Rabatel, A.: Glacier contribution to streamflow in two headwaters of the Huasco River, Dry Andes of Chile, The Cryosphere, 5, 1099-1113, 10.5194/tc-5-1099-2011, 2011.**

**McGrath, D., Sass, L., O'Neel, S., Arendt, A., Wolken, G., Gusmeroli, A., Kienholz, C., and McNeil, C.: End-of-winter snow depth variability on glaciers in Alaska, Journal of Geophysical Research: Earth Surface, 120, 1530-1550, 10.1002/2015jf003539, 2015.**

● The summer reference data sets were acquired on June 27, 2018 and August 5-6, 2018. Were all surfaces completely snow-free during both periods? For some mountain sites with similar elevation here in Pacific Northwest, we see considerable snow in late June. Do you expect differences in the vegetation during these periods? How do you account for errors introduced by these issues?

**All areas were free of snow during both summer flights. We will clarify this accordingly in the revised manuscript. There are indeed differences in the vegetation between the two summer DSMs. Nevertheless, when comparing two snow depth maps, we always resample the winter DSMs to align to the same summer DSM (i.e., either the UAS or the LiDAR). Therefore, the differences between both summer DSMs does not play any part in the comparative outcome of Comparison 2 and Comparison 3.**

● I wasn't clear on the strategy used for blending of the independent Pleiades DSMs to create a final triplet DSM product. If the bundle block adjustment was successful, and both P12 and P13 are usable, shouldn't P23 also be usable, even with more limited convergence angle? The text cites Sirguey and Lewis (2019), which is listed as "Sirguey, P., and Lewis, C.: Topographic mapping of Franz Josef glacier region with Pléiades satellite imagery, University of Otago, 2019." - I can't find a published version of this reference, which I think is a technical paper? Need to describe the method in this text if it is not yet published elsewhere. The last sentence of this section mentions a filter to remove triangulation error >0.5 m on the blended DSM. Why not apply this filter to the individual DSMs before blending?

**The reference cited is indeed a technical report. Our approach to blending surfaces is a simple weighted arithmetic mean between elevations from each bi-stereo surface, with the weight being the ray-intersection error, which is an output of** *point2dem*.

**In order to clarify this, we will modify the manuscript as follows:** *" In tri-stereo configuration, we generate a DSM and map of ray intersection error for each stereo-pair. We blend DSMs with GDAL using a weighted arithmetic mean, whereby the elevation from each constituent DSM is weighted by its corresponding ray intersection error (Sirguey and Lewis, 2019). A map of standard error of the weighted mean is generated by uncertainty propagation."*

**The bundle adjustment was successful and all pairs are usable. When producing the surface however, it became apparent from hillshades that P23 with the lowest B/H ratio resulted in a substantially more noise in areas of poor contrast. We illustrate this below with subset hillshades of the surfaces**

produced form our triangulated triplet, namely the three bi-stereo (a, b, c), our blending of the three bi-stereo (e), the blending of only P12 and P23 (f). The ortho image (d) also illustrates the challenging contrast of our snowpack.

It is clear that decreasing B/H corresponds to increasing amount of noise from the restitution on smooth, undisturbed areas of our snowpack (see a, b, c). This is expected since the lower contrast promotes variability in stereo-matching, which then compounds with lower parallactic angles when B/H decreases, resulting in more dispersion in the triangulated height on poorly textured areas of our snowpack. It appears in our case that restitution with P23 in such poorly textured areas (c) generated a variability that propagated too much in the blended surfaces despite the reduced weight from larger intersection error (e). Note however that with more contrast/texture (e.g., several avalanches are visible on the northern slopes in the ortho), then stereo matching dispersion in P23 compares to other pairs which demonstrates that the issue is not from the triangulation, but rather support our hypothesis that contrast and B/H combine to dictate the dispersion of the restitution. While an in-depth study may be desirable to characterize this further, including testing more stereo-matching options and satellite geometry in such environment, for the purpose of our comparison with other platforms, we only needed to test what we considered to be the best Pleiades surface, and it was obvious that the blending only P12 and P23 was more suitable (e) vs (f).

We do not remove points from individual DSM before the blending because the rationale of the blending is to statistically leverage (with the weighted mean) several estimates of elevation while considering their initial quality, with their contribution weighted by the uncertainty measured by the ray intersection error. We don't believe that applying such a threshold individually to each DSM beforehand finds better justification. On the contrary, we think it is more appropriate to complete the blending initially without prejudice, compute the propagated standard error of the weighted mean, and then apply the binary filtering only once to the latter only. This has the additional advantage of applying the threshold on a single map of standard error (analogous to the ray intersection error but applying to the weighted mean) representative of the statistical leveraging effect.

In order to further clarify our manuscript, we will modify: *"Finally, the  map of standard error for the blended DSM was used to set all cells of the DSM to no data where the ray intersection error is greater than one panchromatic pixel or 0.5 m as larger errors were found to be often indicative of erroneous stereo-matching"*.

[Figure]

(a) P12 front-nadir B/H=0.42

(b) P13 front-back B/H=0.62

(c) P23 nadir-back B/H=0.19

(d) ortho

(e) Blend P12 & P13 & P23

(f) Blend P12 & P13 (best)

● The +/-1 m (total of 2 m magnitude) over 15 km "tilt" of blended Pleiades DSM is a concern, and seems to indicate that the bundle adjustment was unsuccessful, or the GCPs used are inaccurate or poorly distributed. It's not clear why a more rigorous co-registration approach was not used here rather than a hyperplane fit (presumably least-squares? Need to provide more details on this correction) through residual values at sparse points along roads. I would think a 3D rotation would be more suitable. How does the reader know that this grid correction did not introduce additional error over snow-covered surfaces at greater distances from the points on the roads in valley bottoms?

**First, it is important to realize that the process of bundle block adjustment (BBA) will always and inevitably produce some amount of spatial bias with respect to another independent surface. The nature of the adjustment makes it virtually impossible to attain a perfect coherence independently. One challenge of 3D change detection is to detect, characterize, and mitigate/correct systematic biases to improve the signal of change, regardless of the technique used to makes surfaces spatially coherent to each other.**

**Our winter and summer DSMs (ASL and Pleaides DSM, respectively) are such independent products, and the amount of snow and sparsity of stable ground in our image makes an automatic approach to align surfaces (e.g., Iterative Closest Point on stable ground) untenable. Therefore, we must rely on achieving consistent absolute accuracy for each dataset to ensure the coherence.**

**It is reasonable to assume that the summer ALS is more accurate than the Pleaides DSM. The magnitude and spatial structure of the bias (e.g., linear vs nonlinear drift) would then be governed by the quality of the BBA, and in turn, indeed related to the quality in the placement and distribution of GCPs. We show in the additional figure below the distribution of the 14 GCPs we used in the BBA, which we believe are well distributed given the complexity of our terrain.**

**As indicated in our section 3, GCPs have been collected both in winter and summer which complicated the interpretation of some points in the winter image triplet. The visual interpretation of GCPs must also be put into perspective with the 50cm spatial resolution of the imagery. Some variability in GCP placement may have contributed to generate the residual systematic tilt relative to the ALS surface, again assuming that the latter does not or marginally contribute to the tilt. Nonetheless, our Leave-one-out cross-validation (LOOCV) assessment provided a satisfying quality assurance for this step of the process to be valid in view of the 50 cm pixel.**

**One must then keep in mind that systematic spatial errors can generally be revealed and assessed only in retrospect once surfaces are produced. The process by which we determine the remaining tilt is arguably similar to Deschamps-Berger et al. (2020) identifying, characterizing, and mitigating unmodelled residual jitter from evaluation of the snow depth map *a posteriori* because the rigid body transform associated with their alignment strategy could not capture nor correct it.**

**In view of this, we do not agree that a tilt of ±1m over 15km in these conditions can be qualified as an unsuccessful triangulation, nor a cause for concern. On the contrary, we argue that the lack of planimetric offset when differencing surfaces processed independently as we did (which is unforgiving**

in such steep topography when computing 3D changes) is rather a confirmation of the general quality of the BBA. This leaves most systematic bias in the elevation (which again is inevitable), the magnitude of which is believed to be consistent with what an independent photogrammetric process can achieve at this resolution given our constraints.

Our final assessment of the co-registration to detect remaining (and expected) systematic effect once the surface was produced could only be done from critical evaluation of spatial pattern in the difference with ALS, with the roads being the only suitable surfaces free of snow in this image. As shown in our figures below, the distribution of offsets along roads in the city and inside the valley revealed enough linearity to justify the fitting of a plane in 3D space as a correction grid. This is the definition of a best fit hyperplane, which is solved via least squares. We will clarify this in the revised manuscript.

Considering a vertical offset of 1m over a 7500m distance, the difference between this simple grid correction and a rotation involves a difference of less than 0.1mm on the planimetry which is negligible when working with 0.5/2m resolution. We are therefore not sure of what "rigorous approach" the reviewer think could apply better to our case. One option could have been to revisit the BBA in view of the tilt, potentially revisiting the placement of some GCPs. We argue that, while this could mitigate the problem at the expense of time and reprocessing, the effect is not expected to be significantly different to the debiasing we propose *a posteriori*.

We appreciate the reviewer point about the risk for over-correction away from the road. Nevertheless, the correction we apply is a first order linear offset consistent with the residual trend captured from the only suitable locations where such assessment can be done. The locations of the spot points along the roads and inside the city are shown in the figure below (a) and (b). Their distribution along two nearly orthogonal axes effectively constrains the corrective hyperplane as shown in (c) and (d). (a) and (b) also reveal the difference in snow depth mapped before and after correction and does not suggest such problem exist. Finally, snow depth in our test area has also been assessed and validated well away from the roads.

[Figure]

(a) DoD Before tilt correction. Spot points along roads are shown in black and GCPs as triangle.

(b) DoD After tilt correction. Spot points along roads are shown in black and GCPs as triangle.

(c) Hyperplane fitting to residual on spot points.

(d) Hyperplane fitting to residual on spot points.

● Figures 5 and 12 show a clipped version of the Pleaides orthomosaic and snow depth products, with data only shown for the polygon defining the Dischma valley study area. It seems like it might be valuable to show the full unclipped extent of these products, maybe in a supplementary figure. The unclipped products should cover a larger area, adjacent valleys/peaks, and hopefully several of the AWS stations identified in Figure 1.

**Figure 5 (a) shows the whole Pléiades orthoimage, while Figure 12 (a) shows the full map of snow depth derived from Pleiades minus summer ALS; the latter is limited to the area covered by ALS. In Figure 6 the snow depth map calculated with the eBee+ summer flight is shown. Therefore, snow depths exist only for the red polygon. In Figure 12 the snow depth maps calculated with the ALS scan**

**are shown. The summer ALS scan is unfortunately still smaller than the footprints of the Pléiades/Ultracam.**

● The 0.1 m accuracy (is this both horizontal and vertical? Why don't we see expected ~2x higher error in vertical?) for GCP and CP is relatively high for modern GNSS systems, and this will propagate to all of the photogrammetric solutions. Does the 0.1 m value represent RMSE? How was the differential GNSS processing performed? I don't remember seeing any mention of this, so perhaps a real-time correction was used. In the future, I recommend using a survey-grade GNSS for your positions, or longer occupation times with static occupations at each if RTK accuracy is poor.

**Section 4.1 specifies that all GCPs were collected in RTK mode based on a swipos-GIS/GEO correction stream (VRS network). The Trimble Geo XH 6000 is a dual frequency GNSS receiver and was the best equipment available for us at the time of the winter field work. It delivers 10cm accuracy (68 % confidence level) in real time in horizontal and vertical position when connected to a VRS network. We will clarify this in the revised manuscript.**

● For validation, several different reference datasets were used. In the snow depth error analysis, it's unclear whether you can isolate the error from the HS_platform from the error in the "spatial ground truth". Do you assume that HS_ALS(Summer) has 0 error? The lidar vendor documentation/metadata should include some assessment of product error (likely ~1-10 cm). This needs to be accounted for. I think what we really care about here is the accuracy for each HS_platform and whether we can, for example, combine two Pleiades DSMs to independently measure snow depth. If you can isolate platform-specific error (not integrating an external reference DSM), that would be really valuable, though I realize this is challenging.

**We do not assume that the UAS summer flight or the ALS scan are error free. However, these errors cancel each other out within the different comparisons (2+3) since we always compare snow depth maps derived with the same underlying summer reference. We do not believe that it is possible with these data to explicitly limit the platform-specific error when estimating snow depths. Because the quality and the errors of a winter DSM depends on many factors which are not platform specific as the lightning conditions, the experience of the operator, etc. Also, we only have a summer DSM for the UAS. Therefore, we cannot calculate platform specific snow depth maps.**

● Using a 2*STD filter is not necessarily more rigorous. More aggressive, yes. With this filter, you're inevitably classifying many "valid" points as outliers and removing. This filter will always reduce your apparent error, and some might perceive this "cleaning" followed by lower reported error as a bit dishonest. Since you're also defining robust metrics of error - median for bias, NMAD for spread, and using quantile/percentile values (25, 50, 75%) for the boxplots, you should be able to rely on those, rather than these "cleaned" values. I would avoid calling them "cleaned" - you can use "filtered". There is value in presenting and working with "filtered" maps in figures, but try to use the robust descriptive stats in the text/tables.

**We will replace the word "cleaned" with "filtered". While it is true that the 2*STD filter can remove "valid" points, we believe the principle of removing outliers and blunders is legitimate, in particular when comparing with manual snow depth since the number of samples reduces the robustness of**

**metrics to outliers. The reviewer himself is a co-author in Deschamps-Berger et al. (2020) where all snow depth less <1m and >30m are simply set to nodata, and not taken into account in the resulting statistics. We also present results before and after filtering for completeness, and therefore do not accept the comment that our approach and results may be associated with any form of dishonesty. As the reviewer notes, some of our metrics are robust but the 2\*STD filter which is supported by little changes before and after filtering. Nonetheless, the filter allows the mean, RMSE and STD to be more indicative of the performance of the majority of data when stereo-matching would have performed satisfactorily.**

● Using Agisoft's "aggressive" filtering will remove many of the largest outliers. ASP also uses an outlier-removal filter during gridding of triangulated points to produce a DSM (point2dem utility removes points with triangulation error > 3\*75th-percentile). If you had used different filter settings during DSM generation, your 2\*STD filter would produce different results.

**The inherent variability of stereo-matching makes photogrammetric restitution exposed and sensitive to large outliers, and some form of point cloud filtering is a necessity. Agisoft proposes several options but recommends the aggressive filter in normal cases when there is little vegetation. We have tested the different filtering methods and have found that aggressive filtering was more suitable for snow on steep terrain due to the challenging contrast and parallax, which makes it more prone to erroneous stereo-matches. We will clarify this in the revised manuscript.**

**We are aware of the filtering by point2dem but as indicated in Section 4.2, our filtering of the blended DEM is based on removing all points whereby the standard error of the blended elevation is less than 0.5 m (one pixel). We agree that any pre-processing choices designed to eliminate bad stereo matches will propagate to the final performance assessment, but not only to the 2\*STD filter as the reviewer seems to suggest. Therefore, those choices do not negate the merit of reporting metrics about snow depth with and without the 2\*STD on HS.**

● Do your errors actually have a normal distribution? Looking at box plots in figure 9, I would say no. So some of these assumptions about 2\*STD and percentiles may not hold. Showing histograms (with some transparency) of delta_HS from the different sources on a single plot might be more informative than the box plots.

**We think that the boxplot is a useful graph to see the dispersion of the errors, which we calculated for all three comparisons. The histograms show us that the errors are normally distributed except for comparison 1. For comparison 1 there are too few measurements, so we will add a single value plot instead of the histogram to the paper. For comparison 2+3 we will add the histograms. The new figures for comparison 1, 2 and 3 are displayed below.**

**Comparison 1: manual reference**

[Figure]

**Comparison 2: spatially dense UAS reference**

[Figure]

[Figure]

Comparison 3:
Snow depth maps of the entire Dischma valley

● I don't think the variable "HS" was ever defined in the text or captions, though it is used in several figures and tables.

**Thanks for pointing this out. We will define HS in the introduction when we define snow depth.**

● The results section was a bit hard to follow at times - dense presentation of numbers and statistics. The writing could use some additional proofreading as well - hopefully one of the more senior authors can help with this. There was a lot of interpretation and discussion presented in the results section. Some of the more speculative interpretations should probably be moved to the discussion section.

**We will improve the result section and move the speculative interpretations to the discussion section. A native English speaker will review the manuscript again carefully before final submission.**

● Rather than computing a snow depth for 4 DSM inputs against the same summer reference DEM (with it's own spatially variable error), and then doing an accuracy analysis of the snow depth rasters, you might perform your analysis directly on the DSMs. If you trust the DSM_eBee (winter), that can be your reference. Differencing each input DSM against this reference will allow you to characterize the error of the input DSM, without any additional error introduced by the summer reference or the resampling process. Alternatively, you could compute a per-pixel median grid from the input DSM values, and then difference each input against that median, though this works better with n >> 4

**Snow depth maps are the relevant end product for us. Clearly, resampling distorts the snow depth maps. But in order to compare two DSMs we still have to resample at least one DSM. So, technically it makes only a small difference whether you adapt two DSMs to the same other DSM or one DSM to the other. Therefore, the way we make the comparisons, the summer reference does not introduce any errors. Thank you very much for your idea with the median grid. Since the paper is already very extensive and we also find our comparison strategy important, we will keep this idea in mind for a future comparison.**

● I'm not used to seeing the "mean bias error" (MBE) and "median of the bias errors" (MABE). You are calculating these as the mean and median of the signed (not absolute) error values, right? Both will capture any bias in the data. Also, should be definition of MABE be "median absolute bias error"? The

equation in Table 3 does not indicate that absolute values were used. You're missing a "model" superscript on m_BE in the NMAD definition in Table 3.

**The reviewer is right that MBE and MABE are calculated from the signed error. We use the mean bias error (MBE) and the median of the bias errors to measure the bias of the DSM or snow depth against a reference. We will change the abbreviation MABE for the median of the bias errors into MdBE to avoid confusion with mean absolute bias error or median absolute bias error.**

**We will add the missing model superscript.**

● I'm not sure that the "comparison 1" is representative of true accuracy of the snow depth products, as the sample size of the reference manual/pole measurements is pretty limited with n=4-37, and the spatial distribution of the "ground-truth" is not ideal, with manual samples clustered along the valley bottom, where gridded snow depth values show stronger spatial autocorrelation. The ~11 usable pole samples on valley walls have good spatial distribution. I expect that if you isolated the two "ground-truth" sources (manual vs pole), the resulting accuracy statistics would differ considerably. My guess is the apparent error for manual points will be small, but you'll see much larger errors for the ~11 pole samples.

**We thank you very much for the hint. An error has crept into the figure and there are only 10 visible snow poles as described in the text. We will adapt the figure accordingly.**

**We agree that a more scattered meshed network of manual measurements would have been desirable but probing snow in such environment and limited time across a large area is extremely challenging and often dictated by practical consideration that constrain the spatial distribution. In future campaigns we could try to organize a much larger team for manual measurements. We could also try to either make an ALS flight or to make additional reference measurements with an ALS drone. This was not possible for this study. However, Comparison 1 is important and allows for the comparison with independent non-photogrammetric measurements. It confirms that the eBee+ measurements can be used as reference measurements for further comparisons.**

**Below you will find an extended version of Table 4 where the accuracy measures are also calculated for the manual measurements and the snow poles measurements separately. We will add this table to the supplements. The table shows that the snow pole measurements are generally better except for the Pleiades data. Despite the small sample size, we find that calculating the accuracy measures of the manual and snow poles measurements is meaningful.**

|  | Satellite | | | | Airplane | | | |
|---|---|---|---|---|---|---|---|---|
|  | Manual | Snow poles | All data | Filtered | Manual | Snow poles | All data | Filtered |
| RMSE [m] | 0.5 | 1.51 | 0.90 | 0.52 | 0.21 | 0.18 | 0.20 | 0.17 |
| MBE [m] | -0.34 | -0.79 | -0.46 | -0.35 | 0.04 | 0.0 | 0.03 | 0.01 |
| STD [m] | 0.37 | 1.29 | 0.77 | 0.39 | 0.2 | 0.18 | 0.20 | 0.17 |
| MdBE [m] | -0.33 | -0.44 | -0.40 | -0.36 | 0.01 | -0.07 | -0.03 | -0.04 |
| NMAD [m] | 0.41 | 0.71 | 0.44 | 0.47 | 0.21 | 0.09 | 0.17 | 0.17 |

| Number of measurements | 27 | 10 | 37 | 36 | 20 | 7 | 27 | 26 |
|---|---|---|---|---|---|---|---|---|

| | UAS | | | | Terrestrial | |
|---|---|---|---|---|---|
| | Manual | Snow poles | All data | Filtered | Snow poles |
| RMSE [m] | 0.23 | 0.13 | 0.21 | 0.16 | 0.54 |
| MBE [m] | -0.09 | -0.03 | -0.07 | -0.09 | 0.34 |
| STD [m] | 0.21 | 0.13 | 0.20 | 0.13 | 0.42 |
| MdBE [m] | -0.1 | -0.03 | -0.07 | -0.07 | 0.35 |
| NMAD [m] | 0.14 | 0.08 | 0.14 | 0.12 | 0.51 |
| Number of measurements | 27 | 10 | 37 | 34 | 4 |

● If you have maps of vegetation (perhaps from differencing a lidar DSM and DTM, though I think your lidar was unclassified?), you could produce a vegetation mask. This could be used to analyze your HS product accuracy only over pixels that are snow, which will allow you to test whether most of your negative snow depth values are from vegetation (primary hypothesis presented), or due to some other issue (bias, noise in DSM, etc). This is important b/c your HS maps don't cover the same extent, so some are preferentially sampling vegetated areas, and may appear to have more error as a result.

**We agree that this is a good approach to classify vegetation errors in snow depth maps, however, extends beyond the scope of this paper. Also, as mentioned above, we indirectly compare winter DSMs in these comparisons and not snow depth maps except for comparison 1. Vegetation induces negative errors in snow depth maps but these negative values are the same in every snow depth map. Nevertheless, we are also convinced that a pure vegetation mask does not solve the vegetation error problem of a photogrammetric DSM.**

● Figure 7 shows some of the detailed snow depth products, which is really valuable. It shows the noise in the Pleaides HS product compared to the ultracam and eBee HS products. But the focus is on artifacts in all HS maps due to vegetation. It is clear that the source for these artifacts is the summer ALS DSM product that was subtracted from each of the 4 photogrammetry DSMs provided. It might be valuable to show a panel with a color shaded relief map of your summer ALS DSM and a snow-free orthoimage showing this vegetation. Could also show similar panels for a snow-on DSM and orthoimage. This is tied to the suggestion of providing an evaluation of the DSM products, rather than the derivative HS products.

**We thank you for the input. In Figure 7, the snow depth maps were calculated using the eBee+ flight. We will clarify this in the figure caption (see the comment above). We will adapt the figure as shown below.**

[Figure]

**Figure 1: Extract of the Schürlialp snow depth maps shown in** Fehler! Verweisquelle konnte nicht gefunden werden. **with the scale ranging from -3 m to 3 m to illustrate the negative snow depths caused by the vegetation. In this extract the dark red spots are mainly bushes of the species *alnus alnobetula* compressed by the snow. Alnus Alnobetula is visible on every snow depth map and on the summer orthophoto and the hillshade of the ALS scan (panel (a) satellite, panel (b) airplane, panel (c) UAS, panel (d) terrestrial, panel (e) orthophoto of the UAS summer flight, panel (f) the hillsade of the ALS scan). This extract also shows how detailed a photogrammetric snow depth map can be. In the gullies the snow depths reach up to 5m. (Swiss Map Raster© 2019 swisstopo (5 704 000 000), reproduced by permission of swisstopo (JA100118))**

● Can you add some discussion of the maximum snow depths that you're seeing in the gullies? The scales are cut off at 3 m, and I expect that you could have real snow depth of >5 m locally. Your products capture this detail really well, which is a major selling point!

**Yes, we will add a discussion point about the maximum snow depth in the gullies and outline it as important point of our products. We will include the values of the gullies in the caption of figure 7 and in the result section of comparison 2.**

● It's important to remember (and state) that snow depth maps are just a specific application of a DEM difference map. There are many studies out there offering accuracy analysis of the latter, and it's important to consider these. Photogrammetric DEMs can also suffer from elevation- and/or slope-dependent errors (e.g., Shean et al., 2016). It may be worth considering how this affects your results.

**Yes, we agree with you there. We will extend the discussion to clarify this point.**

● There are several figures/tables that could be moved to a supplement, which would help reduce length and improve flow

**Yes, that is a good point. We will evaluate this and move everything we can into the supplements.**

● In your Table 7, you present some qualitative advantages and disadvantages. You might consider breaking out into columns for coverage, acquisition time, cost, accuracy, resolution, repeat, and other. One thing that is missing from this is the "processing time," in addition to acquisition time. It might be useful to mention somewhere the time required to process each dataset (manual interaction and compute), software costs, equipment costs, etc. At the end of the day, these are often the deciding factors...

**We agree with you that coverage, acquisition time, costs, accuracy, resolution, repetition are important criteria. But these are very difficult to capture in concrete terms. For example, we have not achieved the maximum possible coverage. Also, the processing time is very difficult to capture, because it depends very much on the computer used and this can change very quickly. Costs are also difficult to define, as they can vary greatly depending on location/market and are in constant development. So, the aim of this table is to summarize what each sensor can be used for.**

Specific Comments

Rather than "RMSEs" and "NMADs", I recommend "RMSE and NMAD values are…"

**We will adapt this.**

25: "too few"

**We will correct this.**

25: Based on abstract, unclear why ground-based obs were used with eBee but not with snow pits; I think the issue is that the ground-based obs don't intersect with "manual and snow pole measurements".

**We will clarify this in the abstract. Indeed, the issue is that the terrestrial measurements don't intersect with the manual measurements, only with 4 snow pole measurements.**

29: specify "more than two photogrammetry platforms"

**We will add "photogrammetry" in the sentence.**

30: replace "the specific advantages and disadvantages of them" with "their specific advantages and disadvantages"

**We will replace this.**

18: Specify the sensors or methods used for "manual" or "AWS" - e.g., probing, GPR, sonic ranging, etc.; Also, point measurements themselves are not the problem - the issue is their sparse spatial density, especially over large regions. One can probe every 10 cm on a relatively small grid and get lots of valuable information about the local spatial distribution of snow depth.

**We will specify the sensors and methods used for "manual" or "AWS". We will also update the text to be more precise on the issue of point measurements and their sparse spatial density over large regions.**

27: Should include Painter et al reference for ASO: https://doi.org/10.1016/j.rse.2016.06.018

**We will include Painter et al., 2016.**

**Painter, T. H., Berisford, D. F., Boardman, J. W., Bormann, K. J., Deems, J. S., Gehrke, F., Hedrick, A., Joyce, M., Laidlaw, R., Marks, D., Mattmann, C., McGurk, B., Ramirez, P., Richardson, M., Skiles, S. M., Seidel, F. C., and Winstral, A.: The Airborne Snow Observatory: Fusion of scanning lidar, imaging spectrometer, and physically-based modeling for mapping snow water equivalent and snow albedo, Remote Sensing of Environment, 184, 139-152, 10.1016/j.rse.2016.06.018, 2016.**

31: What kind of accuracy? Is this position accuracy, and 0.1 in horizontal and vertical? Or snow depth? Several important factors for TLS, so careful with a blanket statement like this. See also study by Currier et al (2019): https://dx.doi.org/10.1029/2018WR024533

**We will specify this with a sentence as the following one: "Terrestrial laser scanning (TLS) can measure the distance between scanner position and snow surface with accuracies below 0.10 m beyond 1000 m."**

8: Would be valuable to mention other VHR satellite image options for snow depth mapping. See our preliminary publication on WorldView stereo snow depth: "Spatially extensive ground-penetrating radar snow depth observations during NASA's 2017 SnowEx campaign: Comparison with In situ, airborne, and satellite observations D McGrath, R Webb, D Shean, R Bonnell, HP Marshall… - Water Resources Research, 2019"

**We will add "WorldView-3 satellite-derived snow depths were calculated by McGrath et al., 2019 yielding RMSE value of 0.24 m in comparison to GPR measurements."**

**McGrath, D., Webb, R., Shean, D., Bonnell, R., Marshall, H. P., Painter, T. H., Molotch, N. P., Elder, K., Hiemstra, C., and Brucker, L.: Spatially Extensive Ground-Penetrating Radar Snow Depth Observations During NASA's 2017 SnowEx Campaign: Comparison With In Situ, Airborne, and Satellite Observations, Water Resources Research, 55, 10026-10036, 10.1029/2019wr024907, 2019.**

9: Should include medium-format (Phase One) camera SfM mapping: "Assessing the Ability of Structure From Motion to Map High-Resolution Snow Surface Elevations in Complex Terrain: A Case Study From Senator Beck Basin, CO J Meyer, SMK Skiles - Water Resources Research, 2019" Chris Larson's work in Alaska?

**We will add "Meyer and Skiles, 2019 produced DSMs from snow covered surfaces with the RGB camera installed on the lidar-based Airborne Snow Observatory and compared them to simultaneously collected lidar data. They found a NMAD of 0.17 m and a mean relative elevation difference of 0.014 m for a spatial resolution of 1 m."**

**Meyer, J., and Skiles, S. M.: Assessing the Ability of Structure From Motion to Map High-Resolution Snow Surface Elevations in Complex Terrain: A Case Study From Senator Beck Basin, CO, Water Resources Research, 55, 6596-6605, 10.1029/2018wr024518, 2019.**

23-24: Consider rewording to emphasize that few studies have evaluated these platforms for the specific application of snow depth mapping. Many past studies have done this for other surface types and scientific/mapping applications.

**We will reword this sentence as:** *Many studies have investigated the performance of photogrammetry for different surface types and mapping applications. However, few studies have examined the available photogrammetric platforms for their performance on snow (e.g., Bühler et al., 2017, Deschamps-Berger et al., 2020). Therefore, a comprehensive assessment is necessary to compare the snow depth products of terrestrial, UAS, aircraft and satellite platforms.*

**Bühler, Y., Adams, M. S., Stoffel, A., and Boesch, R.: Photogrammetric reconstruction of homogenous snow surfaces in alpine terrain applying near- infrared UAS imagery, International Journal of Remote Sensing, 38, 3135-3158, 10.1080/01431161.2016.1275060, 2017.**

**Deschamps-Berger, C., Gascoin, S., Berthier, E., Deems, J., Gutmann, E., Dehecq, A., Shean, D., and Dumont, M.: Snow depth mapping from stereo satellite imagery in mountainous terrain: evaluation using airborne laser-scanning data, The Cryosphere, 14, 2925-2940, 10.5194/tc-14-2925-2020, 2020b.**

26: Mention high-albedo surfaces (sensor saturation), limited surface texture for fresh snow (poor stereo correlation results)

**We will mention it:** *Each platform has its advantages and disadvantages, but each must be able to cope with the challenges of imaging alpine environments, including steep terrain and rapidly changing weather conditions, high-albedo surfaces (sensor saturation) and limited surface texture for fresh snow (poor stereo-correlation).*

28: UAS can offer cm-scale products; all depends on altitude

**We will add a comment:** *Also, depending on the flying altitude and camera resolution, UAS allow for decimeter-scale to cm-scale snow depth maps even when the accessibility of terrain is restricted i.e. because of avalanche danger and perform well in a winter alpine environment.*

31: Other issues beyond ownership can prevent flying whenever required - regulations, certified pilot availability, weather

**We will add:** *However, factors that may prevent UAS from flying are regulations, availability of certified pilots, harsh alpine weather, etc...*

35: "scales"

**We corrected this spelling mistake.**

19: "surface slope"

**We will replace "slope angle" by "surface slope".**

20: "contrasted snow depth distribution" is a bit unclear - I think you're saying that downslope winds from gullies lead to snow deposition at the base of the gullies?

**We will adapt the sentence to be clearer:** *Interesting features of the Schürlialp area are gullies that channel downslope winds and produce snow deposits on the bottom of the gullies.*

24: "Data… provide" (singular, not plural)

**We will correct this.**

24: not just "snowfall" though, as you have wind redistribution and melt, I think "snow depth evolution" might be better term

**Indeed, "snow depth evolution" is better suited and we will change this.**

25: "it" - I think you mean the snow stations?

**We clarified "it". We mean the data of the snow measurement stations.**

28-29: I agree that things didn't change much, but if you consider the observed change as a percentage, then 20 cm of a ~65-70 cm snowpack is pretty substantial! You might consider framing these observed changes as a percentage of your expected measurement error. For example, 20 cm is well within the Pleiades snow depth accuracy, but much larger than the UAS snow depth accuracy.

**We agree with your objections. However, these 20 cm are only to be found in the lower snow measuring stations which are lower than the lowest altitude of the Schürlialp test site (5DF: 1560 m a.s.l. and 5MA: 1655 m a.s.l.). Also, these are point measurements which are very local and also have errors (2-3 cm for well-maintained automatic stations and 2-4 cm for manual observer measurements). We think that here the real measured values are more useful than percentages.**

10: eBee+ RTK is a "fixed-wing" - should mention this, as many readers may think "drone =quadcopter"

**We will add the term "fixed-wing" in the sentence.**

13: "triangulation" is not the word I would use here, as it could be confused with stereo triangulation to produce a point cloud. I think you mean consistent geolocation.

**We mean triangulation as in the process of block triangulation or aerial triangulation. This refers to the bundle bloc adjustment, the process that needs ground control. Before computer vision used this term to refer to refer to the process of intersection and restitution, it was defined and unambiguous in photogrammetric science (see Granshaw, S. 2016. Photogrammetric Terminology: Third Edition, The Photogrammetric Record 31(154):210-252 DOI: 10.1111/phor.12146).**

17: OK, so these are visible in the Ultracam and eBee imagery, but probably not visible in Pleiades?

**Yes, the additional points are not visible on the Pleiades imagery.**

19: "positioned" to "The positions of all GCPs were determined using…"

**We will change the wording of the sentence as suggested.**

2: "Pleaides-1B stereo image triplet"

**We will add "stereo".**

7-9: I would also recommend providing the combined off-nadir angles for each image, rather than the signed across and along-track components. I think your B/H numbers are just accounting for along-track off-nadir angles? If possible, good to provide convergence angle for each pair in addition to B/H.

**We will add "Along-track incidence angles of -16.3°, 7.6° and 17.9° resulted in combined incidence angles of 23.2°, 14.2° and 20.1°, and three stereo pairs with Base-over-Height ratio…"**

**Our B/H ratio account for both along and across track angles.**

10: Given your scene relief, this is probably acceptable

**Here we are echoing the specifications provided by ASTRIUM. The hypothesis that it is acceptable is unspecific as it would need to define "acceptable". In fact, our result show that in our study, this B/H combines with low contrast to challenge stereo matching and yield significantly more noise in the restitution of poorly textured snow surfaces.**

15: Provide time zone for times (and all other instances in paper)

**We will provide that.**

16: Could delete sentence about not acquiring on same day, as we don't know what the technical issues were

**Since the flight was planned for the same day, but actually could not be carried out due to technical problems on the plane, we think this is important to mention. But we will evaluate if we can delete it.**

20 and Table 1: would be nice to provide sensor dimensions in addition to total pixel count

**We will add the sensor dimensions.**

21: "Large-format CCD"

**We will add this.**

22: focal length in Table 1 is 122.7.

**Yes, that's the exact focal length and we adapted it. For promoting purposes vexcel is using 120mm that's why there was this value in the text first.**

24: "I" is near-infrared? Recommend consistency with terminology used for Pleiades section

**RBGI is the short term which Vexcel Imaging GmbH uses to tag the images. I is NIR infrared. We will be consistent with the terminology and use NIR.**

28-29: OK, so three flights total? Better to state that, rather than changing the battery twice.

**Good point, we will change this.**

Page 8:

7: time format 10.37 vs. 10:37, and time zone

**We will correct this.**

7: What is imaging interval? 1 second, 1 minute? Ah, it wasn't clear that this was a terrestrial photogrammetry survey! I thought you had installed time-lapse stereo cameras. Maybe state this earlier in the section, before you get into details on variable GSD. Could move lines 16-22 or at least 16-19 here.

**We will be more precise with our description and move the lines 16-19.**

11: That sounds really large for a convergence angle, and earlier you said B/H of 0.25-0.42 is acceptable? Why different here?

**A parallactic angle of 90 degrees is optimum to minimize error propagation in ray intersection. This would correspond to a B/H=2. In practice, such B/H requires practical considerations such as potential for obstruction and overlap. For example, vertical aerial photogrammetry (not converging) could not achieve this unless by using relatively short focal (large view angle) at the cost of potential distortion and increased GSD. Tradeoffs are made and B/H ratio of 0.6 have long been the gold standard as it still provided good intersection accuracy, while maintaining optimal overlap. We mentioned B/H of 0.25 to 0.4 as the recommendation from ASTRIUM with respect to stereo acquisition by Pleiades. Again, those refer to practical considerations of acquisition and expected general accuracy of the products.**

19: Really cool to see this setup! :) I used a similar rig for terrestrial and oblique aerial surveys in Greenland and Pacific NW: https://dshean.github.io/technology/sfm/ (Fig 4), back before consumer UAS and integrated cameras were up to the task

**Thank you!**

12: Hmm. I think you can do better than 5 cm on the pole shown in fig 3b. Probably down to 1-2 cm precision. Is the inaccurate reading issue also related to the depression in the snow surface immediately around the pole? If so, state this. For future work, you might consider a small quadcopter that you can fly to "inspect" the poles from closer range to make more precise readings.

**The idea of the quadcopter is interesting and we thank you for the idea. Also, it was rather difficult to quickly locate the snow poles with binoculars or a camera. If one had a quadcopter, one could program the positions and could record more accurate measurements given the proximity to the snow poles.**

**The 5cm are on the one hand because of the small depression or accumulation around the snow pole and on the other hand because of the possible skew of the snow pole. It is possible that the snow pole is slightly inclined because of the pressure of the snow. We will make a corresponding comment.**

4-6: provide date range for ALS survey. I see August 5-6, 2015?

**Exactly, the date range was 5-6 August 2015 and we will add it in the text.**

9: Provide citation for LAStools. Martin is pretty clear about this, especially if you're using an unlicensed version :)

**We will add the citation as Martin is requesting (https://groups.google.com/g/lastools/c/ximEoUEbIl4?pli=1).**

16: "NASA Ames Stereo Pipeline (ASP, Shean et al., 2016, Beyer et al., 2018) version 2.6.2 (corresponding Zenodo DOI)" - please follow latest citation instructions https://github.com/NeoGeographyToolkit/StereoPipeline#citation . Please include Shean et al. 2016, as you are using core ASP functionality for processing Earth observation imagery that was implemented during that effort.

**We will modify accordingly and add those citations.**

**Beyer, Ross A., Oleg Alexandrov, and Scott McMichael. 2018. The Ames Stereo Pipeline: NASA's open source software for deriving and processing terrain data, Earth and Space Science, 5. https://doi.org/10.1029/2018EA000409.**

**Shean, D. E., O. Alexandrov, Z. Moratto, B. E. Smith, I. R. Joughin, C. C. Porter, Morin, P. J. 2016. An automated, open-source pipeline for mass production of digital elevation models (DEMs) from very high-resolution commercial stereo satellite imagery. ISPRS Journal of Photogrammetry and Remote Sensing, 116. https://doi.org/10.1016/j.isprsjprs.2016.03.012.**

**Ross Beyer, Oleg Alexandrov, & ScottMcMichael. (2019, June 17). NeoGeographyToolkit/StereoPipeline: ASP 2.6.2 (Version v2.6.2). Zenodo. http://doi.org/10.5281/zenodo.3247734**

16: not clear what you mean by "GDAL (for satellite data)" - for orthorectification? Resampling?

**We use GDAL in many steps of our processing before and after ASP. That includes resampling, orthorectification, reprojection, grid calculation.**

17: For reference, you can also use the ASP geodiff command or generic gdal_calc.py for simple raster operations outside of ArcGIS.

**We are aware of those and often rely on gdal_calc.py for this. In this study, most of the analysis was performed in ArcGIS Pro.**

17: No mention of co-registration before subtraction here? I assume this will be discussed elsewhere…

**Yes, this is addressed in the description of workflows to achieve consistent geo-referencing between products**

Page 12:

2: Suggest "analyzing" rather than "using"

**We will change the word to "Analyzing".**

9-16: I didn't completely follow all of this, but it sounds like something you clearly thought about carefully :). Any coordinate system transformation of raster data will require some resampling/interpolation, it's a question of error introduced by the transformation. If you can provide some sense of this error, that would be valuable. It's likely several cm. I initially thought the LV03/LN02

were from 2003/2002, but just checked and it was 1903/1902! Good call to convert to LHN95. I would think the lidar vendor should update their products…

**Working with various and independent data products in Switzerland coordinate systems is indeed very interesting and potentially a source of errors if not handled carefully. It is true that resampling and height adjustments can add errors. We handle all datum and height transformation using the latest official grids (REFRAME library from swisstopo). Schlatter and Marti, 2005 indicate that the difference between LN02 and LHN95 is modelled with an accuracy of 1 to 10 cm. We will add this in the revised manuscript.**

**Schlatter, A., and Marti, U.: Höhentransformation zwischen LHN95 und den Gebrauchshöhen LN02, Geomatik Schweiz: Geoinformation und Landmanagement = Géomatique Suisse: géoinformation et gestion du territoire = Geomatica Svizzera : geoinformazione e gestione del territorio, 103, 10.5169/seals-236251, 2005.**

20: Already mentioned version number and citation earlier, don't need to repeat here.

**We will delete version number and citation.**

21: Should state you're using the RPC camera model first, then talk about updating during bundle adjustment

**We will add at the start of the sentence:** *Processing of Pléiades satellite images involved triangulation and refinement of Rational Polynomial Coefficient (RPC) in ERDAS …*

22/30: Should avoid starting sentence with "14" and acronyms "DSMs"

**We will spell out "Fourteen" and "Digital Surface Models" at the start of sentence.**

27: Is it expected that the CE90 and LE90 are identical? I would expect a factor of ~2 degradation in LE90.

**We doubled checked our LOOCV and those values are correct and indeed identical. Airbus specifies its Elevation1 DSM product with GCPs achieving CE90 = LE90 = 1.50 m (see https://www.intelligence-airbusds.com/files/pmedia/public/r49249_9_elevation1_dsm.pdf).**

Page 13:

2: See general comment about the citation here and question about blending. I wouldn't say "with GDAL". This is not a standard GDAL command line utility or API function. I think you created a custom script that would weight values from your 3 input DSMs based on the ASP triangulation error map? I recommend you consider the ASP dem_mosaic weighted average blending approach.

**We answered the general comment and clarified the arithmetic used to do this. We use a single gdal_calc.py instance to implement the weighted mean, indeed fed by DSM and error maps, so our statement is factually correct. As far as we know, dem_mosaic in ASP does not allow the weighting of input DEMs in such a way.**

9-15: This "tilt" is a bit troubling, and potentially indicates issues with the GCPs, either their spatial distribution and/or recorded position errors. See general comment.

**We answered this comment in details earlier.**

13-15: The last sentence should be moved to the end of the above paragraph (line 7).

**We will move the sentence.**

19: Note that Agisoft can process grayscale images, and RPC models these days.

**We appreciate the suggestion and will certainly consider it in the future.**

29: OK, so refining interior camera parameters is not desirable, so you explicitly disabled from the initial alignment solution? Did you add calibrated lens distortion coefficients, or did you allow Agisoft to solve for these? If so, which coefficients? Provide more detail here.

**The Ultracam Eagle M3 is a metric camera precalibrated with no detectable lens distortion as per its calibration report. We believe it would be wrong in such case to use any form of self-calibration (+ subsequent optimization in agisoft), as this would only involve a mathematical optimization on top of the BBA, that would push any departure from the true calibrated values into geometrical distortions of the DSM. So, indeed we set the focal length to calibrated value as well as the principal point coordinates and all lens distortion parameters to 0.**

29: Careful about interchanging GCP and CP here. I don't think you have 29 GCPs, and if you're using points to control geolocation, they are GCPs, not check points.

**We really have 29 GCPs and 15 additional CPs. We are not reusing GCPs.**

2: I'm really glad that you used a factor of 2x here, and did not export an oversampled DSM at the native image GSD! This is a common issue.

**Thank you.**

5: I think this was done using the SenseFly eMotion software? This first paragraph is a bit confusing, as you're already talking about the accuracy of the DSM but you haven't told us how the DSM was produced. I recommend you move this paragraph later, and integrate with the last sentence of paragraph 2 (lines 21-22), where you actually report the observed RMSE.

**We used SenseFly eMotion software only for flight planning and downloading the images from the eBee+. Here we refer to the processing strategy in Agisoft (Integrated Sensor Orientation) with corresponding citations for context, including what sort of accuracy can be achieved with ISO processing. We will move the paragraph as you suggest.**

If your base station position is accurate (I think you used NTRIP caster for real-time corrections, not a local base), then you should be able to achieve the accuracy you report without GCPs.

**This is correct, we used a NTRIP caster for real-time corrections and did not use GCP to process eBee data. Therefore it is true that we expect to achieve the accuracy we report in the context paragraph starting this section.**

15: "EXIF metadata"

**We will change this.**

32: dGNSS - check for consistency in terminology used elsewhere

**We will check for consistency and use DGNSS.**

1-2: I think you mean it was not possible to determine the precise offset of the GNSS antenna phase center relative to the center of the camera detector. Could probably estimate vertical offset and constrain a bit better than 0.2 m horizontal and vertical, but not a huge issue.

**Exactly, that's what we meant. We will clarify the sentence.**

3-5: What you did here should work. Technically, you want to use a consistent grid for all output products, with same grid cell size, projection and origin. If the spatial extent of the two rasters varies significantly, then you can use something like GDAL's -tap option to force output grid extent to use whole integer multiples of the output grid cell size. Can also do this manually in Agisoft when specifying output extent - round left, top, right and bottom bounds to nearest multiple of the cell size. A shared raster origin (upper left coordinates) or this -tap approach avoids the need for a second resampling step, which is going to further degrade resolving power of your products and can introduce additional error.

**Thank you for the input!**

7-9: I like this approach. Approximately how many pixels were sampled within this circle?

**We sampled around 190 pixel for the eBee, around 126 for the ultracam and the ground-based imagery.**

10: So this sounds like nearest neighbor sampling for Pleiades snow depth grid. I recommend bilinear or cubic for point sampling like this, esp if you see relatively large pixel-to-pixel variability in snow depth grid values near the sites.

**We used the ArcGIS Pro function "Extract Values to points". We didn't interpolate because we are interested in the real value of the cells. The product of the Pleiades is already an interpolated product with a much larger cell size than the buffer. We are aware that the nearest neighbor can cause obvious errors in large pixel differences, but when interpolated, the trueness of the values is not guaranteed either. For this reason, we prefer this method.**

Page 18:

12: See general comment about the 2*STD filter

**We responded to this comment earlier.**

12-13: Do your errors actually have a normal distribution? Looking at box plots in figure 9, I would say no. So some of these assumptions may not hold. Overlapping histograms of delta_HS from the 3 sources might be more informative than the box plot.

**We will add histograms to the boxplot figures which show that it's nearly normally distributed. The updated figure was provided in the general comment above.**

Page 19:

3-5: recommend using "area" for all instances instead of "surface"

**We will change all instances to "area".**

6: already said ultracam only covers northern part of valley due to clouds elsewhere (which is really too bad, data quality over cloud-free areas looks great). Does "Dischma valley" = "Dischmatal"?

**We will remove unnecessary repetitions. Yes, "Dischma valley" is "Dischmatal". We changed it to "Dischma valley" as this is the term we use here.**

6-7: what does "good quality of eBee+ flight" mean, and how does an orthoimage illustrate this?

**We agree, "good quality of eBee+ flight" can mean different things. We will change the sentence to be more meaningful.**

Page 21:

3: "graphically" sounds strange to me, I think you mean "qualitatively"

**We will change the word to "qualitatively".**

7-8: "errors introduced during photogrammetric processing" - the processing itself is not responsible for the negative values

**We will change the wording of the sentence.**

9: OK, yes, using a DSM is a problem, but the acquisition date of your summer reference DSM will also be very important in terms of the vegetation growth cycle, leaf-out, etc. Also, you are assuming that your photogrametrically derived DSM is capturing the true top of canopy, which is not necessarily the case (esp with "aggressive" filtering that will remove isolated shrubs/trees)

**We agree with you that the acquisition of the summer DSM adds uncertainties. We will investigate how to correct a photogrammetric summer DSM for photogrammetric snow depth maps depending on vegetation and recording date in further studies.**

9 and 14: recommend "depressed" or "compressed" rather than "pressed down"

**We will change the word to "compressed".**

17-22: What is the total sample size for manual and snow pole measurements again? I would argue that some of your larger apparent error is due to limited sample size, and likely the nearest neighbor sampling approach for the coarser Pleaides snow depth grid.

**The total size of the manual and snow pole measurements is 37. Yes, we agree some error is due to the limited sample size and the coarser Pleiades snow depth grid. We will a comment in the discussion.**

21-23: Careful with this kind of statement, as it sounds like subjective error reporting…

**We will change the wording of the sentence.**

23: "decreased" instead of "deteriorated"

**We will change the word to decreased.**

23: The NMAD and median should not be strongly influenced to one outlier, unless your sample size is an issue.

**Yes, the sample size is an issue in comparison 1 and that's why the NMAD and the median is strongly influenced by the large outlier. We will improve this point as mentioned above in the discussion.**

25: Again, triangulation here is a bit ambiguous, as it could be confused with the triangulation to produce the point cloud. I think you're talking about the integration of GCPs during the bundle adjustment routine.

**As explained above, this is what we mean as per the general terminology in photogrammetric science, not in computer vision.**

27-29: This information (about the roads used to correct Pleaides DSM) should be moved to the methods section.

**This was indicated clearly in section 4.2, only restated here.**

29: I think you're trying to say that one sample Pleaides triplet is not necessarily representative of the capabilities of the sensor.

**Not really, and we think your interpretation may be influenced by your definition of "triangulation". We are discussing the fact that the tilt and its effect on determining snow depth is a limitation having to go through independent block adjustment (triangulation), not a limitation of the sensor and its inherent performance.**

30: "…approach for snow-covered images" - this problem will be mitigated with snow-free conditions

**We will add "for snow-covered images".**

30-31: Not sure this last sentence is necessary

**We think it's important to remind the readers that Pléiades images were recorded from space because this level of detail is quite an achievement. But we will include this point into the discussion and remove it from the result section.**

Page 26:

10: recommend changing "data analysis" to "accuracy analysis of comparison 1 is potentially not representative of the true accuracy of the snow depth products" or something along those lines. See general comment on this issue.

**We will change the wording of this sentence.**

Page 27:

1: Note that this is correlation of per-pixel values, and should not be confused with spatial correlation

**We will specify this in the paper.**

6-7: Might be useful to report the percentage of data removed with your 0-5 m filter here. Based on Figure 8, this should be minimal for ultracam, but you're removing a nontrivial sample of the Pleiades HS values.

**We will report the percentage of the removed values. Yes, it's a nontrivial part of the Pleiades HS values and we will discuss that in the discussion.**

8: Don't start sentence with "0 m…"

**We will change the sentence.**

11: "R^2 values"

Page 29:

3: "imagery" should be "DSMs"

**We will change the word.**

4: You can only use the ultracam HS as ground truth to evaluate the Pleiades HS, right? The way it is stated, it sounds like you're using to evaluate both.

**We will change the sentence.**

Page 32:

8: There are also studies by Shaw et al. (2019?). Please review Simon Gascoin and Etienne Berthier's recent publications, as they are listed on other recent papers using Pleiades for accuracy of DEM difference maps, including snow depth

**We will review the recent publications identified and adapt them to this section.**

12: I don't think it's fair to present your "cleaned" values here

**We will change this.**

12: Again, rather than comparing DSM accuracy, you're comparing snow depth accuracy, which includes error from the reference dataset.

**We will adapt this in the paper so that it's clear that is an indirect comparison of the winter DSMs.**

13: "This is higher" here is a bit confusing, as your numbers are lower. Consider rephrasing. Also "this" is ambiguous.

**We will rephrase.**

14: Why is one pixel considered the maximum??? Sub-pixel correlation should be capable of 0.1-0.2 px accuracy. Is the issue with your manual GCP identification? Still should be able to achieve sub-pixel with this, esp with modern GCP markers.

**We agree with you. We will rephrase this sentence to account for sub-pixel accuracy.**

18-19: I don't follow the last sentence, but I may just be getting tired :) Consider rewording. What is "they"

**We will reword the sentence. With "they" we mean "The RMSE and the NMAD".**

21: Need to qualify this with "from Pleiades, without further correction". We have demonstrated better accuracy with WorldView-3 DEM difference (same as snow depth) products (see McGrath et al, 2019; Shean et al., 2016). And there are several systematic errors in the individual DSM products (e.g., CCD offsets, unmodeled attitude error ["jitter"], parallax issues in L1B camera models) that can be corrected to further improve accuracy. We are actively working on this, so hopefully Pleaides DSM accuracy can be improved!

**We will qualify this and add more information about accuracies of satellite products and errors, including in the most recent studies involving snow depth mapping.**

25: I would avoid using the world "profit" to avoid confusion with commercial applications, "benefit" would be better

**We will use the word "benefit".**

28-30: I disagree. ICP co-registration approaches can work very well, even when only sparse exposed surfaces are available. Co-registration using methods like Nuth and Kaab (2011) can also be used to take advantage of the slope and aspect-dependent dh - can then use limited snow-free surfaces for final vertical correction. I think you may be referring to cases when the entire scene is 100% snow-covered (rare for mountains).

**Thank you very much for the input and we agree with you. We will reword this accordingly.**

Page 33:

14: Issues of contrast can also be problematic in satellite imagery. In my experience, it is more about fresh snowfall and whether image GSD is fine enough to capture relevant length scales of surface roughness.

**We will add a sentence in the discussion section for the satellites (6.1).**

18: An aircraft outfitted with high-end GNSS and IMU should be able to provide very accurate camera position and orientation data, eliminating the need for GCPs (as with your eBee RTK results)

**At first glance, this should indeed be the case. But the accuracy of the Ultracam positions (0.2 m) is worse than the accuracy of the eBee+ (0.02 m for the position and 0.03 m for the height).**

19: Again "higher GSD" - careful here, as lower numeric value means better resolution

**We will adapt the sentence.**

24-25: This is consistent with my experience using eBee RTK platform. But careful about generalizing to all "UAS photogrammetry" - a DJI Phantom with no GCPs is not nearly as capable

**Thanks for this comment and we will consider how we phrase.**

Page 34:

8: This is where using a 3-4 m pole can be beneficial.

**Yes, but it's difficult to have a setup with a 3-4 m pole which is also mobile.**

14: "non-negligible"

**We will adapt the word.**

20: Careful with this - using a DTM may help in some areas, but not others; "bushes" is pretty generic, and a bush with leaves appears very different than a leaf-free bush to a laser or camera

**Thanks for this comment. We will consider this and discuss this accordingly.**

Figure 2:

Check TC date formatting standards - I initially read this as MM.DD If you're going to remake this figure, it might be better to alter the aspect ratio to reduce the width of the x axis, so we can better see the magnitude of the change over the study period (right now all lines look really flat). Either that or add thin horizontal gridlines and a complementary right axis label.

**We will change to the TC formatting standards and try to improve the readability of the figure.**

Figure 4:

I would move this to supplemental figure - it is awfully large for the information you're trying to convey, or could be presented as a table Need to define HS in caption

I'm still confused by comparison 3. Why is comparison 3 HS_platform_ALS?

**We created this flowchart to give an overview over the different comparison strategies. We find this figure necessary given how many data products are discussed and presented in the manuscript. We will define HS in the caption.**

**We find comparison 3 important because otherwise we won't have a comparison or an evaluation of the snow depth maps on larger area. Certainly, a snow depth map for the whole recorded area would be interesting as well as a winter ALS scan over the whole area. However, these data were not captured as part of this study, but we believe we have nonetheless sufficiently imaged an area large enough for a valuable comparison of snow depth distribution.**

Figure 5:

d) I don't see the violet stars? If they are present, recommend making larger and using a color that won't blend in with the orthoimage

**We will add the purple stars, they disappeared in the cycles of redesigning the figure.**

Figure 6:

The color ramp used here makes it very difficult to distinguish snow depth values between 0-1 m. It would be better to use a perceptually uniform, linear color ramp, ideally with labels for increments of 1.0 or 0.5 m intervals. I realize this may not be straightforward in ArcGIS.

**We have used our organization's standard color ramp for snow depth maps, which we feel is sufficient for depicting the overall snow depth distribution across the study site. This color ramp also highlights well the features of the snow depth maps. But we will add the figure shown below to the supplement where we masked all values smaller 0 and greater than 1. Also we changed the color ramp of figure 7 (see figure 7 in the general comments).**

[Figure]

Figure 7:

Maybe use "apparent snow depth" or "snow depth estimate" here and elsewhere in the paper, as it's not physically possible to have a negative snow depth

Why did color ramp change here to what looks like matplotlib plasma? Also, since you are showing values from -3 to 3 m, could be better to use a diverging color ramp. Lots of good resources on the theory behind these visualization approaches:
https://matplotlib.org/3.1.1/tutorials/colors/colormaps.html

**Yes, it's not possible to have a negative snow depth but we think is all right to use only "snow depth" because we do not start from the absolute truth.**

**We thank you a lot for the input with the color ramp. We adapted Figure 7 with a diverging color ramp (see general comments above.)**

Figure 8:

This is a nice figure, but why such a large bin size??? The quantization here makes comparison between different sources really challenging. I would recommend bin size of <5-10 cm, so we can properly assess each distribution. Also, it would be valuable to create a mask for the common intersection of valid HS pixels (ie pixels where all 4 DSMs have a valid elevation), clip each input HS to the same mask, then produce a similar histogram, maybe as a second panel in this figure. This provides an "apples to apples" comparison, as your current histograms are sampling different portions of the domain, and there is no reason to expect your reported mean and std statistics to be the same.

**We will change the bin size of the histogram to 0.1 m. Below you find the new graph with the bin size of 0.1 m. We agree with you but the nice thing to see in this graph, is that also for different sizes of snow depth maps the relative distribution of the snow depth is similar. Therefore, we will keep this graph the way it is.**

[Figure]

In the caption, you can just say "normalized histogram" without details about dividing by total number and multiplying by 100.

**We will make the caption more precise.**

What does "all not shown" mean? You have the Manual and snow poles measurement on histogram on the plot.

**The "all not shown" means that the mean and STD are not shown but this is misleading formulation. We will change this formulation.**

Figure 9:

Is "the median" the same as your MABE metric? I think MBE is just the mean error, right? 5th and 95th are not quartiles, they are percentiles. Fix in all other captions for box plots.

**Yes, the median is the same as the MABE. We will change this in the caption of all boxplot. MBE is just the mean bias error as defined in the section 4.6.4. We will clarify this point. We update the captions for the figures using the 5th and 95th percentiles.**

Figure 11:

I think this is a 2-D histogram showing density? So, not a scatterplot. I don't think your current caption says what color represents.

**It's a scatterplot showing the density. We will rework the caption and state what the color represents.**

Probably want to say "y = x"

**Yes, we will change this.**

Should mention in caption why the bottom row is limited to range of 0-5 m on y axis. I think this is labeled "cleaned" but you're adding another filter here.

**We will mention that in the caption and change the name "cleaned" to "filtered". We also will clarify this filtering in the method section for consistency.**

Table 1:

Time is local or UTC?

**It's local time, which we will state with LT.**

Add a row for sensor dimensions in pixels and/or mm. 450 MP is not really a "resolution" and we don't know dimensions.

**See comment above. We will add a row.**

What do you mean by Pleaides mean GSD of 0.7 (resampled to 0.5). Did you intentionally oversample, or are the L1B products delivered at higher res after some super-resolution processing beyond normal TDI?

**Pleiades native Ground Sampling Distance (nadir) are Panchromatic: 0.7m; Multispectral: 2.8m. the shipped products are resampled by ground segment to 0.5 and 2m, see section "B.2.2 Why 50 cm?" in Pleiades User Manual ([https://www.cscrs.itu.edu.tr/assets/downloads/PleiadesUserGuide.pdf](https://www.cscrs.itu.edu.tr/assets/downloads/PleiadesUserGuide.pdf)).**

Table 3:

Threshold for classifying outliers

**We will change "detecting" to "classifying".**

**Relevant changes**

We have implemented everything we answered to the reviewers. The most important points are summarized below:

- Proof reading and overall improvement of writing
- Improvement of the result and conclusion section
- Improvement of the Figures, 2 figures moved to the supplements
- Check of the references

[revised manuscript text omitted]

10   Therefore, we produced a snow depth map of the entire Dischma valley  with the summer ALS surface as a reference for the _winter_  satellite _and_ airplane _imagery_ and is shown in (Figure 10. Based on the performance  _in_ comparison 1 and comparison 2 we use _the airplane data_ as a ground truth to assess the accuracy of the satellite snow depth map.

15   We have a high RMSE (2.2 m) and STD (2.2 m) (summarized in Table 6 as well as a large difference between MBE (-0.02 m) and MdBE (-0.18 m) when using all data.  After filtering, the RMSE (0.92 m) and STD (0.9 m) _reduce considerably_

[revised manuscript text omitted]